# Improved rate for Locally Differentially Private Linear Bandits

## Abstract

In this paper, we propose a stochastic linear contextual bandit algorithm that ensures local differential privacy (LDP). Our algorithm is $(\epsilon, \delta)$−Locally Differentially Private and guarantees $\tilde{O}\left(\sqrt{d}T^{3/4}\right)$ regret with high probability . This is a factor of $d^{1/4}$ improvement over the previous state-of-the-art (Zheng et al., 2020). Furthermore, our regret guarantee improves to $\tilde{O}\left(\sqrt{dT}\right)$ when the action space is well-conditioned. This regret bound matches the optimal non-private asymptotic bound, thus demonstrating that we can achieve privacy for free even in the stringent LDP model. Our algorithm is the first algorithm that achieves $\tilde{O}(\sqrt{T})$ regret in a privacy setting that is stronger than the central settings.

## 1 Introduction

The stochastic linear contextual bandit problem consists of a sequence of $T$ "rounds" of interaction between a learner and an environment. In the $t$th round, a learner receives context $c_t$, which determines a *decision set* $D_t \coloneqq \{\phi(c_t, a) | a \in \mathcal{A}\} \subset \mathbb{R}^d$ where $\mathcal{A}$ is a set of possible actions and $\phi(\cdot, \cdot)$ is a function that maps context-action pairs to $\mathbb{R}^d$. Then the learner chooses an "action" $a_t \in \mathcal{A}$ which corresponds to a "decision" $x_t \coloneqq \phi(c_t, a_t) \in D_t$, and receives a reward $y_t$ such that $\mathbb{E}[y_t | c_t, a_t] = \langle \theta^\star, x_t \rangle$ for some unknown $\theta^\star \in \mathbb{R}^d$. Similar to other bandit settings, we measure the performance of our algorithm by evaluating the regret, defined as the gap between the cumulative rewards of our algorithm and the best possible cumulative reward:

$$Regret_T = \sum_{t=1}^{T} \left[ \max_{a \in \mathcal{A}} \langle \theta^\star, \phi(c_t, a) \rangle - \langle \theta^\star, \phi(c_t, a_t) \rangle \right]$$

$$= \sum_{t=1}^{T} \left[ \max_{x \in D_t} \langle \theta^\star, x \rangle - \langle \theta^\star, x_t \rangle \right]$$

Our goal is to come up with an algorithm that achieves sublinear regret ($R_T \leq o(T)$), which means that on average we are doing as well as the best possible actions in hindsight. This problem has been well-studied in the literature, and the asymptotically optimal regret is $O(d\sqrt{T})$ (Lattimore & Szepesvári, 2020; Li et al., 2019). Furthermore, we want to ensure that our algorithm enforces a *privacy* guarantee - which we will quantify via the framework of local differential privacy (LDP).

To motivate this contextual bandit setting and the need to ensure privacy, consider a personalized medical app where each user has their own treatment plan based on their medical history and the weekly data they provide to the central server (app provider). This application can be modeled as a contextual bandit problem by letting the medical history/weekly data be the context, the treatment plan be the action, and the user's health outcome be the reward. Another example usage of the linear contextual bandit setting is the personalized recommendation system in streaming services. In this example, the context could be a feature vector that contains the user's personal information (age, demography, past activities), the action is the movie the system recommends, and the reward could be the rating the user gives. It is clear in these scenarios that the context/action and the reward are sensitive information that the user wishes to protect. Thus, in order to maximize outcomes and user's experience, we need a contextual bandit algorithm, while in order to protect sensitive information we need a differential private algorithm to ensure privacy.

Our work will ensure a "local" model of privacy, in contrast to a weaker "central" model. In a central privacy model, the users send their raw data to the central server that will be responsible for making decisions. Then the server would inject sufficient noise into its decisions so that potential attackers cannot tell if a particular user's data is used by the server or not. Even though this central model provides protection from outside attackers, it requires the user to have complete trust in the server. Another way to protect users' privacy is the local model, which is the model that we consider in this paper. In this model, before sending their data to the server, each user would inject noise into their own data. Thus, the local model ensures that every user's data is safe without relying on any external sources. Consider a framework with $T$ local users, each holding their private data, and a central server that collects information from these users. Each user applies a randomized mechanism to their data before sending it to the server. The system satisfies $(\epsilon, \delta)-$Locally Differential Privacy (LDP) if the following condition holds:

**Definition 1.1.** (Local Differential Privacy (Dwork & Roth, 2014)) A randomized algorithm $M\colon \mathcal{X} \mapsto \mathcal{S}$ satisfies $(\epsilon, \delta)-$ local differential privacy ($(\epsilon, \delta)$-LDP) if for any pair of users $x, x' \in \mathcal{X}$ and any event $E \subseteq S$, it holds that:

$$P\left[M(x) \in E\right] \leq \exp(\epsilon)P\left[M(x') \in E\right] + \delta$$

Roughly speaking, LDP ensures that the output of a randomized algorithm on any pair of users would be *almost* indistinguishable with high probability. This local model is a stronger notion of privacy than regular DP in the sense that any algorithm that satisfies LDP also satisfies regular DP (Dwork et al., 2010). Furthermore, LDP is also a more user-friendly notion of privacy than regular DP since it allows the user to protect their own data without relying on a trusted server, making it appealing for real-life application (Cormode et al., 2018). However, it is also a lot more difficult to recover the asymptotically optimal $\tilde{O}(\sqrt{T})$ regret guarantee since the private mechanism in the central model (Shariff & Sheffet, 2018) fails in the local model. Indeed, the current best regret guarantee for LDP is only $\tilde{O}\left((dT)^{3/4}/\sqrt{\epsilon} + d\sqrt{T}\right)$ (Zheng et al., 2020). Another line of work on private stochastic contextual linear bandit is the shuffle model (Erlingsson et al., 2019; Cheu et al., 2019). In this model, there exists a trusted shuffler between the users and the server. The shuffler receives noisy data from the users, and permutes them before sending the data to the server. This shuffling step adds another layer of protection which allows for finer privacy-utility trade-offs compared to the local model. In this model, recent works by (Chowdhury & Zhou, 2022) and (Garcelon et al., 2022) achieve the regret of $\tilde{O}(dT^{3/5})$ and $\tilde{O}(dT^{2/3})$ respectively. However, both of these works fail to achieve the $\tilde{O}\left(\sqrt{T}\right)$ regret in any setting. Furthermore, since both works rely on privacy amplification by shuffling, their guarantees only work for $\epsilon \leq O(1/T^{3/10})$ and $\epsilon \leq O(1/T^{1/4})$ respectively, which are a lot smaller than what is used in practice (which is typically $O(1)$). Therefore, a question naturally arises:

*Is it possible to achieve $\tilde{O}\left(\sqrt{T}\right)$ regret in a stronger privacy setting than the central model?*

In this paper, we will provide sufficient conditions under which the answer to the above question is *yes*.

**Contributions.** We introduce a new private variant of the LinUCB algorithm (Chu et al., 2011; Abbasi-yadkori et al., 2011) where the confidence set is constructed using the predictions of an online learner (Abbasi-Yadkori et al., 2012). By carefully choosing the online learner and the loss function, this new approach allows us to construct tighter confidence sets for the unknown parameter $\theta^\star$ which results in a $O\left(\sqrt{d}T^{3/4}/\epsilon\right)$ regret with high probability, improving the best known bound of $\tilde{O}\left((dT)^{3/4}/\sqrt{\epsilon}\right)$ (Zheng et al., 2020) whenever $\epsilon \geq \frac{1}{\sqrt{d}}$. Further, when the minimum eigenvalue of the expected gram matrix $\mathbb{E}_t[x_t x_t^T]$ is bounded from below, the regret guarantee of the new algorithm improves to $\tilde{O}\left(\sqrt{dT}/\epsilon\right)$, recovering the asymptotic regret guarantee of the central model while guaranteeing LDP. This regret, to the best of our knowledge, is the first $\tilde{O}(\sqrt{T})$ regret guarantee for LDP stochastic contextual linear bandit in any setting. Finally, we test our algorithm in the same experiments as in (Chowdhury & Zhou, 2022) and show that our algorithm has empirical improvements over previous works.

---

**Algorithm 1** Private (Contextual) Online LinUCB

---

1: **Input:** Privacy parameters $\epsilon$, $\delta$, failure parameter $\alpha$, covariance matrix $\Sigma = \mathbb{E}[\eta_{x,t}\eta_{x,t}^T]$, minimum eigenvalue $\lambda_{min}$ of $\mathbb{E}[x_t x_t^T]$, domain diameter $D$, time horizon $T$, threshold $\bar{\lambda}$.
2: Initialize $\theta_1 = 0$, $\tilde{V}_0 = I_{d \times d}$, $\tilde{u}_0 = 0$, $\sigma = 2\sqrt{2\log(1.25/\delta)}/\epsilon$, $\Delta^2 = 0$.
3: **if** $\lambda_{min} \leq \bar{\lambda}$ **then**
4: $\quad$ $\Delta^2 = \bar{\lambda}$
5: **end if**
6: **for** $t = 1 \ldots T$ **do**
7: $\quad$ **Local step performed by user $t$:**
8: $\quad$ Receive $\theta_t$, $\tilde{V}_{t-1}$, $\tilde{u}_{t-1}$ from the server. Construct the confidence set $C_{t-1}$ using Lemma 3.3.
9: $\quad$ $(x_t, \tilde{\theta}_t) = \arg\max_{(x,\theta) \in D_t \times C_{t-1}} \langle x, \theta \rangle$
10: $\quad$ Play $x_t$ and observe reward $y_t$
11: $\quad$ Perturb $x_t$ with a small amount of noise: $\bar{x}_t \leftarrow x_t + \zeta_t$ where $\zeta_t \sim N\left(0, \Delta^2 I_d\right)$.
12: $\quad$ Update $\tilde{x}_t = \bar{x}_t + \eta_{x,t}$, $\tilde{y}_t = y_t + \eta_{y,t}$ where $\eta_{x,t} \sim N(0, \sigma^2 I_d)$, $\eta_{y,t} \sim N(0, \sigma^2)$.
13: $\quad$ Get the loss $l_t(\theta) = (\langle \tilde{x}_t, \theta_t \rangle - \tilde{y}_t)^2 - \|\theta\|_\Sigma^2$
14: $\quad$ Compute $g_t = 2\tilde{x}_t \left(\langle \tilde{x}_t, \theta_t \rangle - \tilde{y}_t\right) - 2\Sigma\theta_t$
15: $\quad$ Send $\tilde{x}_t$, $\tilde{y}_t$, and $g_t$ to the server.
16: $\quad$ **Server step:**
17: $\quad$ Sent $\theta_t$, $\tilde{x}_t$, $\tilde{y}_t$, and $g_t$ to Maler (Algorithm 5) and get back $\theta_{t+1}$.
18: $\quad$ Update the history $\{\theta_1, \ldots, \theta_t\} \cup \{\theta_{t+1}\}$
19: $\quad$ Update $\tilde{V}_t = \tilde{V}_{t-1} + \tilde{x}_t \tilde{x}_t^T$, $\tilde{u}_t = \tilde{u}_{t-1} + \langle \theta_t, \tilde{x}_t \rangle \tilde{x}_t$
20: **end for**
21: **return** $\tilde{x}_1, \ldots, \tilde{x}_T$ and $\tilde{y}_1, \ldots, \tilde{y}_T$

---

## 2 Problem Setup

Let $\mathcal{A}$ be an action space and $\mathcal{C}$ a context space. In the stochastic contextual linear bandit setting that we are considering, in every round $t \in [T]$, the learner receives an i.i.d random context $c_t \in \mathcal{C}$ and a function $\phi(\cdot, \cdot) : \mathcal{C} \times \mathcal{A} \mapsto \mathbb{R}^d$, and picks an action $a_t$ corresponding to $x_t := \phi(c_t, a_t) \in \mathcal{D}_t$. The learner then receives a random reward $y_t = \langle \theta^\star, x_t \rangle + \eta_t$ where $\eta_t$ is a zero-mean and independent $R^2-$ subgaussian random variable where $R$ is a positive constant. Then we define the regret as:

$$Regret_T = \sum_{t=1}^T \left[ \max_{x \in D_t} \langle \theta^\star, x \rangle - \langle \theta^\star, x_t \rangle \right]$$

Our goal is to design an algorithm that achieves sublinear regret bound and guarantees that the actions and rewards sequences $\{(x_1, y_1), \ldots, (x_T, y_T)\}$ are LDP. For $n \in \mathbb{N}$, we denote the set $\{1, \ldots, n\}$ as $[n]$. We use the standard big-$O$ notation to hide constants and $\tilde{O}$ to hide additional logarithmic factors. Throughout the paper, $\|\cdot\|$ is used to indicate the Euclidean norm unless specified otherwise. A symmetric matrix $M \in \mathbb{R}^{d \times d}$ is a positive-semidefinite matrix if $x^T M x \geq 0$ for any $x \in \mathbb{R}^d$ and we define its associated norm as $\|x\|_M = \sqrt{x^T M x}$. We also define $\mathbb{E}_x[\cdot]$ as the expectation over the randomness of some random variable $x$ and $\log(x)$ as the natural logarithm of $x$.

*Assumption*: We assume the reward, and the norm of the parameter $\theta^\star$ and feature map $x_t = \phi(c_t, a_t)$ are all bounded: $\|x_t\| \leq 1$, $|y_t| \leq 1$, $|\langle x_t, \theta^\star \rangle| \leq 1$ for all $t \in [T]$, and $\|\theta^\star\| \leq 1$. We also assume that we have access to $T$ unique users. Note that these are all standard assumptions from the literature (Shariff & Sheffet, 2018; Chowdhury & Zhou, 2022).

# 3  Online LDP LinUCB

Our private method described in Algorithm 1 is a private variant of the LinUCB algorithm (Chu et al., 2011; Abbasi-yadkori et al., 2011). The main task of the algorithm is to derive an ellipsoid confidence set defined as

$$C_{t-1} := \left\{ \theta \in \mathbb{R}^d : \|\theta - V_{t-1}^{-1} u_{t-1}\|_{V_{t-1}} \leq \beta_t \right\} \tag{1}$$

where $V_t = \sum_{i=1}^{t} x_t x_t^T$, $\beta_t$ is the width of the confidence set, and $u_{t-1} = \sum_{i=1}^{t-1} x_i y_i$. Our goal is to pick an appropriate $\beta_t$ such that the optimal parameter $\theta^\star$ is inside the ellipsoid with high probability for all $t \in [T]$. LinUCB identifies $x_t \in D_t$ and $\theta_t \in C_{t-1}$ that maximizes $\langle x, \theta \rangle$ and plays $x_t$. Overall, LinUCB guarantees the following regret:

$$Regret_T \leq \tilde{O}\left( \max_t \beta_t \sqrt{dT} \right)$$

Thus, as long as we can design a tight confidence ellipsoid (small $\beta_t$), our linear bandit algorithm will have a small regret.

To ensure privacy, we unfortunately cannot update our algorithm with the true value of $V_t$ and $u_t$. Instead, we have to use private approximations $\tilde{V}_t$ and $\tilde{u}_t$ to define an analogous $\tilde{C}_{t-1}$. Let $\tilde{V}_t = V_t + H_t$ and $\tilde{u}_t = u_t + h_t$. Assuming $\|H_t\| \leq \rho_{max}$, $\|h_t\|_{H_t^{-1}} \leq \nu$ for some $\rho_{max}, \nu \geq 0$, then from (Shariff & Sheffet, 2018), we know that the confidence width is bounded by:

$$\beta_t \leq \tilde{O}\left( \sqrt{d} + \sqrt{\rho_{max}} + \nu \right) \tag{2}$$

Since higher $\beta_t$ leads to higher regret, we would like to minimize the error measures $\rho_{max}$ and $\nu$ introduced by the private approximations.

We can now shed light on why the regret guarantees for local models are worse than for the central model. In the central model, since the server is allowed to see the raw data $x_t$ and $y_t$, the server can compute $\tilde{V}_t$ and $\tilde{u}_t$ privately using the tree-aggregation mechanism (Chan et al., 2011; Dwork et al., 2010). Then, we have $\rho_{max} \leq \tilde{O}(\sqrt{d}/\epsilon)$ and $Regret_T \leq \tilde{O}(d\sqrt{T} + d^{3/4}\sqrt{T}/\sqrt{\epsilon})$. However, in the local model, since each user perturbs their data before sending them to the server, applying tree-aggregation is off the table. If we naively apply the Gaussian Mechanism with T rounds of compositions, $\rho_{max}$ now is $\tilde{O}(\sqrt{dT}/\epsilon)$ and the regret becomes $\tilde{O}(d\sqrt{T} + d^{3/4}T^{3/4}/\sqrt{\epsilon})$.

In this section, we propose a new approach for designing the confidence sets LDP LinUCB. Our approach is based on the *online-to-confidence-set conversion* in (Abbasi-Yadkori et al., 2012) where the main idea is that the predictions of any online algorithm that predicts the responses of the chosen inputs in a sequential manner can be "converted" to a confidence set. In each round $t$, the online algorithm will receive $x_t$, $y_t$, predict $\theta_t$, and suffer the loss $l_t(\theta_t) = (\langle \theta_t, x_t \rangle - y_t)^2$. The goal of the online learner is to discover the "true" value $\theta_\star$. We measure its performance via its own notion of regret ($Regret_{OL}$), and we define $M_T$ to be a known upper-bound on $Regret_{OL}$.

$$Regret_{OL} \triangleq \sum_{t=1}^{T} l_t(\theta_t) - l_t(\theta^\star) \tag{3}$$

$$M_T \geq Regret_{OL} \tag{4}$$

Intuitively, a low regret means that the online learner is able to predict a good approximate of the optimal $\theta^\star$. Now, (Abbasi-Yadkori et al., 2012) show how to use the bound $M_T$ to construct a confidence set with the width bounded by:

$$\beta_T \leq \tilde{O}\left( \sqrt{M_T} \right)$$

Since the width of the confidence ellipsoid depends on the regret of the online learner, one could hope that with carefully designed online learner and loss function, the confidence width would be small and we

can see improvements in the final regret bound for LinUCB. We will now show that this is indeed the case and Algorithm 1 using this approach can achieve $\tilde{O}(\sqrt{dT})$ regret when the second-moment matrix $\mathbb{E}[x_t x_t^T] \succeq \lambda_{min} I$ for some $\lambda_{min} \approx O(1)$ (refer to Remark 3.7 for more details).

Algorithm 1 is based on the Online LinUCB algorithm (Abbasi-Yadkori et al., 2012) (for more details, refer to Section A.2). In every round $t$, a unique local user $t$ receives some private information from the server that they can use to construct the confidence set $C_t$. Then the user chooses $x_t$ that maximizes the upper confidence bound $\max_{\theta \in C_{t-1}} \langle x, \theta \rangle$ (step 9) and receives reward $y_t$. If $x_t$ is not well-conditioned ($\lambda_{min}$ is smaller than some threshold $\bar{\lambda}$ in step 3), the user perturbs $x_t$ with a small amount of noise in step 11 to make sure the loss $l_t(\theta)$ that is sent to the online learner is strongly-convex in expectation. Finally, user $t$ perturbs $x_t$, $y_t$ with Gaussian noise to maintain LDP, computes the gradient using private information, and sends $\tilde{x}_t$, $\tilde{y}_t$, and $g_t$ to the server. The server then uses this new information to update the prediction of the online learner ($\theta_{t+1}$), and the history $\tilde{V}_{t+1}$ and $\tilde{u}_t$ so that the next user can make a better decision.

Our algorithm elaborates on this Online LinUCB strategy with two key ideas. First, instead of using the intuitive squared loss $l_t(\theta) = (\langle \theta, x_t \rangle - y_t)^2$, we use the more peculiar choice $l_t(\theta) = (\langle \tilde{x}_t, \theta_t \rangle - \tilde{y}_t)^2 - \|\theta\|_\Sigma^2$ for some to-be-specified $\Sigma$. Second, we employ the advanced online learning algorithm Maler (Wang et al., 2020b) as the online learner.

The reason for the choice of the loss is a bit technical. Intuitively, if we want the online learner to accurately approximate $\theta^\star$, we want $\theta^\star$ to be the minimizer of the loss $l_t(\theta)$. However, due to the noise in $x_t$ and $y_t$, $\theta^\star$ is not the minimizer of the square loss $(\langle \tilde{x}_t, \theta_t \rangle - \tilde{y}_t)^2$. To counteract this issue, we incorporate a negative regularizer term, $-\|\theta\|_\Sigma^2$, which serves to neutralize the variance introduced by the privacy noise in $x_t$. Now, with the added regularizer, $\theta^\star$ becomes the minimizer of $\mathbb{E}[l_t(\theta)]$. At first glance, this new loss appears to be intractable because it is non-convex. However, it is convex *in expectation*, which is sufficient to guarantee an $O(\sqrt{T})$ regret. This in turn translates to $\tilde{O}(T^{3/4})$ for the final regret bound.

The online learner can potentially do even better than $O(\sqrt{T})$ regret in certain favorable settings. Specifically, when $\mathbb{E}[x_t x_t^T] \succeq \lambda_{min} I$ for $\lambda_{min} > 0$, $l_t(\theta)$ is *strongly convex in expectation*, despite not being convex. This necessitates the use of an online learner capable of adapting to such advantageous scenarios and refining the regret to $O(\log T)$. This is precisely the scenario where Maler proves invaluable. Maler (described in Algorithm 5) is an online learner that adjusts to achieve the optimal regret across various types of loss functions, including convex, strongly convex, and exp-concave. We demonstrate that implementing Maler with our loss function yields a dimension-independent regret of $O(\log T)$ with high probability. This allows us to attain a regret of $\tilde{O}(\sqrt{dT})$, which is not only asymptotically optimal but also improves upon the non-private worst-case regret guarantee for general settings by an order of $O(\sqrt{d})$ (see Section 3.2 for more discussion).

*Remark* 3.1. Let us discuss one specific example when the condition $\mathbb{E}[x_t x_t^T] \succeq \lambda_{min} I$ is satisfied, giving our algorithm the optimal regret guarantee of $\tilde{O}(\sqrt{dT})$. Assuming we have $k$ available actions, and let each action be $x_i \sim N(0, \sigma^2 I_d)$. Then, for the condition $\mathbb{E}[x_t x_t^T] \succeq \lambda_{min} I$ for some $\lambda_{min} \approx O(1)$ to be true, we need to show that $\mathbb{E}[v^T x_t x_t^T v] \geq C$ where $C$ is a positive constant and $v$ is a unit vector. Notice that $\mathbb{E}[v^T x_t x_t^T v] \geq \mathbb{E}[\min_{1 \leq i \leq k} v^T x_i x_i^T v]$, thus if we can show a constant lower bound for $\mathbb{E}[\min_{1 \leq i \leq k} v^T x_i x_i^T v]$ then we are done. We have $v^T x_i \sim N(0, \sigma^2)$ (since $v$ is a unit vector) for every $i \in [k]$, thus by Lemma H.15, $P[|x_i| \leq t] \leq \frac{\sqrt{2}}{\sigma \sqrt{\pi}} t$ for $t > 0$. Then, we can apply Theorem H.14 to get $\mathbb{E}[\min_{1 \leq i \leq k} v^T x_i x_i^T v] = E[\min_{1 \leq i \leq k} |\langle v, x_i \rangle|^2] \geq \frac{\sigma^2 \pi}{6k^2}$. Thus, as long as the number of actions is not too large, this example would fall under our favorable setting.

Before we prove the utility guarantee of Algorithm 1, let us first show that Algorithm 1 is $(\epsilon, \delta)-$LDP.

**Theorem 3.2.** *(Privacy Guarantee) Algorithm 1 guarantees $(\epsilon, \delta)-$LDP.*

*Proof.* Let us define the local step of Algorithm 1 as the local mechanism $M_t$. Let $x_t'$ and $y_t'$ be the action and reward of a new user at time t. By the boundedness assumption, we have $\max_{x_t, x_t' \in \mathcal{X}} \|x_t - x_t'\| \leq 2$ and $\max_{y_t, y_t' \in \mathcal{Y}} |y_t - y_t'| \leq 2$ for all $t \in [T]$. Thus, by the classic Gaussian Mechanism in (Dwork et al., 2010), the outputs $\tilde{x}_t$ and $\tilde{y}_t$ of the local mechanism $M_t$ satisfy $(\epsilon, \delta)-$LDP for all $t$. Further, since $\theta_t$ and $g_t$ are computed using a sequence of private parameters $\tilde{x}_1, \tilde{y}_1, \ldots, \tilde{x}_{t-1}, \tilde{y}_{t-1}$ and no other sensitive information (we only

want to protect $\{(x_1, y_1), \ldots, (x_t, y_t)\}$), $\theta_t$ and $g_t$ also satisfy $(\epsilon, \delta)-$LDP by post-processing. Consequently, Algorithm 1 guarantees $(\epsilon, \delta)-$LDP for every user $t \in [T]$, as each local mechanism $M_t$ is $(\epsilon, \delta)-$LDP. $\qquad\square$

To make the analysis more succinct and easier to follow, let us define the "good event" $\mathcal{E}$ as in Section B in the Appendix. Roughly speaking, $\mathcal{E}$ is the event in which a small number of standard martingale concentration bounds hold simultaneously. Then, from Lemma B.1, we know that the good event $\mathcal{E}$ happens with high probability. Now, we can show the following result on the confidence set.

**Lemma 3.3.** *We define $\tilde{V}_{N-1} = \sum_{t=1}^{N-1} \tilde{x}_t \tilde{x}_t^T$, $\tilde{u}_{N-1} = \sum_{t=1}^{N-1} \langle \theta_t, \tilde{x}_t \rangle \tilde{x}_t$ ($\theta_t$ is the prediction of the online learner), and $\hat{\theta}_N = \tilde{V}_{N-1}^{-1} \tilde{u}_{N-1}$. Assuming $\|\theta_t\| \leq D$, then under event $\mathcal{E}$, the true parameter $\theta^\star$ lies in the set:*

$$C_{N-1} = \left\{ \theta \in R^d : \|\theta - \hat{\theta}_N\|_{\tilde{V}_{N-1}}^2 \leq M_N + K_N \right\}$$

*for any $N \geq 1$ and*

$$K_N = \gamma D \Delta^2 \log(T/\alpha) \sqrt{N \sum_{t=1}^{N} \|\theta_t - \theta^\star\|^2} + \gamma \left( R + \frac{D\sqrt{\log(1/\delta)}}{\epsilon} + D\Delta \right)^2 \log(T/\alpha)$$

$$+ \gamma \left( \frac{\log(1/\delta)}{\epsilon^2} + \Delta^2 \right) \log(T/\alpha) \sum_{t=1}^{N} \|\theta_t - \theta^\star\|^2$$

*for a sufficient large constant $\gamma > 0$.*

We now provide a sketch of the proof of Lemma 3.3 below. For the full proof, refer to section D in the appendix.

*Proof.* Our proof follows the proof of Theorem 1 in (Abbasi-Yadkori et al., 2012). From our definition of $M_N$ and the loss $l_t$, we have:

$$M_N \geq \sum_{t=1}^{N} (\langle \tilde{x}_t, \theta_t \rangle - \tilde{y}_t)^2 - \|\theta_t\|_\Sigma^2 - (\langle \tilde{x}_t, \theta^\star \rangle - \tilde{y}_t)^2 + \|\theta^\star\|_\Sigma^2$$

Plugging in $\tilde{y}_t = \langle x_t, \theta^\star \rangle + r_t + \eta_{y,t}$ and $\tilde{x}_t = x_t + \zeta_t + \eta_{x,t}$:

$$= \sum_{t=1}^{N} (\langle x_t, \theta_t - \theta^\star \rangle + \langle \eta_{x,t}, \theta_t \rangle + \langle \zeta_t, \theta_t \rangle - r_t - \eta_{y,t})^2 - \|\theta_t\|_\Sigma^2 - (\langle \eta_{x,t}, \theta^\star \rangle + \langle \zeta_t, \theta^\star \rangle - r_t - \eta_{y,t})^2 + \|\theta^\star\|_\Sigma^2$$

Let us denote $z_t = r_t + \eta_{y,t}$. Now expanding the squares and rearranging the terms we get:

$$\sum_{t=1}^{N} (\langle x_t, \theta_t - \theta^\star \rangle)^2 \leq M_N + \sum_{t=1}^{N} \underbrace{-2(\langle \eta_{x,t} + \zeta_t, \theta_t \rangle - z_t)\langle x_t, \theta_t - \theta^\star \rangle}_{A_t} \underbrace{+2z_t \langle \eta_{x,t} + \zeta_t, \theta_t - \theta^\star \rangle}_{B_t} \underbrace{-\langle \zeta_t, \theta_t \rangle^2 + \langle \zeta_t, \theta^\star \rangle^2}_{C_t}$$

$$\underbrace{-2(\langle \zeta_t, \theta_t \rangle \langle \eta_{x,t}, \theta_t \rangle - \langle \zeta_t, \theta^\star \rangle \langle \eta_{x,t}, \theta^\star \rangle)}_{D_t} \underbrace{-(\theta_t - \theta^\star)^T (\eta_{x,t} \eta_{x,t}^T - \Sigma)(\theta_t + \theta^\star)}_{E_t} \tag{5}$$

Under event $\mathcal{E}$, we have

$$A_t + B_t = -\sum_{t=1}^{N} 2(\langle \eta_{x,t} + \zeta_t, \theta_t \rangle - z_t)\langle x_t, \theta_t - \theta^\star \rangle + 2z_t \langle \eta_{x,t} + \zeta_t, \theta_t - \theta^\star \rangle$$

$$\leq \tilde{O} \left( \sqrt{\sum_{t=1}^{N} (\langle x_t, \theta_t - \theta^\star \rangle)^2} + \sqrt{\sum_{t=1}^{N} \langle \eta_{x,t} + \zeta_t, \theta_t - \theta^\star \rangle^2} \right) \tag{6}$$

Notice that the first sum in the right-hand side of Eq.6 is exactly the sum in the left-hand side of Eq.5. Thus, we can use Proposition H.8 and H.9 to bound this term. For the second term in the right-hand side of Eq.6, we can again use the fact that we are under the good event $\mathcal{E}$ to control the sum.

The sum of $C_t$ and $D_t$ can be written as follows:

$$\sum_{t=1}^{N}\langle\zeta_t,\theta^\star\rangle^2 - \langle\zeta_t,\theta_t\rangle^2 + 2(\langle\zeta_t,\theta^\star\rangle\langle\eta_{x,t},\theta^\star\rangle - \langle\zeta_t,\theta_t\rangle\langle\eta_{x,t},\theta_t\rangle) = \sum_{t=1}^{N}\langle\zeta_t,\theta^\star-\theta_t\rangle\langle\zeta_t,\theta^\star+\theta_t\rangle$$
$$+ 2(\theta^\star-\theta_t)^T\zeta_t\eta_{x,t}^T(\theta^\star-\theta_t) + 2\theta_t^T\zeta_t r_t^T(\theta^\star-\theta_t) + 2(\theta^\star-\theta_t)^T\zeta_t r_t^T\theta_t$$

Using norm bound of Gaussian random vector and corollary H.12:

$$C_t + D_t \leq O\left(D\Delta^2\log(T/\alpha)\sqrt{N\sum_{t=1}^{N}\|\theta_t-\theta^\star\|^2}\right) + O\left(\frac{D\Delta\sqrt{\log(1/\delta)}}{\epsilon}\sqrt{\log(T/\alpha)\sum_{t=1}^{T}\|\theta_t-\theta^\star\|^2}\right)$$

For the term $E_t$ in Eq.5, since $\mathbb{E}\left[\eta_{x,t}\eta_{x,t}^T\right] = \Sigma$, it is a Martingale difference sequence and by Theorem H.2 we have:

$$|\sum_{t=1}^{N}(\theta_t-\theta^\star)^T(\eta_{x,t}\eta_{x,t}^T - \Sigma)(\theta_t+\theta^\star)| \leq \tilde{O}\left(\frac{D\log(1/\delta)\log(T/\alpha)}{\epsilon^2}\sqrt{\sum_{t=1}^{N}\|\theta_t-\theta^\star\|^2}\right)$$

Now we can combine the bounds of all the terms and use Proposition H.8 to get:

$$\sum_{t=1}^{N}(\langle\tilde{x}_t,\theta_t-\theta^\star\rangle)^2 \leq M_N + K_N$$

Let us denote the set $C_{N-1}$ as the ellipsoid underlying the covariance matrix $\tilde{V}_{N-1} = I + \sum_{t=1}^{N-1}\tilde{x}_t\tilde{x}_t^T$ and centering at

$$\hat{\theta}_N = \arg\min_{\theta\in R^d}\left(\|\theta\|_2^2 + \sum_{t=1}^{N-1}(\langle\tilde{x}_t,\theta_t-\theta\rangle)^2\right)$$
$$= \tilde{V}_{N-1}^{-1}\left(\sum_{t=1}^{N-1}\langle\theta_t,\tilde{x}_t\rangle\tilde{x}_t\right)$$
$$= \tilde{V}_{N-1}^{-1}\tilde{u}_{N-1}$$

We can thus express the ellipsoid as:

$$\hat{C}_{N-1} = \left\{\theta\in R^d : (\theta-\hat{\theta}_N)^T\tilde{V}_{N-1}(\theta-\hat{\theta}_N) + \|\hat{\theta}_N\|_2^2 + \sum_{t=1}^{N-1}(\langle\tilde{x}_t,\theta_t-\hat{\theta}_N\rangle)^2 \leq M_N + K_N\right\}$$

The ellipsoid is contained in a larger ellipsoid

$$\hat{C}_{N-1} \subseteq C_{N-1} = \left\{\theta\in R^d : \|\theta-\hat{\theta}_N\|_{\tilde{V}_{n-1}}^2 \leq M_N + K_N\right\}$$

Thus, $\theta^\star$ lies in $C_{N-1}$ with high probability. $\qquad\square$

With Lemma 3.3 in hand, we can show a general regret bound:

**Theorem 3.4.** *(Utility guarantee) Recall that $M_T$ is the regret of our online learner (see equation (4)), and $K_T$ is as defined in Lemma 3.3. Under event $\mathcal{E}$, the regret of Algorithm 1 is:*

$$Regret_T \leq \tilde{O}\left(\sqrt{M_T + K_T}\sqrt{2Td\log\left(1 + \frac{T}{d}\right)}\right)$$

As we can see, this regret bound is quite similar to the regret bound of its non-private counterpart assuming $M_T$ and $K_T$ can be controlled. Now we show that at worst, this bound is $\tilde{O}\left(\sqrt{d}T^{3/4}/\epsilon\right)$ which is $O\left(d^{1/4}\right)$ improvement over the current best known bound for LDP Stochastic Linear Bandit whenever $\epsilon \geq \frac{1}{\sqrt{d}}$. Then, we show that in certain settings, this bound becomes $\tilde{O}(\sqrt{dT})$, which to the best of our knowledge, is the new state-of-the-art bounds for any stronger privacy model than the central model.

## 3.1 $\tilde{O}(\sqrt{dT})$ regret bound

From Theorem 3.4, it is clear that if we want to have $\tilde{O}(\sqrt{T})$ regret, we need our online learner to have logarithmic regret i.e $M_T = O\left(\log T\right)$. However, for this to be true, one might think that we need our loss to be either strongly convex or exp-concave with a sufficiently large strong-convexity/exp-concavity constant. Unfortunately, the loss $l_t(\theta) = (\langle \tilde{x}_t, \theta_t \rangle - \tilde{y}_t)^2 - \|\theta\|_\Sigma^2$ does not fall into either of these family of functions. Surprisingly, it may still be possible to guarantee low regret with high probability using $l_t(\theta_t)$. Denote $L(\theta_t) = \mathbb{E}[l_t(\theta_t)]$. Then:

$$\nabla L(\theta_t) = \mathbb{E}\left[2(\bar{x}_t + \eta_{x,t})(\langle \bar{x}_t, \theta_t \rangle + \langle \eta_{x,t}, \theta_t \rangle - y_t - \eta_{y,t}) - 2\Sigma\theta_t\right]$$

Since $\eta_{x,t}, \eta_{y,t}$ are zero-mean and $\bar{x}_t, \eta_{x,t}, \eta_{y,t}$ are independent:

$$\nabla L(\theta_t) = \mathbb{E}[2\bar{x}_t(\langle \bar{x}_t, \theta_t \rangle - y_t) + 2\eta_{x,t}\langle \eta_{x,t}, \theta_t \rangle - 2\Sigma\theta_t]$$
$$\Rightarrow \nabla^2 L(\theta_t) = \mathbb{E}[2\bar{x}_t\bar{x}_t^T] = \mathbb{E}[2x_t x_t^T + 2\zeta_t\zeta_t^T]$$

Since $x_t x_t^T$ is a positive semi-definite matrix for all $t \in [T]$, we have $\mathbb{E}[x_t x_t^T] \succeq \lambda_{min} I_d$ for some $\lambda_{min} \geq 0$. Thus, $l_t(\theta_t)$ is $\mu$−strongly convex in expectation where $\mu = 2\left(\lambda_{min} + \Delta^2\right)$. This is great news since even though $l_t(\theta_t)$ is not strongly convex, we can still show that the online learner Maler guarantees logarithmic regret with high probability using the following lemma:

**Lemma 3.5.** *(Strongly convex regret) Assuming $L(\theta) = E[l_t(\theta_t)]$ is $\mu$−strongly convex and $\max_{t,t'} \|\theta_t - \theta'_t\| \leq 2D$. Then w.p at least $1 - \alpha$, Maler (Algorithm 5) under the event $\mathcal{E}$ guarantees:*

$$\sum_{t=1}^T l_t(\theta_t) - l_t(\theta^\star) \leq O\left(\left(\frac{G^2}{\mu} + GD\right)\log T + \frac{G^2}{\mu}\log(1/\alpha)\right)$$

*Furthermore, we have:*

$$\sum_{t=1}^T \|\theta_t - \theta^\star\|^2 \leq O\left(\left(\frac{G^2}{\mu^2} + \frac{GD}{\mu}\right)\log T + \frac{G^2}{\mu^2}\log(1/\alpha)\right)$$

The proof for Lemma 3.5 is provided in Section C in the Appendix. Notice that from this lemma, we immediately have a high probability bound for the confidence ellipsoid in Lemma 3.3. Specifically, with $\alpha \geq \frac{1}{T}$, we have:

$$M_T \leq O\left(\left(\frac{G^2}{\lambda_{min} + \Delta^2} + GD\right)\log T\right)$$

And,

$$K_T = O\left(D\Delta^2\log(T/\alpha)\sqrt{T\log T\left(\frac{G^2}{(\lambda_{min} + \Delta^2)^2} + \frac{GD}{\lambda_{min} + \Delta^2}\right)}\right)$$
$$+ O\left(\left(R + \frac{D\sqrt{\log(1/\delta)}}{\epsilon} + D\Delta\right)^2\log(T/\alpha)\right)$$
$$+ O\left(\log^2 T\left(\frac{\log(1/\delta)}{\epsilon^2} + \Delta^2\right)\left(\frac{G^2}{(\lambda_{min} + \Delta^2)^2} + \frac{GD}{\lambda_{min} + \Delta^2}\right)\right)$$

Let $H = \frac{G^2}{(\lambda_{min}+\Delta^2)^2} + \frac{GD}{\lambda_{min}+\Delta^2}$. Then:

$$K_T \leq O\left(D\Delta^2 \log(T/\alpha)\sqrt{TH\log T} + \left(R + \frac{D\sqrt{\log(1/\delta)}}{\epsilon}\right)^2 \log(T/\alpha) + \frac{H\log(1/\delta)}{\epsilon^2}\log^2 T\right)$$

Now plugging $K_T$ and $M_T$ into Theorem 3.4 we get the following corollary:

**Corollary 3.6.** *Assuming $\|\theta_t\| \leq D$. Under event $\mathcal{E}$, for any $\lambda_{min} \geq 0$ such that $\mathbb{E}[x_t x_t^T] \succeq \lambda_{min} I_{d \times d}$ and $\alpha \geq 1/T$, the regret of Algorithm 1 with Maler as the online learner and with threshold $\lambda = \frac{1}{T^{1/4}}$ is upper bounded by*

$$O\left(\left(\left(R + \frac{D\sqrt{\log(1/\delta)}}{\epsilon}\right)\sqrt{\log(T/\alpha)} + \sqrt{D\Delta^2\log(T/\alpha)\sqrt{TH\log T}} + \frac{\log T\sqrt{H\log(1/\delta)}}{\epsilon}\right)\right.$$
$$\left. \times \sqrt{dT\log(T/d)}\right)$$
$$\leq \tilde{O}\left(\min\left\{\frac{\sqrt{d}T^{3/4}}{\epsilon}, \frac{\sqrt{dT}}{\epsilon\lambda_{min}}\right\}\right)$$

*where $H = \frac{G^2}{(\lambda_{min}+\Delta^2)^2} + \frac{GD}{\lambda_{min}+\Delta^2}$.*

Let us discuss the implication of the regret bound in Corollary 3.6. Consider the best-case scenario, which is when $\lambda_{min} = O(1)$ (e.g. $x_t$ follows some random distribution with constant variance). Then, $\Delta^2 = 0$ and Algorithm 1 is $(\epsilon, \delta)-$LDP and guarantees $\tilde{O}(\sqrt{dT}/\epsilon)$ regret with high probability. Thus, Algorithm 1 guarantees the same regret bound asymptotically of non-private LinUCB. Further, by running our online learner on a bounded domain, our online regret depends on the radius $D$ of the domain (which is set by the user) rather than the dimension $d$ of the feature vector $x_t$. As a result, we are able to improve the dimension dependence from $O(d^{3/4})$ in previous works to $O(\sqrt{d})$. Our regret bound would get worse as $\lambda_{min}$ decreases. However, when $\lambda_{min} \leq \frac{1}{T^{1/4}}$, notice that now $\Delta^2 = \frac{1}{T^{1/4}}$ and Algorithm 1 guarantees $\tilde{O}(\sqrt{d}T^{3/4}/\epsilon)$ regret, which is the same asymptotic regret as the current best regret for LDP (contextual) linear bandit (Shariff & Sheffet, 2018) but with improved dimension dependence. Overall, the regret of Algorithm 1 is always between $\tilde{O}(\sqrt{dT}/\epsilon)$ and $\tilde{O}(\sqrt{d}T^{3/4}/\epsilon)$.

*Remark* 3.7. Since the regret in (Chowdhury & Zhou, 2022) is $\tilde{O}(dT^{3/5})$, Algorithm 1 has a better regret as long as $\lambda_{min} \geq \Omega\left(\frac{1}{T^{1/10}}\right)$ while also providing stronger privacy guarantee and having no restriction on $\epsilon$.

### 3.2 Comparisons with previous results

In the worst-case scenario ($\lambda_{min} \leq \frac{1}{T^{1/4}}$), Algorithm 1 provides a regret bound of $\tilde{O}(\sqrt{d}T^{3/4}/\epsilon + \sqrt{dT})$ with high probability. This surpasses the state-of-the-art result for LDP stochastic linear bandit, which is $\tilde{O}((dT)^{3/4}/\sqrt{\epsilon} + d\sqrt{T})$ whenever $\epsilon \geq \frac{1}{\sqrt{d}}$ (which covers many practical scenarios where $\epsilon$ is typically larger than 1). Although both exhibit the same asymptotic regret of $\tilde{O}(T^{3/4})$, Algorithm 1 demonstrates superior dimension dependence in both private and non-private terms. The key to this improvement lies in the unique approach of Algorithm 1 concerning noise injection and the employment of an online learner with a dimension-free regret. In (Zheng et al., 2020), the privacy guarantee is achieved by adding a Gaussian matrix $H_t$ to $V_t$ and a Gaussian vector $h_t$ to $u_t$. From the concentration inequality of the Gaussian matrices, $\|H_t\|$ is bounded by $\tilde{O}(\sqrt{dT}/\epsilon)$ with high probability. Thus, plugging this back in Eq. 1 and Eq. 2 yields the $\tilde{O}((dT)^{3/4}/\sqrt{\epsilon} + d\sqrt{T})$ regret. On the other hand, Algorithm 1 injects noise directly to $x_t$ and $y_t$, instead of $V_t$ and $u_t$. Thus, we are not restricted by the $\sqrt{d}$ factor that comes from the concentration bound of the Gaussian matrix. The regret now hinges on the performance of the online learner Maler, which is $O(D\sqrt{T})$ where $D$ is the user-set bound for $\theta_t$. As a result, we are able to improve the dimension dependence to $O(\sqrt{d})$.

In the favorable settings ($\lambda_{min}$ is $O(1)$), the regret of Algorithm 1 is improved to $O(\sqrt{dT})$. Comparatively, this shows a notable advancement over other private algorithms. Specifically, Algorithm 1 outperforms the shuffle model (Chowdhury & Zhou, 2022), which has a regret of $\tilde{O}(dT^{3/5})$, and matches the asymptotic regret of the central model. However, our algorithm demonstrates a more favorable dimension dependence, attributable to

---

**Algorithm 2** Private Bandits Combiner

---

1: **Input:** Receive base learner 1 (Algorithm 9) and base learners $i$ for $i \in (1, M]$(Algorithm 1). Constants $L_1, \ldots, L_M, \alpha_1, \ldots, \alpha_M, R_1, \ldots, R_M, T$ users, failure probability $\alpha$, universal constant $p$, privacy noise variance $\sigma^2$, privacy parameters $\epsilon, \delta$, covariance matrix $\Sigma$, domain diamater $D$, power constant $k$.

2: Initialize base learner 1 with $\epsilon, \delta, \alpha, \Sigma, D, \lambda_{min} = \frac{1}{T^{1/8}}$ and $k = 1/8$.

3: Initialize each base learner $i$ with $\epsilon, \delta, \alpha, \Sigma, D, \lambda_{min} = \lambda_i = \frac{2^{i-1}}{T^{1/8}}$ for $i \in (1, M]$, and $\bar{\lambda} = 1/T^{1/8}$.

4: Set $T(i,0) = 0$ and $\hat{\mu}_0^i = 0$ for all $i$, and set $I_1 = \{1, \ldots, M\}$.

5: Initialize $\theta_1^i = 0$, $\tilde{V}_0^i = I_{d \times d}$, $\tilde{u}_0^i = 0$ for all $i \in [M]$.

6:

7: **for** $t = 1 \ldots T$ **do**

8:     **For the server:**

9:     Set $U(i, t-1) = \hat{\mu}_{T(i,t-1)}^i + \min\left(1, \frac{L_i T(i,t-1)^{\alpha_i} + p\sqrt{T(i,t)(1+\sigma^2)\log\left(\frac{T^3 M \log T(i,t)(1+\sigma^2)}{\alpha}\right)}}{T(i,t-1)}\right) - \frac{R_i}{T}$ for all $i$.

10:     Set $i_t = \arg\max_{i \in I_t} U(i, t-1)$.

11:     **For the local user t:**

12:     Receive the base learner index $i_t$ and $\theta_t^{i_t}$, $\tilde{V}_{t-1}^{i_t}$, $\tilde{u}_{t-1}^{i_t}$ from the server.

13:     User $t$ follows the policy of the base learner $i_t$ and play $x_t^{i_t}$, receive reward $y_t^{i_t}$, and $\theta_{t+1}$.

14:     Send the noisy feature vector $\tilde{x}_t^{i_t}$, noisy reward $\tilde{y}_t^{i_t}$, and parameter $\theta_{t+1}$ to the server.

15:     **For the server:**

16:     Update $T(i_t, t) = T(i_t, t-1) + 1$ and $T(j, t) = T(j, t-1)$ for $j \neq i_t$.

17:     Update $\theta_{t+1}^{i_t} = \theta_{t+1}$, $\tilde{V}_t^i = \tilde{V}_{t-1}^i + \tilde{x}_t^{i_t}(\tilde{x}_t^{i_t})^T$, $\tilde{u}_t^{i_t} = \tilde{u}_{t-1}^{i_t} + \langle\theta_t^{i_t}, \tilde{x}_t^{i_t}\rangle\tilde{x}_t^{i_t}$.

18:     Update $\theta_{t+1}^i = \theta_t^i$, $\tilde{V}_t^i = \tilde{V}_{t-1}^i$, $\tilde{u}_t^i = \tilde{u}_{t-1}^i$ for all $i \neq i_t$.

19:     Update $\hat{\mu}_{T(i_t,t)}^{i_t} = \frac{1}{T(i_t,t)}\sum_{\tau=1}^{T(i_t,t)} \tilde{y}_t^{i_t}$.

20:     **if** $\sum_{\tau=1}^{T(i_t,t)} \hat{\mu}_{\tau-1}^{i_t} - \hat{t}_\tau^{i_t} \geq L_{i_t} T(i_t, t)^{\alpha_{i_t}} + p\sqrt{T(i,t)(1+\sigma^2)\log\left(\frac{T^3 M \log T(i,t)(1+\sigma^2)}{\alpha}\right)}$ **then**

21:         $I_t = I_{t-1} - \{i_t\}$

22:     **else**

23:         $I_t = I_{t-1}$

24:     **end if**

25: **end for**

---

the same reasons discussed in relation to (Zheng et al., 2020). Interestingly, Algorithm 1 also exhibits better dimension dependence than even non-private algorithms (Abbasi-Yadkori et al., 2012; Abbasi-yadkori et al., 2011) in this specific setting. This is because the non-private models consider worst-case bounds universally, suggesting that these bounds might be further optimized under certain favorable conditions.

## 4 Online model selection for LDP LinUCB

From the previous section, we show that Algorithm 1 achieves $\tilde{O}(\sqrt{dT}/\epsilon)$ regret when $\lambda_{min} \approx O(1)$ and degrades to $\tilde{O}(\sqrt{d}T^{3/4}/\epsilon)$ when $\lambda_{min} \leq \frac{1}{T^{1/4}}$. However, to run Algorithm 1, we need to know the minimum eigenvalue $\lambda_{min}$ of $\mathbb{E}[x_t x_t^T]$ (the online learner does not need to know $\lambda_{min}$ but we still need $\lambda_{min}$ to set the confidence width). One might think that we can run a grid search through a range of values for $\lambda_{min}$ to find the one that works the best. However, running the algorithm multiple times using the same dataset can degrade our privacy guarantee (though in practice people still tune their algorithms). Moreover, in a truly online setting it may be that the minimum eigenvalue observed during this "training" period does not capture the later values. In this section, we present a new algorithm that bypasses this problem.

Algorithm 2 is built around a meta-learner that has access to $M$ distinct base learners (which are $M-1$ copies of Algorithm 1 and 1 copy of Algorithm 9 initialized with different guesses of $\lambda_{min} = \lambda_i$ for $i \in [M]$).

The reason we have to use two different types of base learners is that model selection algorithms usually require anytime regret guarantees, which Algorithm 1 satisfies only under the condition where perturbation of $x_t$ is unnecessary (i.e., $\Delta^2 = 0$). Thus, in scenarios where the actual $\lambda_{min}$ is small, necessitating the perturbation of $x_t$, we instead utilize the anytime variant of Algorithm 1, which is obtained by applying the classic "doubling trick" to Algorithm 1, as described in Algorithm 9. This variant serves as the first base learner of Algorithm 2. Then, by setting $\lambda_i = \frac{2^{i-1}}{T^{1/8}}$ for $i \in [M]$, we can guarantee that at least one of our guesses would be at most a constant factor away from the actual $\lambda_{min}$ if $\lambda_{min} \geq \frac{1}{T^{1/8}}$. In the scenario where $\lambda_{min} < \frac{1}{T^{1/8}}$, the actual value of $\lambda_{min}$ is not as important. This is due to our setting of the threshold $\bar{\lambda}$ at $\frac{1}{T^{1/8}}$. Under this condition, the base learner with the smallest $\lambda$ estimate automatically ensures a regret of $\tilde{O}(\sqrt{d}T^{3/4})$. The goal of the meta-learner is to "combine" the outputs of $M$ base learners into one output in such a way that the final regret is not much worse than if we had selected the best base learner in hindsight.

We employ the Bandit Combiner Algorithm in (Cutkosky et al., 2020) as our meta-learner. We note that there are more recent meta-learners with more refined guarantees and techniques (Pacchiano et al., 2023; Cutkosky et al., 2021), as well as techniques that work even in the fully adversarial setting (Agarwal et al., 2017; Pacchiano et al., 2020). However, (Cutkosky et al., 2020) is somewhat easier to use in our setting because it applies out-of-the-box to combine base learners whose individual regret bounds have different asymptotic rates.

The Bandit Combiner Algorithm (Algorithm 2) employs the use of the Upper Confidence Bound (UCB) strategy by treating each base learner as an arm in a multi-armed bandit setup. This approach involves using the average reward received by each base learner and their respective regret to establish the upper-confidence bound. Through this method, the algorithm sequentially identifies the most effective learner. Overall, Algorithm 2 guarantees $O(R_T^J)$ regret where $R_T^J$ is the regret of the best base learner. For a more detailed discussion of the algorithm, refer to (Cutkosky et al., 2020). We have the following guarantee for Algorithm 2:

**Theorem 4.1.** *(see Corollary 2 (Cutkosky et al., 2020)) Let* $\eta_1 = \frac{\epsilon T^{1/8}}{\sqrt{dT}(P \log^{3/2}(T)\epsilon T^{1/8}+1)}$ *and* $\eta_i = \frac{\epsilon}{P' \log^{3/2}(T)\sqrt{dT}}$, $L_1 = P \log^{3/2}(T)\sqrt{d}\left(\frac{\epsilon T^{1/8}+1}{\epsilon T^{1/8}}\right)$ *and* $L_i = P' \log^{3/2}(T)\frac{\sqrt{d}}{\epsilon \lambda_i}$ *for positive constants* $P$ *and* $P'$, $\alpha_1 = \frac{3}{4}$ *and* $\alpha_i = \frac{1}{2}$ *for* $i \in [2, M]$, *and set* $R_i$ *via:*

$$R_i = L_i T^{\alpha_i} + \frac{(1-\alpha_i)^{\frac{1-\alpha_i}{\alpha_i}}(1+\alpha_i)^{\frac{1}{\alpha_i}}}{\alpha_i^{\frac{1-\alpha_i}{\alpha_i}}} L_i^{\frac{1}{\alpha_i}} T \eta_i^{\frac{1-\alpha_i}{\alpha_i}} + 288 \log(T^3 N/\alpha) T \eta_i + \sum_{k \neq i} \frac{1}{\eta_k}$$

*for all* $i$. *Let* $j$ *be the index of the base learner with the smallest regret. If* $\lambda_{min} \geq \frac{1}{T^{1/8}}$, *then w.p at least* $1 - 3\alpha$, *the regret of Algorithm 2 under event* $\mathcal{E}$ *satisfies:*

$$Regret_T \leq \tilde{O}\left(\frac{\sqrt{dT}}{\epsilon \lambda_{min}^2}\right)$$

*If* $0 \leq \lambda_{min} < \frac{1}{T^{1/8}}$, *then w.p at least* $1 - 3\alpha$, *the regret of Algorithm 2 under event* $\mathcal{E}$ *satisfies:*

$$Regret_T \leq \tilde{O}\left(\sqrt{d}T^{5/6} + \frac{\sqrt{d}T^{17/24}}{\epsilon}\right)$$

Overall, Algorithm 2 guarantees $\tilde{O}(\sqrt{dT}/\epsilon)$ regret when $\lambda_{min}$ is $O(1)$ without requiring the knowledge of $\lambda_{min}$. The guarantee then degrades with a rate of $O(1/\lambda_{min}^2)$, yet it remains below $\tilde{O}\left(\sqrt{d}T^{5/6} + \frac{\sqrt{d}T^{17/24}}{\epsilon}\right)$. This result reveals the trade-offs involved when combining base learners that have different asymptotic regret guarantees. Since the general regret guarantee of Algorithm 2 is $\tilde{O}\left(L_j^{\frac{1}{a_j}} T \eta_j^{\frac{1-\alpha_j}{\alpha_j}} + \sum_{k \neq i} \frac{1}{\eta_k}\right)$, different asymptotic rate with different $a_j$ requires different settings of $\eta_j$ for the final rate to be optimal. Thus, to adapt to the $\tilde{O}(\sqrt{dT})$ rate, we suffer the worst case rate of $\tilde{O}(\sqrt{d}T^{5/6})$ instead of $\tilde{O}(\sqrt{d}T^{3/4})$ as in Algorithm

1. However, if we have a reason to believe that we are not in the favorable setting or if we want to preserve the worst-case rate of Algorithm 1, we can instead run Algorithm 1 with $\lambda_{min} = 0$ and $\bar{\lambda} = \frac{1}{T^{1/4}}$ to always guarantee $\tilde{O}(\sqrt{d}T^{3/4}/\epsilon)$ regret.

## 5    Experiments

In this section, we will compare the empirical performance of Algorithm 1 to that of previous works. Specifically, we compare our algorithm to Shuffle Private LinUCB (SDP) (Chowdhury & Zhou, 2022), Joint Differentially Private LinUCB (JDP) (Shariff & Sheffet, 2018), Locally Private LinUCB (LDP) (Zheng et al., 2020), and the Non-private LinUCB (LinUCB) (Abbasi-yadkori et al., 2011). For our algorithm (OnlineUCB), we implement Algorithm 1 with Maler as the online learner. In our experiment, we consider 100 arms with dimension $d = 5$. We run our algorithm over $T = 20000$ rounds and average the regrets over 50 trials. We generate the optimal parameter $\theta^\star$ and the feature vector $x_t$ by sampling a $(d-1)-$dimensional vector of norm $1/\sqrt{2}$ uniformly at random and append it with a $1/\sqrt{2}$ entry. We also use Bernoulli rewards to ensure boundedness. This is the same exact setting as in (Chowdhury & Zhou, 2022). As we can see from Figure 1,

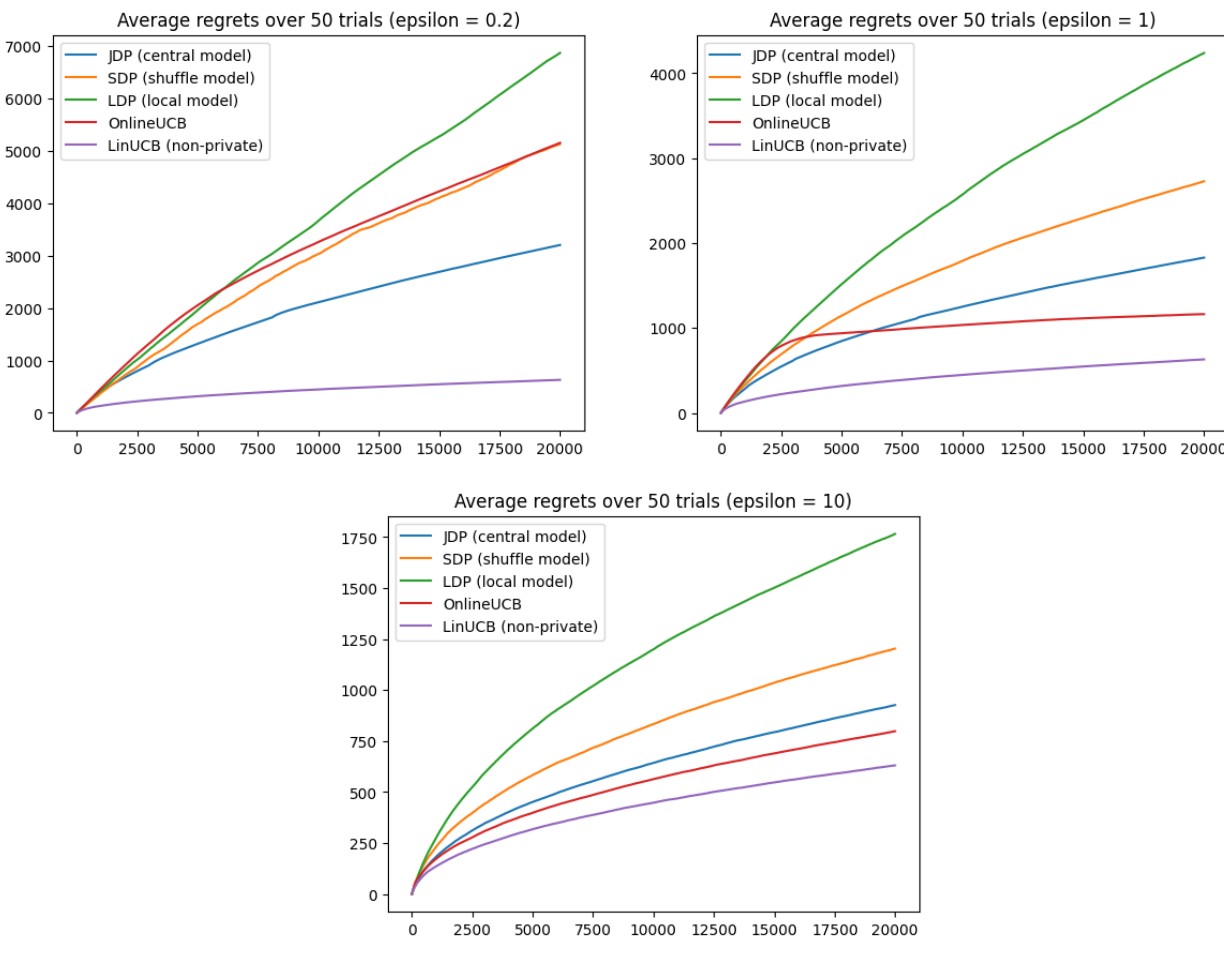

Figure 1: (a) $\epsilon = 0.2$ (b) $\epsilon = 1$ (c) $\epsilon = 10$

in the low-privacy domain ($\epsilon = 1$ and $\epsilon = 10$), our OnlineUCB algorithm significantly outperforms not only the previous best-known LDP algorithm but also SDP and JDP (which use the weaker shuffle and central models of differential privacy respectively) while having stronger privacy guarantee. This result is consistent with the theory since when there exists a $\lambda_{min} = O(1)$ such that $\mathbb{E}_t\left[x_t x_t^T\right] \succeq \lambda_{min} I$ (which is our experiment settings), OnlineUCB has a better regret guarantee than all of the previous private stochastic linear bandits

algorithms. In the high privacy domain ($\epsilon = 0.2$), OnlineUCB does not do as well but still outperforms LDP LinUCB and has comparable performance to Shuffle LinUCB. We also note that even though we use Maler to be consistent with the theory, one could also use Online Gradient Descent (which runs a lot faster than Maler) as the online learner to get the same empirical performance.

## 6 Conclusions

In this paper, we present a new algorithm for differentially private linear (contextual) stochastic bandit in the local settings that uses an online learner to construct the confidence set. By carefully choosing the online learner as well as the loss function sent to the online learner, our algorithm guarantees the regret $\tilde{O}\left(\min\left\{\frac{\sqrt{d}T^{3/4}}{\epsilon}, \frac{\sqrt{dT}}{\epsilon\lambda_{min}}\right\}\right)$ with high probability where $\lambda_{min}$ is the lower bound on $\mathbb{E}_t[x_t x_t^T]$. Thus, when $\lambda_{min} \approx O(1)$, our algorithm is the first algorithm that guarantees $\tilde{O}(\sqrt{T})$ for the LDP setting. Further, by running the online learner on a bounded domain, we are able to improve the regret dependence on the dimension $d$ of the feature vector $x_t$ from $O(d^{3/4})$ to $O(\sqrt{d})$. There are several limitations that one could further explore to improve the results of this paper. The most natural question is if it is possible to guarantee $\tilde{O}\left(\sqrt{T}\right)$ for the local model without any further assumption. In our current result, we still rely heavily on the fact that $\mathbb{E}_t[x_t x_t^T] \succeq \lambda_{min} I_d$ to get $\tilde{O}\left(\sqrt{T}\right)$ regret. If that is not possible, what are other settings that allow us to achieve $\tilde{O}\left(\sqrt{T}\right)$ regret? One could hope that by further exploiting the properties of specific online learners from the rich literature of online optimization, we can find other favorable domains as well.

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

# A    Backgrounds on LinUCB

## A.1    LinUCB

---
**Algorithm 3** LinUCB
---
1: **for** $t = 1 \ldots T$ **do**
2:      $(x_t, \tilde{\theta}_t) = \arg\max_{(x,\theta) \in D_t \times C_{t-1}} \langle x, \theta \rangle$
3:      Play $x_t$ and observe reward $y_t$
4:      Update $C_t$
5: **end for**

---

At its core, our algorithm is simply the private version of LinUCB (Algorithm 3). Since its introduction in 2011 (Chu et al., 2011; Abbasi-yadkori et al., 2011), LinUCB has been the method of choice for linear bandit problems due to its good theoretical guarantee and its effectiveness in practice. LinUCB is based on the idea of *optimisim-in-the-face-of-uncertainty* (OFU). The OFU principle elegantly solves the exploration-exploitation dilemma of bandit problems. The basic idea of this principle is to maintain a confidence set for the vector of coefficients of the linear function. At every round $t$, LinUCB constructs a confidence set $C_t$ that contains the optimal parameter $\theta^\star$ with high probability. It then computes an upper confidence bound on the reward of each action in the decision set $\mathcal{D}_t$, then chooses the action with the highest upper confidence bound: $x_t \leftarrow \arg\max_{x \in \mathcal{D}_t}(\max_{\theta \in C_t} \langle \theta, x \rangle)$. Let $V_t = \sum_{i=1}^{t} x_i x_i^T + \lambda I$, $u_t = \sum_{i=1}^{t} x_i y_i$, $X_{<t} \in \mathbb{R}^{(t-1) \times d}$ where $X_{<t,s} = x_s^T$ for $s < t$, and $Y_{<t} \in \mathbb{R}^{t-1}$ be the vector of rewards up to round t. Since in our setting, the rewards are linear functions of the actions with subgaussian noises, it is natural to center the confidence set $C_t$ on the linear regression estimate:

$$\hat{\theta}_t = \arg\min_{\theta \in \mathbb{R}^d} \|\theta X_{<t}\theta - Y_{<t}\|^2 + \|\theta\|^2$$
$$= V_t^{-1} u_t$$

Whenever we get a new reward $y_t$, it gives us information about the projection of $\theta^\star$ onto the action space, thus $\hat{\theta}$ is closer to $\theta^\star$ along the directions where many actions have been taken. This motivates the use of ellipsoid confidence set that is smaller in such directions. LinUCB defines the confidence set as follows:

$$C_t = \left\{ \theta \in \mathbb{R}^d : \|\theta - \hat{\theta}_t\|_{V_t} \le \beta_t \right\} \tag{7}$$

where $\beta_t$ is the width of the confidence set. (Abbasi-yadkori et al., 2011) then shows that the optimal $\theta^\star$ is inside $C_t$ with high probability with appropriate $\beta_t$ by bounding the difference between the actual rewards and the predicted rewards using Algorithm 3.

**Theorem A.1.** *(Theorem 2 in (Abbasi-yadkori et al., 2011)) Define $y_t = \langle x_t, \theta^\star \rangle + \eta_t$ where $\eta_t$ is $R-$subgaussian noise and assume that $\|\theta^\star\| \le S$. Then for any $\delta > 0$, with probability at least $1 - \delta$, for all $t \ge 1$, $\theta^\star$ lies in the set:*

$$C_t = \left\{ \theta \in \mathbb{R}^d : \|\theta - \hat{\theta}_t\|_{V_t} \le \beta_t \right\}$$

*where $\beta_t = R\sqrt{2\log\left(\frac{det(V_t)^{1/2}det(\lambda I)^{-1/2}}{\delta}\right)} + \lambda^{1/2}S$.*

Finally, (Abbasi-yadkori et al., 2011) proves the following regret bound:

$$Regret_T \le \tilde{O}(\beta_t \sqrt{dn})$$

Now, we can plug in the value of $\beta_t$ in Theorem A.1 and use the fact that $\log(det(V_t)) \le O(d)$ (Lemma 10 in (Abbasi-yadkori et al., 2011)) to get the $\tilde{O}(d\sqrt{T})$ regret of non-private LinUCB.

## A.2 LinUCB with online-to-confidence conversion

---

**Algorithm 4** Online LinUCB

---
1: **for** $t = 1 \ldots T$ **do**
2:     Construct confidence set $C_{t-1}$
3:     $(x_t, \tilde{\theta}_t) = \arg \max_{(x,\theta) \in D_t \times C_{t-1}} \langle x, \theta \rangle$
4:     Get prediction $\hat{y}_t$ from online learner $M$
5:     Play $x_t$ and observe reward $y_t$
6:     Update $C_t$
7: **end for**

---

Building on the original LinUCB algorithm, (Abbasi-Yadkori et al., 2012) introduces a new variant of LinUCB where we construct the confidence set using the predictions from an online optimization algorithm. The main procedure is described in Algorithm 4. In every round t, Algorithm 4 sends the action $x_t$, the reward $y_t$ to an online learner M and gets back some predictions $\hat{y}_t$. The online learner then suffers a loss $l_t(\hat{y}_t)$ (e.g: the squared loss $l_t(\hat{y}_t) = (\hat{y}_t - y_t)^2$). Then, if we define $M_n$ as the upper bound on the regret at time $n$ of the online learner M ($M_n \geq \sum_{t=1}^{N} l_t(\theta_t) - l_t(\theta^\star)$), (Abbasi-Yadkori et al., 2012) shows that we can construct a confidence set that depends on $M_n$ where $\theta^\star$ is inside the confidence set with high probability.

**Theorem A.2.** *(Corollary 2 in (Abbasi-Yadkori et al., 2012)) Assume $\|\theta^\star\| \leq S$. Then for any $\delta \in (0, 1/4]$, with probability at least $1 - \delta$, the true parameter $\theta^\star$ lies in the intersections of the sets*

$$C_n = \left\{ \|\theta\|^2 + \sum_{t=1}^{n} (\hat{y}_t - \langle \theta, x_t \rangle)^2 \leq \beta_n^2 \right\}$$

*where $\beta_n^2 = S^2 + 1 + 2M_n + 32R^2 \log \left( \frac{R\sqrt{8} + \sqrt{1+M_n}}{\delta} \right)$*

For a more detailed analysis, please refer to (Abbasi-Yadkori et al., 2012) or our proofs of Lemma 3.3. Since Algorithm 1 is based on LinUCB with online-to-confidence conversion, the analysis of Algorithm 1 and Algorithm 4 are very similar. We have the final regret bound for Algorithm 4 as follows:

$$Regret_T \leq \tilde{O} \left( \max_{1 \leq t \leq T} \beta_t \sqrt{dT} \right)$$

At first glance, this regret bound is exactly the same as the regret bound of the original LinUCB discussed in Section A.1. However, notice that the confidence width of the original LinUCB in Section A.1 is always $O(\sqrt{d})$ due to its dependence on $\log(det(V_t))$ while Online LinUCB depends on the regret of the online learner $M$ instead. Though the worst case bound of both algorithms are $\tilde{O}(d\sqrt{T})$, Online LinUCB allows us the flexibility of choosing our own online optimization algorithms to adapt to different problem settings. Specifically, (Abbasi-Yadkori et al., 2012) shows that when $\theta^\star$ is a sparse vector, Online LinUCB guarantees $\tilde{O}(\sqrt{dT\|\theta^\star\|_0})$ which is better than the worst-case bound $\tilde{O}(d\sqrt{T})$. This is the reason why we chose Online LinUCB as the base algorithm for our private method since it allows us to potentially improve the final regret bound for favorable scenarios.

## B  Good Event and High-confidence Argument

Let $z_t = r_t + \eta_{y,t}$, $\theta_t$ is the prediction of the online learner in Algorithm 1, $l_t(\theta_t)$ is the loss we sent to the online learner, $L(\theta_t) = \mathbb{E}\left[l_t(\theta_t)\right]$, and $G$ is the clipping constant. We define the following good events:

$$\mathcal{E}_1 = \left\{\forall t \in [T] : \|\eta_{x,t}\| \le \sigma\sqrt{2\log(2T/\alpha)}\right\}$$

$$\mathcal{E}_2 = \left\{\forall t \in [T] : \|\eta_{y,t}\| \le \sigma\sqrt{2\log(2T/\alpha)}\right\}$$

$$\mathcal{E}_3 = \left\{\forall t \in [T] : \|\zeta_t\| \le \Delta\sqrt{2\log(2T/\alpha)}\right\}$$

$$\mathcal{E}_4 = \left\{\forall t \in [T] : \left\|\eta_{x,t}\eta_{x,t}^T - \Sigma\right\|_{op} \le 2DC\sigma^2(d + \log(2T/\alpha))\right\}$$

$$\mathcal{E}_5 = \left\{\forall N \in [T] : |\sum_{t=1}^N (z_t - \langle \eta_{x,t} + \zeta_t, \theta_t\rangle)\langle x_t, \theta_t - \theta^\star\rangle|\right.$$
$$\left. \le \left(R + \frac{2(D+1)\sqrt{2\log(1.25/\delta)}}{\epsilon} + D\Delta\right)\sqrt{2\left(1 + \sum_{t=1}^N \langle x_t, \theta_t - \theta^\star\rangle^2\right)}\sqrt{\log\left(\frac{\sqrt{1 + \sum_{t=1}^N \langle x_t, \theta_t - \theta^\star\rangle^2}}{\alpha}\right)}\right\}$$

$$\mathcal{E}_6 = \left\{\forall N \in [T] : |\sum_{t=1}^N z_t\langle \eta_{x,t} + \zeta_t, \theta_t - \theta^\star\rangle| \le \left(R + \frac{2\sqrt{2\log(1.25/\delta)}}{\epsilon}\right)\sqrt{2\left(1 + \sum_{t=1}^N \langle \eta_{x,t} + \zeta_t, \theta_t - \theta^\star\rangle^2\right)}\right\}$$

$$\mathcal{E}_7 = \left\{\forall N \in [T] : \left|\sum_{t=1}^N (\theta^\star - \theta_t)^T \zeta_t\eta_{x,t}^T(\theta^\star - \theta_t)\right|\right.$$
$$\left. \le \Delta\|\theta_t - \theta^\star\|\sqrt{2\left(1 + \sum_{t=1}^N \langle \eta_{x,t}, \theta^\star - \theta_t\rangle^2\right)\log\left(\frac{\sqrt{1 + \sum_{t=1}^N \langle \eta_{x,t}, \theta^\star - \theta_t\rangle^2}}{\alpha}\right)}\right\}$$

$$\mathcal{E}_8 = \left\{\forall N \in [T] : \left|\sum_{t=1}^N \theta_t^T\zeta_t r_t^T(\theta^\star - \theta_t)\right| + \left|\sum_{t=1}^N (\theta^\star - \theta_t)^T\zeta_t r_t^T\theta_t\right|\right.$$
$$\le \Delta\|\theta_t\|\sqrt{2\left(1 + \sum_{t=1}^N \langle \eta_{x,t}, \theta^\star - \theta_t\rangle^2\right)\log\left(\frac{\sqrt{1 + \sum_{t=1}^N \langle \eta_{x,t}, \theta^\star - \theta_t\rangle^2}}{\alpha}\right)}$$
$$\left. +\sigma\|\theta_t\|\sqrt{2\left(1 + \sum_{t=1}^N \langle \zeta_t, \theta^\star - \theta_t\rangle^2\right)\log\left(\frac{\sqrt{1 + \sum_{t=1}^N \langle \zeta_t, \theta^\star - \theta_t\rangle^2}}{\alpha}\right)}\right\}$$

$$\mathcal{E}_9 = \left\{\forall N \in [T] : \left\|\sum_{t=1}^N (\theta^\star - \theta_t)^T(\eta_{x,t}\eta_{x,t}^T - \Sigma)(\theta_t + \theta^\star)\right\|\right.$$
$$\le 20\sigma^2(D+1)\log(2T/\alpha)\sqrt{\sum_{t=1}^N \|\theta^\star - \theta_t\|^2\log\left(\frac{16}{\alpha}\left[\log\left(e^2\left[\sqrt{\sum_{t=1}^N \frac{16(D+1)^2\sigma^4\log^2\left(\frac{2T}{\alpha}\right)\|\theta^\star - \theta_t\|^2}{\nu^2}}\right]\right)\right]_1\right)^2}$$
$$\left. +23\max(\nu, \max_t 4(D+1)\sigma^2\log\left(\frac{2T}{\alpha}\right)\|\theta^\star - \theta_t\|)\log\left(\frac{224}{\alpha}\left[\log\left(\frac{2e^2\max(\nu, \max_t 4(D+1)\sigma^2\log\left(\frac{2T}{\alpha}\right)\|\theta^\star - \theta_t\|)}{\nu}\right)\right]^2\right)\right\}$$

for some arbitrary constant $\nu \geq 0$ and $[x]_1 := \max(1, x)$.

$$\mathcal{E}_{10} = \left\{ \sum_{t=1}^{T} \langle \nabla L(\theta_t) - \nabla l_t(\theta_t), \theta_t - \theta^\star \rangle \leq 2\sqrt{2G^2 \log(1/\alpha) \sum_{t=1}^{T} \|\theta_t - \theta^\star\|^2} \right\}$$

$$\mathcal{E} = \bigcup_{i=1}^{10} \mathcal{E}_i$$

**Lemma B.1.** *Event $\mathcal{E}$ happens with probability at least $1 - 11\alpha$.*

*Proof.* Since $\eta_{x,t} \sim N(0, \sigma_X^2)$, from Theorem H.10, w.p at least $1 - \alpha/T$:

$$\|\eta_{x,t}\| \leq \sigma\sqrt{2\log(2T/\alpha)}$$

Using the union bound over all $t \in [T]$ to get that event $\mathcal{E}_1$ happens w.p at least $1 - \alpha$. Now repeat the same argument with $\eta_{y,t} \sim N(0, \sigma^2)$ and $\zeta_t \sim N(0, \Delta^2)$ to get the high probability bound for $\mathcal{E}_2$ and $\mathcal{E}_3$.

For $\mathcal{E}_4$, we can use Theorem H.7 to get w.p at least $1 - \alpha$ for every $t \in [T]$:

$$\left\| \eta_{x,t} \eta_{x,t}^T - \Sigma \right\|_{op} \leq 2DC \left( d + \log(2T/\alpha) \right) \|\Sigma\|_{op}$$
$$= 2DC\sigma^2 \left( d + \log(2T/\alpha) \right)$$

for an universal constant $C > 0$.

$\mathcal{E}_5$: It's easy to see that $\left\{ \sum_{t=1}^{N} (z_t - \langle \eta_{x,t} + \zeta_t, \theta_t \rangle) \langle x_t, \theta_t - \theta^\star \rangle \right\}_{N=0}^{\infty}$ is a martingale sequence and $z_t - \langle \eta_{x,t} + \zeta_t, \theta_t \rangle$ is $\sqrt{R^2 + \sigma^2 + (\sigma^2 + \Delta^2)\|\theta_t\|^2}$ − subgaussian. Then using corollary H.12,w.p at least $1 - \alpha$ we have:

$$|\sum_{t=1}^{N} (z_t - \langle \eta_{x,t} + \zeta_t, \theta_t \rangle) \langle x_t, \theta_t - \theta^\star \rangle| \leq \left( R + \frac{2(D+1)\sqrt{2\log(1.25/\delta)}}{\epsilon} + D\Delta \right) \sqrt{2\left(1 + \sum_{t=1}^{N} \langle x_t, \theta_t - \theta^\star \rangle^2\right)}$$
$$\times \sqrt{\log\left(\frac{\sqrt{1 + \sum_{t=1}^{N} \langle x_t, \theta_t - \theta^\star \rangle^2}}{\alpha}\right)}$$

$\mathcal{E}_6$: Similarly, we can apply corollary H.12 to get $\mathcal{E}_6$. W.p at least $1 - \alpha$:

$$|\sum_{t=1}^{N} z_t \langle \eta_{x,t} + \zeta_t, \theta_t - \theta^\star \rangle| \leq \left( R + \frac{2\sqrt{2\log(1.25/\delta)}}{\epsilon} \right) \sqrt{2\left(1 + \sum_{t=1}^{N} \langle \eta_{x,t} + \zeta_t, \theta_t - \theta^\star \rangle^2\right)}$$
$$\times \sqrt{\log\left(\frac{\sqrt{1 + \sum_{t=1}^{N} \langle \eta_{x,t} + \zeta_t, \theta_t - \theta^\star \rangle^2}}{\alpha}\right)}$$

$\mathcal{E}_7$: $\left\{ \sum_{t=1}^{N} (\theta^\star - \theta_t)^T \zeta_t \eta_{x,t}^T (\theta^\star - \theta_t) \right\}_{N=0}^{\infty}$ is a martingale sequence and $(\theta^\star - \theta_t)^T \zeta_t$ is $\Delta\|\theta_t - \theta^\star\|$−subgaussian. Using corollary H.12, w.p $1 - \alpha$ we get:

$$\left| \sum_{t=1}^{N} (\theta^\star - \theta_t)^T \zeta_t \eta_{x,t}^T (\theta^\star - \theta_t) \right| \leq \Delta\|\theta_t - \theta^\star\| \sqrt{2\left(1 + \sum_{t=1}^{N} \langle \eta_{x,t}, \theta^\star - \theta_t \rangle^2\right) \log\left(\frac{\sqrt{1 + \sum_{t=1}^{N} \langle \eta_{x,t}, \theta^\star - \theta_t \rangle^2}}{\alpha}\right)}$$

$\mathcal{E}_8$: Both $\left\{\sum_{t=1}^{N}\theta_t^T\zeta_t\eta_{x,t}^T(\theta^\star-\theta_t)\right\}_{N=0}^{\infty}$ and $\left\{\sum_{t=1}^{N}(\theta^\star-\theta_t)^T\zeta_t\eta_{x,t}^T\theta_t\right\}_{N=0}^{\infty}$ are martingale sequences. Thus we can again apply corollary H.12 w.p at least $1-2\alpha$:

$$\left|\sum_{t=1}^{N}\theta_t^T\zeta_t\eta_{x,t}^T(\theta^\star-\theta_t)\right| + \left|\sum_{t=1}^{N}(\theta^\star-\theta_t)^T\zeta_t\eta_{x,t}^T\theta_t\right|$$

$$\leq \Delta\|\theta_t\|\sqrt{2\left(1+\sum_{t=1}^{N}\langle\eta_{x,t},\theta^\star-\theta_t\rangle^2\right)\log\left(\frac{\sqrt{1+\sum_{t=1}^{N}\langle\eta_{x,t},\theta^\star-\theta_t\rangle^2}}{\alpha}\right)}$$

$$+ \sigma\|\theta_t\|\sqrt{2\left(1+\sum_{t=1}^{N}\langle\zeta_t,\theta^\star-\theta_t\rangle^2\right)\log\left(\frac{\sqrt{1+\sum_{t=1}^{N}\langle\zeta_t,\theta^\star-\theta_t\rangle^2}}{\alpha}\right)}$$

$\mathcal{E}_9$: We have w.p at least $1-\alpha$, $\|(\theta_t-\theta^\star)^T\eta_{x,t}\eta_{x,t}^T(\theta_t+\theta^\star)\| \leq 2\sigma^2\|\theta_t-\theta^\star\|\|\theta_t+\theta^\star\|\log(2T/\alpha)$ and $\|(\theta_t-\theta^\star)^T\Sigma(\theta_t+\theta^\star)\| \leq 2\sigma^2\|\theta_t-\theta^\star\|\|\theta_t+\theta^\star\|\log(2T/\alpha)$. Thus:

$$\|(\theta^\star-\theta_t)^T(\eta_{x,t}\eta_{x,t}^T-\Sigma)(\theta_t+\theta^\star)\|^2 \leq 2\|(\theta^\star-\theta_t)^T\eta_{x,t}\eta_{x,t}^T(\theta_t+\theta^\star)\|^2 + 2\|(\theta^\star-\theta_t)^T\Sigma(\theta_t+\theta^\star)\|^2$$

$$\leq 16\sigma^4\log^2(2T/\alpha)\|\theta^\star-\theta_t\|^2\|\theta^\star+\theta_t\|^2$$

$$\leq 16(D+1)^2\sigma^4\log^2(2T/\alpha)\|\theta^\star-\theta_t\|^2$$

Let $X_t = (\theta_t-\theta^\star)^T(\eta_{x,t}\eta_{x,t}^T-\Sigma)(\theta_t+\theta^\star) \Rightarrow \mathbb{E}[X_t|\eta_{x,1},\theta_1,...,\eta_{x,t-1},x_{t-1}] = 0$. Thus, $X_t$ is a martingale difference sequence and $X_t$ is $\left(4(D+1)\sigma^2\log(2T/\alpha)\|\theta^\star-\theta_t\|, 8(D+1)\sigma^2\log(2T/\alpha)\|\theta^\star-\theta_t\|\right)$ sub-exponential by Proposition H.1. Now applying Theorem H.2, w.p at least $1-2\alpha$ we get

$$\left\|\sum_{t=1}^{N}(\theta^\star-\theta_t)^T(\eta_{x,t}\eta_{x,t}^T-\Sigma)(\theta_t+\theta^\star)\right\|$$

$$\leq 5\sqrt{\sum_{t=1}^{N}16(D+1)^2\sigma^4\log^2\left(\frac{2T}{\alpha}\right)\|\theta^\star-\theta_t\|^2\log\left(\frac{16}{\alpha}\left[\log\left(\left[\sqrt{\sum_{t=1}^{N}\frac{16(D+1)^2\sigma^4\log^2\left(\frac{2T}{\alpha}\right)\|\theta^\star-\theta_t\|^2}{\nu^2}}\right]_1\right)+2\right]^2\right)}$$

$$+ 23\max(\nu,\max_t 4(D+1)\sigma^2\log\left(\frac{2T}{\alpha}\right)\|\theta^\star-\theta_t\|)\log\left(\frac{224}{\alpha}\left[\log\left(\frac{2\max(\nu,\max_t 4(D+1)\sigma^2\log\left(\frac{2T}{\alpha}\right)\|\theta^\star-\theta_t\|)}{\nu}\right)+2\right]^2\right)$$

$$= 20\sigma^2(D+1)\log(2T/\alpha)\sqrt{\sum_{t=1}^{N}\|\theta^\star-\theta_t\|^2\log\left(\frac{16}{\alpha}\left[\log\left(e^2\left[\sqrt{\sum_{t=1}^{N}\frac{16(D+1)^2\sigma^4\log^2\left(\frac{2T}{\alpha}\right)\|\theta^\star-\theta_t\|^2}{\nu^2}}\right]_1\right)\right]^2\right)}$$

$$+ 23\max(\nu,\max_t 4(D+1)\sigma^2\log\left(\frac{2T}{\alpha}\right)\|\theta^\star-\theta_t\|)\log\left(\frac{224}{\alpha}\left[\log\left(\frac{2e^2\max(\nu,\max_t 4(D+1)\sigma^2\log\left(\frac{2T}{\alpha}\right)\|\theta^\star-\theta_t\|)}{\nu}\right)\right]^2\right)$$

for some arbitrary constant $\nu \geq 0$ and $[x]_1 := \max(1,x)$.

$\mathcal{E}_{10}$: Since $\{\langle\nabla L(\theta_t)-\nabla l_t(\theta_t),\theta_t-\theta^\star\rangle\}_{t=1}^{T}$ is a martingale difference sequence and $-2G\|\theta_t-\theta^\star\| \leq \langle\nabla L(\theta_t)-\nabla l_t(\theta_t),\theta_t-\theta^\star\rangle \leq 2G\|\theta_t-\theta^\star\|$, applying Azuma-Hoeffding inequality (Theorem H.16), we have w.p at least $1-\alpha$:

$$\sum_{t=1}^{T}\langle\nabla L(\theta_t)-\nabla l_t(\theta_t),\theta_t-\theta^\star\rangle \leq 2\sqrt{2G^2\log(1/\alpha)\sum_{t=1}^{T}\|\theta_t-\theta^\star\|^2}$$

Now we can use the union bound over all events to conclude the result. □

## C   Maler Optimizer (Wang et al., 2020a)

We denote our loss as $l_t(\theta_t)$ and $L(\theta_t) = \mathbb{E}[l_t(\theta_t)]$. We have the following lemma on the Lipschitz constant of $l_t(\theta_t)$ under the good event $\mathcal{E}$.

**Corollary C.1.** *Under the event $\mathcal{E}$ we have:*

$$\max_t \|\nabla l_t(\theta_t)\| \leq G$$

*where $G = 2\sqrt{2\log(2T/\alpha)}(2D\sigma + 2\sigma + D\Delta + \Delta) + 2\log(2T/\alpha)(2D\Delta^2 + 4D\Delta\sigma + 2\Delta\sigma + 2D\sigma^2 + 2\sigma^2 + DC\sigma^2) + 2D + 2$.*

*Proof.* We have:

$$\begin{aligned}
\nabla l_t(\theta_t) &= 2\tilde{x}_t(\langle \tilde{x}_t, \theta_t \rangle - \tilde{y}_t) - 2\Sigma\theta_t \\
&= 2(x_t + \zeta_t + \eta_{x,t}))(\langle x_t + \zeta_t + \eta_{x,t}, \theta_t \rangle - \tilde{y}_t) - 2\Sigma\theta_t
\end{aligned}$$

Then,

$$\begin{aligned}
\|\nabla l_t(\theta_t)\| \leq 2 \big( &\|x_t\|^2\|\theta_t\| + 2\|x_t\|\|\eta_{x,t}\|\|\theta_t\| + \|x_t\|\|y_t\| + \|x_t\|\|\eta_{y,t}\| + 2\|x_t\|\|\zeta_t\|\|\theta_t\| + \|\zeta_t\|^2\|\theta_t\| + 2\|\zeta_t\|\|\eta_t\|\|\theta_t\| \\
&+ \|\zeta_t\|\|y_t\| + \|\zeta_t\|\|\eta_{y,t}\| + \|\eta_{x,t}\|^2\|\theta_t\| + \|\eta_{x,t}\|\|\eta_{y,t}\| + \|\eta_{x,t}\|\|y_t\| + \|(\eta_{x,t}\eta_{x,t}^T - \Sigma)\theta_t\|\big)
\end{aligned}$$

Thus, under the event $\mathcal{E}$:

$$\max_t \|\nabla l_t(\theta_t)\| \leq G$$

$\square$

Using the result of the above Corollary we have $\|\nabla l_t(\theta_t)\| \leq G$ and the domain $\mathcal{D}$ of $\theta_t$ is bounded by $2D$ for all $t \in [T]$. To run Maler, we need to define the surrogate loss as follows:

$$\begin{aligned}
s_t^r(\theta) &= -\eta(\theta_t - \theta)^T g_t + \eta^2 G^2\|\theta_t - \theta\|^2 \\
l_t^r(\theta) &= -\eta(\theta_t - \theta)^T g_t + \eta^2(\theta - \theta_t)^T g_t g_t^T(\theta - \theta_t) \\
c_t(\theta) &= -\eta^c(\theta_t - \theta)^T g_t + (2\eta^c GD)^2
\end{aligned}$$

We present the Maler algorithm in Algorithm 5:

**Theorem C.2.** *Assuming $L(\theta_t) = \mathbb{E}[l_t(\theta_t)]$ is $\mu-$strongly convex and $\max_{t,t'} \|\theta_t - \theta'_t\| \leq 2D$. Maler (Algorithm 5) under the event $\mathcal{E}$ guarantees:*

$$\begin{aligned}
\sum_{t=1}^T L(\theta_t) - L(\theta^\star) &\leq \left(40GD + \frac{18G^2}{\mu}\right)\left(2ln\left(\sqrt{3}\left(\frac{1}{2}\log_2 T + 3\right)\right) + 1 + \log T\right) + \frac{32G^2 \log(1/\alpha)}{\mu} \\
&\leq O\left(\left(\frac{G^2}{\mu} + GD\right)\log T + \frac{G^2}{\mu}\log(1/\alpha)\right)
\end{aligned}$$

*Furthermore, we have:*

$$\begin{aligned}
\sum_{t=1}^T \|\theta_t - \theta^\star\|^2 &\leq \left(\frac{80GD}{\mu} + \frac{36G^2}{\mu^2}\right)\left(2ln\left(\sqrt{3}\left(\frac{1}{2}\log_2 T + 3\right)\right) + 1 + \log T\right) + \frac{64G^2 \log(1/\alpha)}{\mu^2} \\
&\leq O\left(\left(\frac{G^2}{\mu^2} + \frac{GD}{\mu}\right)\log T + \frac{G^2}{\mu^2}\log(1/\alpha)\right)
\end{aligned}$$

---

**Algorithm 5** Maler

---

1: **Input:** Learning rate $\eta^c, \eta_1, \eta_2, \ldots$, prior weights $\pi_1^c, \pi_1^{\eta_1,s}, \pi_1^{\eta_2,s}, \ldots$ and $\pi_1^{\eta_1,l}, \pi_1^{\eta_2,l}, \ldots$
2: **for** $t = 1, \ldots, T$ **do**
3:     Get predictions $\theta_t^c$ from Algorithm 6, $\theta_t^{\eta,\ell}$ from Algorithm 7, and $\theta_t^{\eta,s}$ from Algorithm 8 for all $\eta$.
4:     Play $\theta_t = \dfrac{\pi_t^c \eta^c \theta_t^c + \sum_\eta \pi_t^{\eta,s} \eta \theta_t^{\eta,s} + \pi_t^{\eta,\ell} \eta \theta_t^{\eta,\ell}}{\pi_t^c \eta^c + \sum_\eta \pi_t^{\eta,s} \eta + \pi_t^{\eta,\ell} \eta}$
5:     Observe gradient $g_t$ and send it to the experts
6:     Update weights:

$$\pi_{t+1}^c = \frac{\pi_t^c e^{-c_t(\theta_t^c)}}{\phi_t}$$

$$\pi_{t+1}^{\eta,s} = \frac{\pi_t^{\eta,s} e^{-s_t^\eta(\theta_t^{\eta,s})}}{\phi_t}$$

$$\pi_{t+1}^{\eta,\ell} = \frac{\pi_t^{\eta,\ell} e^{-l_t^\eta(\theta_t^{\eta,\ell})}}{\phi_t}$$

    where:

$$\phi_t = \pi_t^c e^{-c_t(\theta_t^c)} + \sum_\eta \pi_t^{\eta,s} e^{-s_t^\eta(\theta_t^{\eta,s})} + \pi_t^{\eta,\ell} e^{-l_t^\eta(\theta_t^{\eta,\ell})}$$

7: **end for**

---

**Algorithm 6** Convex Expert Algorithm

---

1: Initialize $\theta_1^c = 0$
2: **for** $t = 1, \ldots, T$ **do**
3:     Send $\theta_t^c$ to Algorithm 5
4:     Receive gradient $g_t$ from Algorithm 5
5:     Update $\theta_{t+1}^c = \Pi_{\mathcal{D}}^{I_d} \left( \theta_t^c - \frac{2D}{\eta^c G \sqrt{t}} \nabla c_t(\theta_t^c) \right)$
6: **end for**

---

*Proof.* Our analysis follows the analysis of Maler which in turn follows the analysis of Metagrad. We have:

$$\sum_{t=1}^T L(\theta_t) - L(\theta^\star) \leq \sum_{t=1}^T \langle \nabla L(\theta_t), \theta_t - \theta^\star \rangle - \frac{\mu}{2} \|\theta_t - \theta^\star\|^2$$

$$= \sum_{t=1}^T \langle \nabla l_t(\theta_t), \theta_t - \theta^\star \rangle - \frac{\mu}{2} \|\theta_t - \theta^\star\|^2 + \langle \nabla L(\theta_t) - \nabla l_t(\theta_t), \theta_t - \theta^\star \rangle$$

$$= \sum_{t=1}^T \langle g_t, \theta_t - \theta^\star \rangle - \frac{\mu}{2} \|\theta_t - \theta^\star\|^2 + \langle \nabla L(\theta_t) - \nabla l_t(\theta_t), \theta_t - \theta^\star \rangle + \langle \nabla l_t(\theta_t) - g_t, \theta_t - \theta^\star \rangle$$

Under event $\mathcal{E}$, $g_t = \nabla l_t(\theta_t)$:

$$= \sum_{t=1}^T \langle g_t, \theta_t - \theta^\star \rangle - \frac{\mu}{2} \|\theta_t - \theta^\star\|^2 + \langle \nabla L(\theta_t) - \nabla l_t(\theta_t), \theta_t - \theta^\star \rangle$$

$$= \frac{\sum_{t=1}^T -s_t^r(\theta^\star) + \eta^2 G^2 \|\theta^\star - \theta_t\|^2}{\eta} - \frac{\mu}{2} \|\theta_t - \theta^\star\|^2 + \langle \nabla L(\theta_t) - \nabla l_t(\theta_t), \theta_t - \theta^\star \rangle$$

Using Lemma H.6, we get

$$\leq \left( 10GD + \frac{9G^2}{2\mu} \right) \left( 2ln \left( \sqrt{3} \left( \frac{1}{2} \log_2 T + 3 \right) \right) + 1 + \log T \right) + \langle \nabla L(\theta_t) - \nabla l_t(\theta_t), \theta_t - \theta^\star \rangle$$

---

**Algorithm 7** Exp-concave Expert Algorithm

---

1: **Input:** Learning rate $\eta$
2: $\theta_1^{\eta,l} = 0$, $\alpha = \frac{1}{2}\min\{\frac{2}{G^\ell D}, 1\}$ where $G^\ell = \frac{7}{50D}$, $\Sigma_1 = \frac{1}{4\alpha^2 D^2}I_d$
3: **for** $t = 1, \ldots, T$ **do**
4:     Send $\theta^{\eta,\ell}$ to Algorithm 5
5:     Receive gradient $g_t$ from Algorithm 5
6:     Update

$$\Sigma_{t+1} = \Sigma_t + \nabla l_t^\eta\left(\theta_t^{\eta,l}\right)\left(\nabla l_t^\eta\left(\theta_t^{\eta,l}\right)\right)^T$$

$$\theta_{t+1}^{\eta,l} = \Pi_{\mathcal{D}}^{\Sigma_{t+1}}\left(\theta_t^{\eta,l} - \frac{1}{\alpha}\Sigma_{t+1}^{-1}\nabla l_t^\eta\left(\theta_t^{\eta,l}\right)\right)$$

7: **end for**

---

**Algorithm 8** Strongly-convex Expert Algorithm

---

1: **Input:** Learning rate $\eta$
2: $\theta_1^{\eta,s} = 0$
3: **for** $t = 1, \ldots, T$ **do**
4:     Send $\theta^{\eta,s}$ to Algorithm 5
5:     Receive gradient $g_t$ from Algorithm 5
6:     Update

$$\theta_{t+1}^{\eta,s} = \Pi_{\mathcal{D}}^{I_d}\left(\theta_t^{\eta,s} - \frac{1}{2\eta^2 G^2 t}\nabla s_t^\eta\left(\theta_t^{\eta,s}\right)\right)$$

7: **end for**

---

We have:

$$\sum_{t=1}^T L(\theta_t) - L(\theta^\star) \leq \left(20GD + \frac{9G^2}{2\mu}\right)\left(2ln\left(\sqrt{3}\left(\frac{1}{2}\log_2 T + 3\right)\right) + 1 + \log T\right) + 2\sqrt{2G^2\log(1/\alpha)\sum_{t=1}^T \|\theta_t - \theta^\star\|^2}$$

$$\leq \left(20GD + \frac{9G^2}{2\mu}\right)\left(2ln\left(\sqrt{3}\left(\frac{1}{2}\log_2 T + 3\right)\right) + 1 + \log T\right) + 4\sqrt{\frac{G^2}{\mu}\log(1/\alpha)\sum_{t=1}^T L(\theta_t) - L(\theta^\star)}$$

Applying Proposition H.8:

$$\sum_{t=1}^T L(\theta_t) - L(\theta^\star) \leq \left(40GD + \frac{18G^2}{\mu}\right)\left(2ln\left(\sqrt{3}\left(\frac{1}{2}\log_2 T + 3\right)\right) + 1 + \log T\right) + \frac{32G^2\log(1/\alpha)}{\mu}$$

Furthermore, from strong convexity, we have: $\|\theta_t - \theta^\star\|^2 \leq \frac{2(L(\theta_t) - L(\theta^\star))}{\mu}$, thus:

$$\sum_{t=1}^T \|\theta_t - \theta^\star\|^2 \leq \left(\frac{80GD}{\mu} + \frac{36G^2}{\mu^2}\right)\left(2ln\left(\sqrt{3}\left(\frac{1}{2}\log_2 T + 3\right)\right) + 1 + \log T\right) + \frac{64G^2\log(1/\alpha)}{\mu^2}$$

$$\square$$

**Theorem C.3.** *(Strongly convex regret) Assuming $L(\theta) = E[l_t(\theta_t)]$ is $\mu-SC$ and $\max_{t,t'}\|\theta_t - \theta_t'\| \leq 2D$. Then w.p at least $1 - \alpha$, Maler under the event $\mathcal{E}$ guarantees:*

$$\sum_{t=1}^T l_t(\theta_t) - l_t(\theta^\star) \leq O\left(\left(\frac{G^2}{\mu} + GD\right)\log T + \frac{G^2}{\mu}\log(1/\alpha)\right)$$

*Proof.* Let us define $A_t = l_t(\theta_t) - l_t(\theta^\star) - (L(\theta_t) - L(\theta^\star))$. It's easy to see that $\mathbb{E}[A_t|A_1, \ldots, A_{t-1}] = 0$ and $\mathbb{E}[A_t] \leq \infty$. Thus the sequence $\{A_t\}_{t=1}^T$ is a Martingale difference sequence. Furthermore, we have

$$\begin{aligned}
\mathbb{E}[\|A_t\|^2|A_1, \ldots, A_t] &= \mathbb{E}[\|l_t(\theta_t) - l_t(\theta^\star) - (L(\theta_t) - L(\theta^\star))\|^2] \\
&\leq \mathbb{E}[2\|l_t(\theta_t) - l_t(\theta^\star)\|^2 + 2\|(L(\theta_t) - L(\theta^\star))\|^2] \\
&\leq 4G^2\|\theta_t - \theta^\star\|^2
\end{aligned}$$

where the last inequality comes from lipschitz assumption. Similarly, we also have $\|A_t\| \leq 2G\|\theta_t - \theta^\star\|$. Thus by Proposition H.1, $X_t$ is $(2G\|\theta_t - \theta^\star\|, 4G\|\theta_t - \theta^\star\|)$ sub-exponential. Now applying Theorem H.2, for some arbitrary constant $\nu \geq 0$, we have:

$$\|\sum_{t=1}^T l_t(\theta_t) - l_t(\theta^\star) - (L(\theta_t) - L(\theta^\star))\| \leq 5\sqrt{4G^2\sum_{t=1}^T \|\theta_t - \theta^\star\|^2 \log\left(\frac{16}{\alpha}\left[\log\left(\left[\sqrt{4G^2\sum_{t=1}^T \|\theta_t - \theta^\star\|^2/\nu^2}\right]_1\right) + 2\right]^2\right)}$$

$$+ 23\max(\nu, \max_{i \leq t} b_i) \log\left(\frac{224}{\alpha}\left[\log\left(\frac{2\max(\nu, \max_{i \leq t} b_i)}{\nu}\right) + 2\right]^2\right)$$

where $[x]_1 = \max(1, x)$. Now let $M_1 = 23\max(\nu, \max_{i \leq t} b_i) \log\left(\frac{224}{\alpha}\left[\log\left(\frac{2\max(\nu, \max_{i \leq t} b_i)}{\nu}\right) + 2\right]^2\right)$ and denote $Regret_T^{\text{SC}} = \sum_{t=1}^T L(\theta_t) - L(\theta^\star)$ we get:

$$\left\|\sum_{t=1}^T l_t(\theta_t) - l_t(\theta^\star) - (L(\theta_t) - L(\theta^\star))\right\| \leq 5\sqrt{\frac{8G^2 Regret_T^{\text{SC}}}{\mu} \log\left(\frac{16}{\alpha}\left[\log\left(\left[\sqrt{\frac{8G^2 Regret_T^{\text{SC}}}{\mu\nu^2}}\right]_1\right) + 2\right]^2\right)} + M_1$$

Thus w.p at least $1 - 2\alpha$

$$\sum_{t=1}^T l_t(\theta_t) - l_t(\theta^\star) \leq Regret_T^{\text{SC}} + 5\sqrt{\frac{8G^2 Regret_T^{\text{SC}}}{\mu} \log\left(\frac{16}{\alpha}\left[\log\left(\left[\sqrt{\frac{8G^2 Regret_T^{\text{SC}}}{\mu\nu^2}}\right]_1\right) + 2\right]^2\right)} + M_1$$

Plug in $Regret_T^{\text{SC}}$ from Theorem C.2:

$$\sum_{t=1}^T l_t(\theta_t) - l_t(\theta^\star) \leq O\left(\left(\frac{G^2}{\mu} + GD\right)\log T + \frac{G^2}{\mu}\log(1/\alpha)\right)$$

$\square$

## D  Proofs of section 3

### D.1  Proofs of general results

**Lemma 3.3.** *We define $\tilde{V}_{N-1} = \sum_{t=1}^{N-1} \tilde{x}_t \tilde{x}_t^T$, $\tilde{u}_{N-1} = \sum_{t=1}^{N-1} \langle \theta_t, \tilde{x}_t \rangle \tilde{x}_t$ ($\theta_t$ is the prediction of the online learner), and $\hat{\theta}_N = \tilde{V}_{N-1}^{-1} \tilde{u}_{N-1}$. Assuming $\|\theta_t\| \leq D$, then under event $\mathcal{E}$, the true parameter $\theta^\star$ lies in the set:*

$$C_{N-1} = \left\{\theta \in R^d : \|\theta - \hat{\theta}_N\|_{\tilde{V}_{N-1}}^2 \leq M_N + K_N\right\}$$

*for any $N \geq 1$ and*

$$K_N = \gamma D \Delta^2 \log(T/\alpha) \sqrt{N \sum_{t=1}^{N} \|\theta_t - \theta^\star\|^2} + \gamma \left( R + \frac{D\sqrt{\log(1/\delta)}}{\epsilon} + D\Delta \right)^2 \log(T/\alpha)$$

$$+ \gamma \left( \frac{\log(1/\delta)}{\epsilon^2} + \Delta^2 \right) \log(T/\alpha) \sum_{t=1}^{N} \|\theta_t - \theta^\star\|^2$$

*for a sufficient large constant $\gamma > 0$.*

*Proof.* We have:

$$M_n \geq \sum_{t=1}^{N} (\langle \tilde{x}_t, \theta_t \rangle - \tilde{y}_t)^2 - \|\theta_t\|_\Sigma^2 - (\langle \tilde{x}_t, \theta^\star \rangle - \tilde{y}_t)^2 + \|\theta^\star\|_\Sigma^2$$

$$= \sum_{t=1}^{N} (\langle x_t, \theta_t - \theta^\star \rangle + \langle \eta_{x,t}, \theta_t \rangle + \langle \zeta_t, \theta_t \rangle - r_t - \eta_{y,t})^2 - \|\theta_t\|_\Sigma^2 - (\langle \eta_{x,t}, \theta^\star \rangle + \langle \zeta_t, \theta^\star \rangle - r_t - \eta_{y,t})^2 + \|\theta^\star\|_\Sigma^2$$

Let $z_t = r_t + \eta_{y,t}$:

$$M_N \geq \sum_{t=1}^{N} (\langle x_t, \theta_t - \theta^\star \rangle)^2 + 2\langle x_t, \theta_t - \theta^\star \rangle \langle \eta_{x,t}, \theta_t \rangle + (\langle \eta_{x,t}, \theta_t \rangle)^2 + 2\left( \langle \zeta_t, \theta_t \rangle - z_t \right) \left( \langle x_t, \theta_t - \theta^\star \rangle + \langle \eta_{x,t}, \theta_t \rangle \right)$$

$$+ \langle \zeta_t, \theta_t \rangle^2 - 2z_t \langle \zeta_t, \theta_t \rangle + z_t^2 - \|\theta_t\|_\Sigma^2 - (\langle \eta_{x,t}, \theta^\star \rangle)^2 - 2(\langle \zeta_t, \theta^\star \rangle - z_t)\langle \eta_{x,t}, \theta^\star \rangle - \langle \zeta_t, \theta^\star \rangle^2$$

$$+ 2z_t \langle \zeta_t, \theta^\star \rangle - z_t^2 + \|\theta^\star\|_\Sigma^2$$

$$= \sum_{t=1}^{N} (\langle x_t, \theta_t - \theta^\star \rangle)^2 + 2(\langle \eta_{x,t} + \zeta_t, \theta_t \rangle - z_t)\langle x_t, \theta_t - \theta^\star \rangle - 2z_t \langle \eta_{x,t}, \theta_t - \theta^\star \rangle + \langle \zeta_t, \theta_t \rangle^2 - \langle \zeta_t, \theta^\star \rangle^2$$

$$+ 2(\langle \zeta_t, \theta_t \rangle \langle \eta_{x,t}, \theta_t \rangle - \langle \zeta_t, \theta^\star \rangle \langle \eta_{x,t}, \theta^\star \rangle) - 2z_t \langle \zeta_t, \theta_t - \theta^\star \rangle + \theta_t^T (\eta_{x,t} \eta_{x,t}^T - \Sigma)\theta_t - (\theta^\star)^T (\eta_{x,t} \eta_{x,t}^T - \Sigma)\theta^\star$$

$$= \sum_{t=1}^{N} (\langle x_t, \theta_t - \theta^\star \rangle)^2 + 2(\langle \eta_{x,t} + \zeta_t, \theta_t \rangle - z_t)\langle x_t, \theta_t - \theta^\star \rangle - 2z_t \langle \eta_{x,t}, \theta_t - \theta^\star \rangle + \langle \zeta_t, \theta_t \rangle^2 - \langle \zeta_t, \theta^\star \rangle^2$$

$$+ 2(\langle \zeta_t, \theta_t \rangle \langle \eta_{x,t}, \theta_t \rangle - \langle \zeta_t, \theta^\star \rangle \langle \eta_{x,t}, \theta^\star \rangle) - 2z_t \langle \zeta_t, \theta_t - \theta^\star \rangle + (\theta_t - \theta^\star)^T (\eta_{x,t} \eta_{x,t}^T - \Sigma)(\theta_t + \theta^\star)$$

Rearrange the terms:

$$\sum_{t=1}^{N} (\langle x_t, \theta_t - \theta^\star \rangle)^2 \leq M_N \sum_{t=1}^{N} \underbrace{-2(\langle \eta_{x,t} + \zeta_t, \theta_t \rangle - z_t)\langle x_t, \theta_t - \theta^\star \rangle}_{A_t} \underbrace{+2z_t \langle \eta_{x,t} + \zeta_t, \theta_t - \theta^\star \rangle}_{B_t} \underbrace{-\langle \zeta_t, \theta_t \rangle^2 + \langle \zeta_t, \theta^\star \rangle^2}_{C_t}$$

$$\underbrace{-2(\langle \zeta_t, \theta_t \rangle \langle \eta_{x,t}, \theta_t \rangle - \langle \zeta_t, \theta^\star \rangle \langle \eta_{x,t}, \theta^\star \rangle)}_{D_t} \underbrace{-(\theta_t - \theta^\star)^T (\eta_{x,t} \eta_{x,t}^T - \Sigma)(\theta_t + \theta^\star)}_{E_t}$$

**Bounding $\sum_{t=1}^{N} A_t$:** Using the result from $\mathcal{E}_5$:

$$\left| \sum_{t=1}^{N} (z_t - \langle \eta_{x,t} + \zeta_t, \theta_t \rangle)\langle x_t, \theta_t - \theta^\star \rangle \right| \leq \left( R + \frac{2(D+1)\sqrt{2\log(1.25/\delta)}}{\epsilon} + D\Delta \right) \sqrt{2\left( 1 + \sum_{t=1}^{N} \langle x_t, \theta_t - \theta^\star \rangle^2 \right)}$$

$$\times \sqrt{\log\left( \frac{\sqrt{1 + \sum_{t=1}^{N} \langle x_t, \theta_t - \theta^\star \rangle^2}}{\alpha} \right)}$$

$$\leq \tilde{O}\left( \left( R + \frac{D\sqrt{\log(1/\delta)}}{\epsilon} + D\Delta \right) \sqrt{\log(N/\alpha) \sum_{t=1}^{N} \langle x_t, \theta_t - \theta^\star \rangle^2} \right)$$

**Bounding $\sum_{t=1}^{N} B_t$:** From $\mathcal{E}_6$,

$$|\sum_{t=1}^{N} z_t \langle \eta_{x,t} + \zeta_t, \theta_t - \theta^\star \rangle| \le \left( R + \frac{2\sqrt{2\log(1.25/\delta)}}{\epsilon} \right) \sqrt{2 \left( 1 + \sum_{t=1}^{N} \langle \eta_{x,t} + \zeta_t, \theta_t - \theta^\star \rangle^2 \right)}$$

$$\times \sqrt{\log\left( \frac{\sqrt{1 + \sum_{t=1}^{N} \langle \eta_{x,t} + \zeta_t, \theta_t - \theta^\star \rangle^2}}{\alpha} \right)}$$

Since we're under good event $\mathcal{E}$:

$$\le \left( R + \frac{2\sqrt{2\log(1.25/\delta)}}{\epsilon} \right) \sqrt{2 \left( 1 + 2\log(2T/\alpha) \sum_{t=1}^{N} (\sigma^2 + \Delta^2) \|\theta_t - \theta^\star\|^2 \right)}$$

$$\times \sqrt{\log\left( \frac{\sqrt{1 + 2\log(2T/\alpha) \sum_{t=1}^{N} (\sigma^2 + \Delta^2) \|\theta_t - \theta^\star\|^2}}{\alpha} \right)}$$

$$\le O\left( \left( R + \frac{\sqrt{\log(1/\delta)}}{\epsilon} \right) \sqrt{\log^2(2T/\alpha) \sum_{t=1}^{N} \left( \frac{\log(1/\delta)}{\epsilon^2} + \Delta^2 \right) \|\theta_t - \theta^\star\|^2} \right)$$

**Bounding $\sum_{t=1}^{N} C_t$:** We have:

$$\sum_{t=1}^{N} \langle \zeta_t, \theta^\star \rangle^2 - \langle \zeta_t, \theta_t \rangle^2 = \sum_{t=1}^{N} \langle \zeta_t, \theta^\star - \theta_t \rangle \langle \zeta_t, \theta_t + \theta^\star \rangle$$

Using $\|\theta_t + \theta^\star\| \le D + 1$,

$$\sum_{t=1}^{N} \langle \zeta_t, \theta^\star \rangle^2 - \langle \zeta_t, \theta_t \rangle^2 \le \Delta(D+1)\sqrt{2\log(2T/\alpha)} \sum_{t=1}^{N} |\langle \zeta_t, \theta^\star - \theta_t \rangle|$$

$$\le \Delta(D+1)\sqrt{2\log(2T/\alpha)} \sqrt{N \sum_{t=1}^{N} \langle \zeta_t, \theta^\star - \theta_t \rangle^2}$$

$$\le \Delta(D+1)\sqrt{2\log(2T/\alpha)} \sqrt{2N \log(2T/\alpha)\Delta^2 \sum_{t=1}^{N} \|\theta_t - \theta^\star\|^2}$$

$$\le O\left( D\Delta^2 \log(T/\alpha) \sqrt{N \sum_{t=1}^{N} \|\theta_t - \theta^\star\|^2} \right)$$

**Bounding $\sum_{t=1}^{N} D_t$:** We have

$$\sum_{t=1}^{N} \langle \zeta_t, \theta^\star \rangle \langle \eta_{x,t}, \theta^\star \rangle - \langle \zeta_t, \theta_t \rangle \langle \eta_{x,t}, \theta_t \rangle = (\theta^\star - \theta_t)^T \zeta_t \eta_{x,t}^T (\theta^\star - \theta_t) + \theta_t^T \zeta_t \eta_{x,t}^T \theta^\star + (\theta^\star)^T \zeta_t \eta_{x,t}^T \theta_t - 2\theta_t^T \zeta_t \eta_{x,t}^T \theta_t$$

$$= (\theta^\star - \theta_t)^T \zeta_t \eta_{x,t}^T (\theta^\star - \theta_t) + \theta_t^T \zeta_t \eta_{x,t}^T (\theta^\star - \theta_t) + (\theta^\star - \theta_t)^T \zeta_t \eta_{x,t}^T \theta_t$$

From $\mathcal{E}_7$:

$$\left| \sum_{t=1}^{N} (\theta^\star - \theta_t)^T \zeta_t \eta_{x,t}^T (\theta^\star - \theta_t) \right| \leq \Delta \|\theta_t - \theta^\star\| \sqrt{2 \left( 1 + \sum_{t=1}^{N} \langle \eta_{x,t}, \theta^\star - \theta_t \rangle^2 \right) \log \left( \frac{\sqrt{1 + \sum_{t=1}^{N} \langle \eta_{x,t}, \theta^\star - \theta_t \rangle^2}}{\alpha} \right)}$$

$$\leq O \left( D\Delta \sqrt{\log(T/\alpha) \sum_{t=1}^{N} \sigma^2 \|\theta_t - \theta^\star\|^2} \right)$$

$$= O \left( \frac{D\Delta \sqrt{\log(1/\delta)}}{\epsilon} \sqrt{\log(T/\alpha) \sum_{t=1}^{N} \|\theta_t - \theta^\star\|^2} \right)$$

Since $\mathcal{E}_8 \subset \mathcal{E}$:

$$\left| \sum_{t=1}^{N} \theta_t^T \zeta_t \eta_{x,t}^T (\theta^\star - \theta_t) \right| + \left| \sum_{t=1}^{N} (\theta^\star - \theta_t)^T \zeta_t \eta_{x,t}^T \theta_t \right|$$

$$\leq \Delta \|\theta_t\| \sqrt{2 \left( 1 + \sum_{t=1}^{N} \langle \eta_{x,t}, \theta^\star - \theta_t \rangle^2 \right) \log \left( \frac{\sqrt{1 + \sum_{t=1}^{N} \langle \eta_{x,t}, \theta^\star - \theta_t \rangle^2}}{\alpha} \right)}$$

$$+ \sigma \|\theta_t\| \sqrt{2 \left( 1 + \sum_{t=1}^{N} \langle \zeta_t, \theta^\star - \theta_t \rangle^2 \right) \log \left( \frac{\sqrt{1 + \sum_{t=1}^{N} \langle \zeta_t, \theta^\star - \theta_t \rangle^2}}{\alpha} \right)}$$

$$\leq O \left( D\Delta \sqrt{\sum_{t=1}^{N} \langle \eta_{x,t}, \theta^\star - \theta_t \rangle^2 \log (T/\alpha)} + D\sigma \sqrt{\sum_{t=1}^{N} \langle \zeta_t, \theta^\star - \theta_t \rangle^2 \log (T/\alpha)} \right)$$

Plugging in $\sigma$:

$$\leq O \left( \frac{D\Delta \sqrt{\log(1/\delta)}}{\epsilon} \sqrt{\log(T/\alpha) \sum_{t=1}^{N} \|\theta^\star - \theta_t\|^2} \right)$$

Overall,

$$\sum_{t=1}^{N} \langle \zeta_t, \theta^\star \rangle \langle \eta_{x,t}, \theta^\star \rangle - \langle \zeta_t, \theta_t \rangle \langle \eta_{x,t}, \theta_t \rangle \leq O \left( \frac{D\Delta \sqrt{\log(1/\delta)}}{\epsilon} \sqrt{\log(T/\alpha) \sum_{t=1}^{N} \|\theta_t - \theta^\star\|^2} \right)$$

**Bounding $\sum_{t=1}^{N} E_t$:** From $\mathcal{E}_9$:

$$\left\| \sum_{t=1}^{N} (\theta^\star - \theta_t)^T (\eta_{x,t} \eta_{x,t}^T - \Sigma)(\theta_t + \theta^\star) \right\|$$

$$\leq 20\sigma^2(D+1)\log(2T/\alpha)\sqrt{\sum_{t=1}^{N} \|\theta^\star - \theta_t\|^2 \log\left(\frac{16}{\alpha}\left[\log\left(e^2\left[\sqrt{\sum_{t=1}^{N}\frac{16(D+1)^2\sigma^4\log^2\left(\frac{2T}{\alpha}\right)\|\theta^\star - \theta_t\|^2}{\nu^2}}\right]\right)\right]_1\right)^2}$$

$$+ 23\max(\nu, \max_t 4(D+1)\sigma^2\log\left(\frac{2T}{\alpha}\right)\|\theta^\star - \theta_t\|)\log\left(\frac{224}{\alpha}\left[\log\left(\frac{2e^2\max(\nu, \max_t 4(D+1)\sigma^2\log\left(\frac{2T}{\alpha}\right)\|\theta^\star - \theta_t\|)}{\nu}\right)\right]^2\right)$$

$$\leq \tilde{O}\left(\frac{D\log(1/\delta)\log(T/\alpha)}{\epsilon^2}\sqrt{\sum_{t=1}^{N}\|\theta_t - \theta^\star\|^2}\right)$$

Now combine all the bounds we get:

$$\sum_{t=1}^{N}(\langle x_t, \theta_t - \theta^\star\rangle)^2 \leq M_N + O\left(\left(R + \frac{D\sqrt{\log(1/\delta)}}{\epsilon} + D\Delta\right)\sqrt{\log(T/\alpha)\sum_{t=1}^{N}\langle x_t, \theta_t - \theta^\star\rangle^2}\right)$$

$$+ O\left(\left(R + \frac{\sqrt{\log(1/\delta)}}{\epsilon}\right)\sqrt{\log^2(T/\alpha)\sum_{t=1}^{N}\left(\frac{\log(1/\delta)}{\epsilon^2} + \Delta^2\right)\|\theta_t - \theta^\star\|^2}\right)$$

$$+ O\left(D\Delta^2\log(T/\alpha)\sqrt{N\sum_{t=1}^{N}\|\theta_t - \theta^\star\|^2}\right)$$

$$+ O\left(\frac{D\Delta\sqrt{\log(1/\delta)}}{\epsilon}\sqrt{\log(T/\alpha)\sum_{t=1}^{N}\|\theta_t - \theta^\star\|^2}\right) + \tilde{O}\left(\frac{D\log(1/\delta)\log(T/\alpha)}{\epsilon^2}\sqrt{\sum_{t=1}^{N}\|\theta_t - \theta^\star\|^2}\right)$$

$$\leq M_N + O\left(\left(R + \frac{D\sqrt{\log(1/\delta)}}{\epsilon} + D\Delta\right)\sqrt{\log(T/\alpha)\sum_{t=1}^{N}\langle x_t, \theta_t - \theta^\star\rangle^2}\right)$$

$$+ O\left(\left(R + \frac{\sqrt{\log(1/\delta)}}{\epsilon}\right)\sqrt{\log^2(T/\alpha)\sum_{t=1}^{N}\left(\frac{\log(1/\delta)}{\epsilon^2} + \Delta^2\right)\|\theta_t - \theta^\star\|^2}\right)$$

$$+ O\left(D\Delta^2\log(T/\alpha)\sqrt{N\sum_{t=1}^{N}\|\theta_t - \theta^\star\|^2}\right) + \tilde{O}\left(\frac{D\log(1/\delta)\log(T/\alpha)}{\epsilon^2}\sqrt{\sum_{t=1}^{N}\|\theta_t - \theta^\star\|^2}\right)$$

Applying Proposition H.9 and Proposition H.8 to get:

$$\sum_{t=1}^{N}(\langle x_t, \theta_t - \theta^\star\rangle)^2 \leq O\left(M_N + \left(R + \frac{\sqrt{\log(1/\delta)}}{\epsilon}\right)\sqrt{\log^2(T/\alpha)\sum_{t=1}^{N}\left(\frac{\log(1/\delta)}{\epsilon^2} + \Delta^2\right)\|\theta_t - \theta^\star\|^2}\right)$$

$$+ O\left(D\Delta^2\log(T/\alpha)\sqrt{N\sum_{t=1}^{N}\|\theta_t - \theta^\star\|^2}\right) + \tilde{O}\left(\frac{D\log(1/\delta)\log(T/\alpha)}{\epsilon^2}\sqrt{\sum_{t=1}^{N}\|\theta_t - \theta^\star\|^2}\right)$$

$$+ O\left(\left(R + \frac{D\sqrt{\log(1/\delta)}}{\epsilon} + D\Delta\right)^2\log(T/\alpha)\right)$$

Further:

$$
\begin{aligned}
\sum_{t=1}^{N}(\langle \tilde{x}_t, \theta_t - \theta^\star\rangle)^2 &= \sum_{t=1}^{N}(\langle x_t, \theta_t - \theta^\star\rangle + \langle \eta_{x,t} + \zeta_t, \theta_t - \theta^\star\rangle)^2 \\
&\leq \sum_{t=1}^{N} 2(\langle x_t, \theta_t - \theta^\star\rangle)^2 + 2(\langle \eta_{x,t} + \zeta_t, \theta_t - \theta^\star\rangle)^2 \\
&\leq \sum_{t=1}^{N} 2(\langle x_t, \theta_t - \theta^\star\rangle)^2 + 4\left(\frac{8\log(1.25/\delta)}{\epsilon^2} + \Delta^2\right)\log(2T/\alpha)\sum_{t=1}^{N}\|\theta_t - \theta^\star\|^2 \\
&\leq O\left(D\Delta^2\log(T/\alpha)\sqrt{N\sum_{t=1}^{N}\|\theta_t - \theta^\star\|^2} + \left(R + \frac{D\sqrt{\log(1/\delta)}}{\epsilon} + D\Delta\right)^2\log(T/\alpha)\right) \\
&\quad + O\left(\left(\frac{\log(1/\delta)}{\epsilon^2} + \Delta^2\right)\log(T/\alpha)\sum_{t=1}^{N}\|\theta_t - \theta^\star\|^2\right) + M_N
\end{aligned}
$$

Let $\gamma$ be a sufficiently large positive constant and

$$
\begin{aligned}
K_N &= \gamma D\Delta^2\log(T/\alpha)\sqrt{N\sum_{t=1}^{N}\|\theta_t - \theta^\star\|^2} + \gamma\left(R + \frac{D\sqrt{\log(1/\delta)}}{\epsilon} + D\Delta\right)^2\log(T/\alpha) \\
&\quad + \gamma\left(\frac{\log(1/\delta)}{\epsilon^2} + \Delta^2\right)\log(T/\alpha)\sum_{t=1}^{N}\|\theta_t - \theta^\star\|^2
\end{aligned}
$$

Then,

$$
\sum_{t=1}^{N}(\langle \tilde{x}_t, \theta_t - \theta^\star\rangle)^2 \leq M_N + K_N
$$

Let us denote the set $C_{n-1}$ as an ellipsoid underlying the covariance matrix $\tilde{V}_{N-1} = I + \sum_{t=1}^{N-1}\tilde{x}_t\tilde{x}_t^T$ and centering at

$$
\begin{aligned}
\hat{\theta}_N &= \arg\min_{\theta \in R^d}\left(\|\theta\|_2^2 + \sum_{t=1}^{N-1}(\langle \tilde{x}_t, \theta_t - \theta\rangle)^2\right) \\
&= \tilde{V}_{N-1}^{-1}\left(\sum_{t=1}^{N-1}\langle \theta_t, \tilde{x}_t\rangle\tilde{x}_t\right) \\
&= \tilde{V}_{N-1}^{-1}\tilde{u}_{N-1}
\end{aligned}
$$

We can thus express the ellipsoid as:

$$
\hat{C}_{N-1} = \left\{\theta \in R^d : (\theta - \hat{\theta}_n)^T\tilde{V}_{n-1}(\theta - \hat{\theta}_N) + \|\hat{\theta}_N\|_2^2 + \sum_{t=1}^{N-1}(\langle \tilde{x}_t, \theta_t - \hat{\theta}_N\rangle)^2 \leq M_N + K_N\right\}
$$

The ellipsoid is contained in a larger ellipsoid

$$
\hat{C}_{N-1} \subseteq C_{N-1} = \left\{\theta \in R^d : \|\theta - \hat{\theta}_N\|_{\tilde{V}_{N-1}}^2 \leq M_N + K_N\right\}
$$

Thus, $\theta^\star$ lies in $C_{N-1}$ with high probability. $\qquad\square$

**Theorem 3.4.** *(Utility guarantee) Recall that $M_T$ is the regret of our online learner (see equation (4)), and $K_T$ is as defined in Lemma 3.3. Under event $\mathcal{E}$, the regret of Algorithm 1 is:*

$$Regret_T \leq \tilde{O}\left(\sqrt{M_T + K_T}\sqrt{2Td\log\left(1 + \frac{T}{d}\right)}\right)$$

*Proof.* First we bound the instantaneous regret using $(x_t, \tilde{\theta}_t) = \arg\max_{(x,\theta) \in D_t \times C_{t-1}} \langle x, \theta \rangle$:

$$\begin{aligned}
\langle x^\star, \theta^\star \rangle - \langle x_t, \theta^\star \rangle &\leq \langle x_t, \tilde{\theta}_t \rangle - \langle x_t, \theta^\star \rangle \\
&= \langle x_t, \tilde{\theta}_t - \theta^\star \rangle \\
&= \langle x_t, \tilde{\theta}_t - \hat{\theta}_t \rangle + \langle x_t, \hat{\theta}_t - \theta^\star \rangle \\
&\leq \|x_t\|_{\tilde{V}_{t-1}^{-1}} \|\tilde{\theta}_t - \hat{\theta}_t\|_{\tilde{V}_{t-1}} + \|x_t\|_{\tilde{V}_{t-1}^{-1}} \|\hat{\theta}_t - \theta^\star\|_{\tilde{V}_{t-1}} \\
&\leq \tilde{O}\left(\sqrt{M_{t-1} + K_{t-1}}\right) \times \|x_t\|_{\tilde{V}_{t-1}^{-1}}
\end{aligned}$$

Use the assumption $|\langle x, \theta^\star \rangle| \leq 1$ and sum over all t to get the regret:

$$\begin{aligned}
Regret_T &= \sum_{t=1}^{T} \langle x^\star - x_t, \theta^\star \rangle \\
&\leq \sum_{t=1}^{T} \tilde{O}\left(\sqrt{M_T + K_T}\right) \min\{1, \|x_t\|_{\tilde{V}_{t-1}^{-1}}\} \\
&\leq \tilde{O}\left(\sqrt{M_T + K_T}\right) \times \sqrt{T \times \sum_{t=1}^{T} \min\{1, \|x_t\|_{\tilde{V}_{t-1}^{-1}}^2\}}
\end{aligned}$$

Applying Lemma H.3, we get:

$$Regret_T \leq \tilde{O}\left(\sqrt{M_T + K_T}\sqrt{2Td\log\left(1 + \frac{T}{d}\right)}\right)$$

$\square$

## E  Anytime version of Algorithm 1

Since the Bandit Combiner Algorithm (Algorithm 2) requires the base learner to have anytime guarantee, we can not directly use Algorithm 1 for all the base learners since the regret of Algorithm 1 depends on the total iterations $T$ when $\Delta^2 \neq 0$. Fortunately, we can easily convert any non-anytime algorithm to an anytime algorithm using the doubling trick. We will describe the anytime version of Algorithm 1 in Algorithm 9: We

---

**Algorithm 9** Anytime Private (Contextual) Online LinUCB

---

**Input:** Privacy parameters $\epsilon$, $\delta$, failure parameter $\alpha$, covariance matrix $\Sigma = \mathbb{E}[\eta_{x,t}\eta_{x,t}^T]$, minimum eigenvalue $\lambda_{min}$ of $\mathbb{E}[x_t x_t^T]$, domain diameter $D$, universal constant $C$, power $k$.
**for** $m = 0, 1, 2, \ldots$ **do**
    Initialize Algorithm 1 with $\epsilon$, $\delta$, $\alpha$, $\Sigma$, $\lambda_{min}$, $D$, $C$, and set $\bar{\lambda} = \Delta_m^2 = \frac{1}{(2^m)^k}$.
    Run Algorithm 1 for $[2^m, 2^{m+1} - 1]$ rounds.
    Reset Algorithm 1.
**end for**

---

have the following guarantee:

**Theorem E.1.** *Let $T$ be the maximum number of iterations and $P > 0$ be an absolute constant. For any $t \in [T]$, under event $\mathcal{E}$, Algorithm 9 guarantees:*

$$Regret_t \leq P \log^{3/2}(t) \times \left( \sqrt{d} t^{3/4} + \frac{\sqrt{d} t^{5/8}}{\epsilon} \right)$$

*Proof.* In Algorithm 9, in each round $m$, each copy of Algorithm 1 runs at most $T_m = 2^m$ iterations. From Corollary 3.6, we have the regret of Algorithm 1 after $T_m$ iterations is upper bounded by:

$$O \left( \left( \left( R + \frac{D\sqrt{\log(1/\delta)}}{\epsilon} \right) \sqrt{\log(T_m/\alpha)} + \sqrt{D\Delta_m^2 \log(T_m/\alpha)} \sqrt{T_m H_m \log T_m} + \frac{\log T_m \sqrt{H_m \log(1/\delta)}}{\epsilon} \right) \right.$$
$$\left. \times \sqrt{dT_m \log(T_m/d)} \right)$$

where $H_m = \frac{G^2}{(\lambda_{min}+\Delta_m^2)^2} + \frac{GD}{\lambda_{min}+\Delta_m^2}$. Thus, there exists absolute constant $K > 0$ such that the regret of Algorithm 1 of round $m \in [\lceil \log_2 t \rceil]$ is:

$$Regret_m \leq K \log^{3/2}(T_m) \times \left( \sqrt{\Delta_m^2 \sqrt{H_m}} \sqrt{d} T_m^{3/4} + \frac{\sqrt{H_m}}{\epsilon} \sqrt{dT_m} \right)$$

Denote $M = \lfloor \log_2 t \rfloor$, then the total regret of Algorithm 9 at any iteration $t \in [T]$ is:

$$Regret_t \leq \sum_{m=0}^{M} Regret_m$$
$$\leq \sum_{m=0}^{M} K \log^{3/2}(T_m) \times \left( \sqrt{\Delta_m^2 \sqrt{H_m}} \sqrt{d} T_m^{3/4} + \frac{\sqrt{H_m}}{\epsilon} \sqrt{dT_m} \right)$$

Since $\log(\cdot)$ is monotonically increasing and $T_m \leq t$,

$$\leq K \log^{3/2}(t) \sum_{m=0}^{M} \left( \sqrt{\Delta_m^2 \sqrt{H_m}} \sqrt{d} T_m^{3/4} + \frac{\sqrt{H_m}}{\epsilon} \sqrt{dT_m} \right)$$

We have for every $m \in [M]$ that:

$$\sqrt{\Delta_m^2 \sqrt{H_m}} \leq \sqrt{\Delta_m^2 \left( \frac{G}{\lambda_{min} + \Delta_m^2} + \frac{\sqrt{GD}}{\sqrt{\lambda_{min} + \Delta_m^2}} \right)}$$
$$\leq \sqrt{\Delta_m^2 \left( \frac{G}{\Delta_m^2} + \frac{\sqrt{GD}}{\sqrt{\Delta_m^2}} \right)}$$
$$= \sqrt{G + \lambda_m \sqrt{GD}}$$

Since $\lambda_m \leq 1$,

$$\leq \sqrt{G + \sqrt{GD}}$$

Since $\lambda_m$ decreases as $m$ increases, $H_m \leq H_M \leq H_t = \frac{G^2}{(\lambda_{min}+\Delta_t^2)^2} + \frac{GD}{\lambda_{min}+\Delta_t^2}$ where $\Delta_t^2 = \frac{1}{t^k}$, then:

$$Regret_t \leq K\log^{3/2}(t)\left(\sqrt{d(G+\sqrt{GD})}\left(\sum_{m=0}^{M}T_m^{3/4}\right) + \frac{\sqrt{dH_t}}{\epsilon}\left(\sum_{m=1}^{M}\sqrt{T_m}\right)\right)$$

$$\leq K\log^{3/2}(t)\left(\sqrt{d(G+\sqrt{GD})}\left(\sum_{m=0}^{\lfloor\log_2 t\rfloor}(2^{3/4})^m\right) + \frac{\sqrt{dH_t}}{\epsilon}\left(\sum_{m=0}^{\lfloor\log_2 t\rfloor}(\sqrt{2})^m\right)\right)$$

$$= K\log^{3/2}(t)\left(\sqrt{d(G+\sqrt{GD})}\frac{(2^{3/4})^{\lfloor\log_2 t\rfloor+1}-1}{2^{3/4}-1} + \frac{\sqrt{dH_t}}{\epsilon}\frac{(2^{1/2})^{\lfloor\log_2 t\rfloor+1}-1}{2^{1/2}-1}\right)$$

$$\leq K\log^{3/2}(t)\left(3\sqrt{d(G+\sqrt{GD})}t^{3/4} + \frac{4\sqrt{dH_t}}{\epsilon}\sqrt{t}\right)$$

Now we can choose an absolute constant $P$ sufficiently large to conclude the result. $\qquad\square$

## F   Proofs of section 4

The result of Algorithm 2 is simply a straightforward application of the result of Algorithm 1 in (Cutkosky et al., 2020). The only difference is the concentration result in Lemma 8 (Cutkosky et al., 2020) since the reward $\tilde{y}_t$ is not bounded by $[-1, 1]$ due to the added noise to ensure privacy. We will prove a modified result of Lemma 8 below. For ease of analysis, let us redefine some notations used in (Cutkosky et al., 2020). Let $r(a, c) := \langle \theta^\star, \phi(c, a)\rangle$ and $\hat{r}(c, a, \eta) := \langle \theta^\star, \phi(c, a)\rangle + \eta$. We also use the shorthand notations $r_t^i = r(a_t^i, c_t)$ and $\hat{r}_t = \hat{r}(a_t^i, c_t, \eta_t)$ to denote the random reward and the expected reward that the base learner $i$ receives at round t. Thus $\mathbb{E}_{\eta_t, c_t}[\hat{r}_t] = r_t$ and $\hat{r}_t = \tilde{y}_t$. We have the following lemma:

**Lemma F.1.** *For all rounds $t \in [T]$ and base learner $i \in [M]$, the following inequalities hold*

$$-k\sqrt{T(i,t)(1+\sigma^2)\log\left(\frac{T^3M\log T(i,t)(1+\sigma^2)}{\alpha}\right)} \leq \sum_{\tau=1}^{T(i,t)}\hat{r}_t^i - r_t^i \leq k\sqrt{T(i,t)(1+\sigma^2)\log\left(\frac{T^3M\log T(i,t)(1+\sigma^2)}{\alpha}\right)}$$

*w.p at least $1 - \alpha/(T^3M)$*

*Proof.* The proof is based on the proof of Lemma B.2 in (Pacchiano et al., 2023). Let $\mathcal{F}_t$ be the sigma-field induced by all variables up to round $t$ before the reward is revealed, i.e., $\mathcal{F}_t = \sigma\left(\{a_l, c_l, i_l, \eta_l\}_{l\in[t-1]} \cup \{a_t, c_t, i_t, \eta_t\}\right)$. Since $\eta_{y,t}$ is mean zero Gaussian noise for all $t \in [T]$, $\{X_t = \hat{r}_t^i - r_t^i\}_{t=1}^{T(i,t)}$ is a martingale difference sequence w.r.t $\mathcal{F}_t$. Using the terminology and definition in (Howard et al., 2021), the process $S_{T(i,t)} = \sum_{t=1}^{T(i,t)}X_t$ is a sub-$\psi_N$ with variance process $V_{T(i,t)} = T(i,t)(\sigma^2+1)$. Thus using the boundary choice in Eq.11 of (Howard et al., 2021), we get:

$$S_{T(i,t)} \leq 1.7\sqrt{V_{T(i,t)}(\log\log(2V_{T(i,t)})) + 0.72\log(5.2/\alpha)}$$
$$= 1.7\sqrt{T(i,t)(\sigma^2+1)(\log\log(2T(i,t)(\sigma^2+1)) + 0.72\log(5.2/\alpha)}$$

Applying the same argument to $-S_{T(i,t)}$ gives that:

$$|S_{T(i,t)}| \leq 3 \vee 1.7\sqrt{T(i,t)(\sigma^2+1)(\log\log(2T(i,t)(\sigma^2+1)) + 0.72\log(10.4/\alpha)}$$

Now, set $\alpha = \alpha/(T^3M)$, and pick the absolute constant $k$ sufficiently large to conclude the proof. $\qquad\square$

The rest of the analysis follows the analysis in (Cutkosky et al., 2020). We have the following general regret guarantee for UCB combiner algorithm.

**Theorem F.2.** *(Corollary 2 (Cutkosky et al., 2020)) Suppose $j$ is the index of the best base learner and w.p at least $1 - \alpha$, we have $\sum_{t=1}^{\tau} \max_{x \in \mathcal{D}_t} \langle \theta^\star, x \rangle - \langle \theta^\star, x_\tau^j \rangle \leq L_j t^{\alpha_j}$. Further, suppose we are given $M$ positive real numbers $\eta_1, \ldots, \eta_M$. Set $R_i$ via:*

$$R_i = L_i T^{\alpha_i} + \frac{(1 - \alpha_i)^{\frac{1-\alpha_i}{\alpha_i}} (1 + \alpha_i)^{\frac{1}{\alpha_i}}}{\alpha_i^{\frac{1-\alpha_i}{\alpha_i}}} L_i^{\frac{1}{\alpha_i}} T \eta_i^{\frac{1-\alpha_i}{\alpha_i}} + 288 \log(T^3 N / \alpha) T \eta_i + \sum_{k \neq i} \frac{1}{\eta_k}$$

*Then, w.p at least $1 - 3\alpha$, the regret of Algorithm 2 satisfies:*

$$Regret_T \leq \tilde{O}\left( L_j T^{\alpha_j} + L_j^{\frac{1}{a_j}} T \eta_j^{\frac{1-\alpha_j}{\alpha_j}} + T \eta_j + \sum_{k \neq i} \frac{1}{\eta_k} \right)$$

Before we state the regret of Algorithm 2, let us first show that with our settings of $\lambda_i$, we are guaranteed to have at least a base learner that has a regret bound that is the same up to some constant factor as the regret bound of a base learner that is initialized with the correct value of $\lambda_{min}$.

**Lemma F.3.** *Let $\lambda_{min}^\star$ be the actual minimum eigenvalue of $\mathbb{E}[x_t x_t^T]$ and $Regret_T^\star$ be the regret of an instance of Algorithm 1 using the correct minimum eigenvalue. Let $\lambda_i = \frac{2^{i-1}}{T^{1/8}}$ for $i \in [M]$ be the $\lambda_{min}$ of the base learner $i$ in Algorithm 2. If $M = \lceil \frac{1}{8} \log_2 T \rceil + 1$, then there exists at least a base learner $i$ with regret guarantee $Regret_T^i$ such that:*

$$Regret_T^i = K Regret_T^\star$$

*for some constant $K > 0$.*

*Proof.* Let us consider the case where $\lambda_{min}^\star \geq \frac{1}{T^{1/8}}$. Since $\|x_t\| \leq 1$ for all $x_t \in \mathcal{D}_t$ and $\lambda_{min}^\star$ is the minimum eigenvalue of $\mathbb{E}[x_t x_t^T]$, $\frac{1}{T^{1/8}} \leq \lambda_{min}^\star \leq 1$. Thus,

$$\frac{2^0}{T^{1/8}} \leq \lambda_{min}^\star \leq \frac{2^{(1/8 \log_2 T + 1 - 1)}}{T^{1/8}}$$
$$\Leftrightarrow \quad \lambda_1 \leq \lambda_{min}^\star \leq \lambda_M$$

In other words, the actual minimum eigenvalue $\lambda_{min}^\star$ is always within the range covered by our guess of $\lambda_{min}$. If $\lambda_{min}^\star = \lambda_1$ or $\lambda_{min}^\star = \lambda_M$, then the first statement of the lemma is true with $K = 1$. If this is not the case, that means there exist a learner $k$ such that $\lambda_k \leq \lambda_{min}^\star \leq \lambda_{k+1}$. We have:

$$\frac{\lambda_{min}^\star}{\lambda_k} \leq \frac{\lambda_{k+1}}{\lambda_k} = 2$$

Since the regret of any instance of Algorithm 2 that uses $\lambda \leq \lambda_{min}^\star$ as the input is $\tilde{O}\left( \frac{\sqrt{dT}}{\lambda} \right)$, the regret of the base learner $k$ with $\lambda_k$ is at most a constant factor worse than $Regret_T^\star$.

When $\lambda_{min}^\star < \frac{1}{T^{1/8}}$, since we also set the threshold $\bar{\lambda} = \frac{1}{T^{1/8}}$, from Theorem E.1, we know that the first base learner guarantees:

$$Regret_t \leq P \log^{3/2}(T) \times \left( \sqrt{d} T^{3/4} + \frac{\sqrt{d} T^{5/8}}{\epsilon} \right)$$

for a sufficiently large positive constant $P$. Since we also have $R_T^1 \leq \tilde{O}\left( \log^{3/2}(T) \times \left( \sqrt{d} T^{3/4} + \frac{\sqrt{d} T^{5/8}}{\epsilon} \right) \right)$, $Regret_T^1 = O(Regret_T^\star)$. □

We are now ready to state our guarantee for Algorithm 2.

**Theorem 4.1.** *(see Corollary 2 (Cutkosky et al., 2020)) Let* $\eta_1 = \frac{\epsilon T^{1/8}}{\sqrt{dT}(P \log^{3/2}(T)\epsilon T^{1/8}+1)}$ *and* $\eta_i = \frac{\epsilon}{P' \log^{3/2}(T)\sqrt{dT}}$, $L_1 = P \log^{3/2}(T)\sqrt{d}\left(\frac{\epsilon T^{1/8}+1}{\epsilon T^{1/8}}\right)$ *and* $L_i = P' \log^{3/2}(T)\frac{\sqrt{d}}{\epsilon \lambda_i}$ *for positive constants* $P$ *and* $P'$, $\alpha_1 = \frac{3}{4}$ *and* $\alpha_i = \frac{1}{2}$ *for* $i \in [2, M]$, *and set* $R_i$ *via:*

$$R_i = L_i T^{\alpha_i} + \frac{(1-\alpha_i)^{\frac{1-\alpha_i}{\alpha_i}}(1+\alpha_i)^{\frac{1}{\alpha_i}}}{\alpha_i^{\frac{1-\alpha_i}{\alpha_i}}} L_i^{\frac{1}{\alpha_i}} T \eta_i^{\frac{1-\alpha_i}{\alpha_i}} + 288 \log(T^3 N/\alpha)T\eta_i + \sum_{k \neq i}\frac{1}{\eta_k}$$

*for all* $i$. *Let* $j$ *be the index of the base learner with the smallest regret. If* $\lambda_{min} \geq \frac{1}{T^{1/8}}$, *then w.p at least* $1 - 3\alpha$, *the regret of Algorithm 2 under event* $\mathcal{E}$ *satisfies:*

$$Regret_T \leq \tilde{O}\left(\frac{\sqrt{dT}}{\epsilon \lambda_{min}^2}\right)$$

*If* $0 \leq \lambda_{min} < \frac{1}{T^{1/8}}$, *then w.p at least* $1 - 3\alpha$, *the regret of Algorithm 2 under event* $\mathcal{E}$ *satisfies:*

$$Regret_T \leq \tilde{O}\left(\sqrt{d}T^{5/6} + \frac{\sqrt{d}T^{17/24}}{\epsilon}\right)$$

*Proof.* Let us first prove the first statement of Theorem 4.1. If $\frac{1}{T^{1/8}} \leq \lambda_{min}$, then from Lemma F.3, there exists at least a base learner $j$ such that, $Regret_T^j \leq \tilde{O}\left(\frac{\sqrt{dT}}{\epsilon \lambda_j}\right)$. Then, plug in $\eta_i$, $C_i$, and $\alpha_i$ we have

$$Regret_T \leq \tilde{O}\left(\frac{\sqrt{dT}}{\epsilon \lambda_j} + \frac{(\sqrt{d})^2}{(\epsilon \lambda_j)^2}T\frac{\epsilon}{\sqrt{dT}} + T\frac{\epsilon}{\sqrt{dT}} + \sum_{k \neq \{1,j\}}\frac{\sqrt{dT}}{\epsilon} + \sqrt{dT}\left(1 + \frac{1}{\epsilon T^{1/8}}\right)\right)$$

$$\leq \tilde{O}\left(\frac{\sqrt{dT}}{\epsilon \lambda_j^2}\right) = \tilde{O}\left(\frac{\sqrt{dT}}{\epsilon \lambda_{min}^2}\right)$$

If $\lambda_{min} < \frac{1}{T^{1/8}}$, then the regret of an instance of Algorithm 1 that uses the correct $\lambda_{min}$ is $\tilde{O}\left(\sqrt{d}T^{3/4} + \frac{\sqrt{d}T^{5/8}}{\epsilon}\right)$. From Lemma F.3, we know that the regret of the first base learner is also $\tilde{O}\left(\sqrt{d}T^{3/4} + \frac{\sqrt{d}T^{5/8}}{\epsilon}\right)$. Thus,

$$Regret_T \leq \tilde{O}\left(\sqrt{d}\left(\frac{\epsilon T^{1/8}+1}{\epsilon T^{1/8}}\right)T^{3/4} + \left(\sqrt{d}\left(\frac{\epsilon T^{1/8}+1}{\epsilon T^{1/8}}\right)\right)^{4/3}T\left(\frac{\epsilon T^{1/8}}{\sqrt{dT}(\epsilon T^{1/8}+1)}\right)^{1/3}\right.$$

$$\left. +T\frac{\epsilon T^{1/8}}{\sqrt{dT}(\epsilon T^{1/8}+1)} + \sum_{k \neq 1}\frac{\sqrt{dT}}{\epsilon}\right)$$

$$\leq \tilde{O}\left(\sqrt{d}T^{5/6} + \frac{\sqrt{d}T^{17/24}}{\epsilon}\right)$$

$\square$

## G  Extra experiments details

In this section, we compute the empirical minimum eigenvalue of $x_t x_t^T$ where each $x \in \mathcal{D}_t$ is uniformly sampled from a sphere as in Section 5. We compute this value over multiple settings of $k \in \{5, 25, 100\}$ and $T \in \{200, 2000, 20000\}$ where $k$ is the number of arms and $T$ is the number of iterations. For all experiments, the dimension of the arm $d = 5$ and we compute the minimum eigenvalue using the non-private LinUCB. We report the portion of training where the minimum eigenvalue is $O(1)$ in the table below:

As we can see from Table 1, our experiments in Section 5 is in the favorable settings for Algorithm 1 with high probability, especially when the number of iterations is significantly larger than the number of arms.

| | T=200 | T=2000 | T=20000 |
|---|---|---|---|
| k=5 | 99.5% | 100% | 100% |
| k=25 | 97% | 97.7% | 99.9% |
| k=100 | 88% | 89.1% | 99.8% |

Table 1: Percentage of iterations where the minimum eigenvalue of $\frac{1}{M}\sum_{m=1}^{M} x_{t,m}x_{t,m}^T$ (M is the number of repeated experiments, $x_{t,m}$ is the arm played at iteration $t$ and sampled at experiment m) is $O(1)$ for multiple settings of k and T

# H    Technical results

**Proposition H.1.** *(Proposition 17 in (Zhang & Cutkosky, 2022)) Suppose $\{X_t, F_t\}$ is a Martingale difference sequence such that $\mathbb{E}[X_t^2|F_t] \leq \sigma_t^2$ and $|X_t| \leq b_t$ almost everywhere for all $t$ for some sequence of random variable $\{\sigma_t, b_t\}$ such that $\sigma_t, b_t$ is $F_{t-1}-$ measurable. Then $X_t$ is $(\sigma_t, 2b_t)$ sub-exponential.*

**Theorem H.2.** *(Theorem 19 in (Zhang & Cutkosky, 2022)) Suppose that $\{X_t, F_t\}$ is a vector-valued martingale difference sequence such that $\mathbb{E}[\|X_t\|^2|F_{t-1}] \leq \sigma_t^2$ and $\|X_t\| \leq b_t$ almost everywhere for some sequence $\{\sigma_t, b_t\}$ such that $\sigma_t, b_t$ is $F_{t-1}$-measurable. Let $\nu \geq 0$ be an arbitrary constant. Then with probability at least $1 - \alpha$, for all $t$ we have:*

$$\left\|\sum_{i=1}^{t} X_i\right\| \leq 5\sqrt{\sum_{i=1}^{t} \sigma_i^2 \log\left(\frac{16}{\alpha}\left[\log\left(\left[\sqrt{\sum_{i=1}^{t} \sigma_i^2/\nu^2}\right]_1\right) + 2\right]^2\right)}$$
$$+ 23\max(\nu, \max_{i\leq t} b_i)\log\left(\frac{224}{\alpha}\left[\log\left(\frac{2\max(\nu, \max_{i\leq t} b_i)}{\nu}\right) + 2\right]^2\right)$$

*where $[x]_1 = \max(1, x)$.*

**Lemma H.3.** *(Lemma 11 in (Abbasi-Yadkori et al., 2012)) Let $x_1, \ldots, x_n \in R^d$ and let $V_t = I + \sum_{s=1}^{t} x_s x_s^T$, then it holds that*

$$\sum_{t=1}^{T} \min\left\{1, \|x_t\|_{V_{t-1}^{-1}}^2\right\} \leq 2\log(det(V_T))$$

*Furthermore, if $\|x_t\|_2 \leq X$ for all $t$ then*

$$\log(det(V_T)) \leq d\log\left(1 + \frac{TX^2}{d}\right)$$

**Lemma H.4.** *(Lemma 1 in (Wang et al., 2020a)) Define $s_t^r(x) = \eta(x_t - x)^T g_t + \eta^2 G^2\|x_t - x\|^2$. For every grid point $\eta$, we have:*

$$\sum_{t=1}^{T} s_t^r(x_t) - s_t^r(x_t^{\eta,s}) \leq 2ln\left(\sqrt{3}\left(\frac{1}{2}\log_2 T + 3\right)\right)$$

**Lemma H.5.** *(Lemma 2 in (Wang et al., 2020a)) Define $s_t^r(x) = \eta(x_t - x)^T g_t + \eta^2 G^2\|x_t - x\|^2$. For every grid point $\eta$ and any $u \in R^d$, we have:*

$$\sum_{t=1}^{T} s_t^r(x_t^{\eta,s})) - s_t^\eta(u) \leq 1 + \log T$$

**Lemma H.6.** *(Corollary 2 and Theorem 1 in (Wang et al., 2020a)) Suppose the loss function is $G-$Lipschitz and the diameter of domain is bounded by $D$. For $\mu-$SC functions, the regret of Maler is upper bounded by*

$$R(T) \leq \left(10GD + \frac{9G^2}{2\mu}\right)\left(2ln\left(\sqrt{3}\left(\frac{1}{2}\log_2 T + 3\right)\right) + 1 + \log T\right)$$
$$\leq O\left(\frac{1}{\mu}\log T\right)$$

*For general convex loss, the regret of Maler is bounded by:*

$$R(T) \leq \left(2ln3 + \frac{3}{2}\right)GD\sqrt{T}$$

**Theorem H.7.** *(Exercise 4.7.3 - (Vershynin, 2018)) Let $x_1, \ldots, x_n$ be an i.i.d sequence of $\sigma$ sub-gaussian random vectors such that $\Sigma = \mathbb{E}[x_i x_i^T]$ for all $i$ and $\hat{\Sigma}_n = \frac{1}{n}\sum_{i=1}^n x_i x_i^T$ be the emprical covariance matrix. Then there exists a universal constant $C > 0$ such that for $\alpha \in (0, 1)$, w.p at least $1 - \alpha$*

$$\|\hat{\Sigma}_n - \Sigma\|_{op} \leq C \max\left\{\sqrt{\frac{d + \log(2/\alpha)}{n}}, \frac{d + \log(2/\alpha)}{n}\right\}\|\Sigma\|_{op}$$

**Proposition H.8.** *(Square-root trick (Abbasi-Yadkori et al., 2012)) Let $a, b \geq 0$. If $z^2 \leq a + bz$, then $z \leq b + \sqrt{a}$.*

**Proposition H.9.** *(Logarithmic Trick (Abbasi-Yadkori et al., 2012)) Let $c \geq 1$, $f > 0$, $\alpha \in (0, 1/4]$. If $z \geq 1$, and $z \leq c + f\sqrt{\log(z/\alpha)}$ then $z \leq c + f\sqrt{2\log\left(\frac{c+f}{\alpha}\right)}$*

**Theorem H.10.** *Let $X \sim N(\mu, \sigma^2 I)$ where $\mu \in R^d$ and $\sigma^2 \in R$. Then:*

$$P[\|X - \mu\|_2 > t] \leq 2\exp\left(-\frac{t^2}{2d\sigma^2}\right)$$

**Theorem H.11.** *(Self-normalized bound for martingales (Abbasi-Yadkori et al., 2012)) Let $\{F_t\}_{t=1}^\infty$ be a filtration. Let $\tau$ be a stopping time w.r.t the filtration $\{F_{t+1}\}_{t=1}^\infty$ i.e the event $\{\tau \leq t\}$ belongs to $F_{t+1}$. Let $\{Z_t\}_{t=1}^\infty$ be a sequence of real-valued variables such that $Z_t$ is $F_t-$measurable. Let $\{r_t\}_{t=1}^\infty$ be a sequence of real-valued random variables such that $r_t$ is $F_{t+1}-$measurable and is continuously $R-$sub-Gaussian. Let $V > 0$ be deterministic. Then, for any $\alpha > 0$, with probability at least $1 - \alpha$*

$$\frac{\left(\sum_{t=1}^\tau r_t Z_t\right)^2}{V + \sum_{t=1}^\tau Z_t^2} \leq 2R^2 \log\left(\frac{\sqrt{V + \sum_{t=1}^\tau Z_t^2}}{\alpha\sqrt{V}}\right)$$

**Corollary H.12.** *(Uniform Bound (Abbasi-Yadkori et al., 2012)) Under the same assumptions as in Theorem H.11, for any $\alpha > 0$, w.p at least $1 - \alpha$, for all $n \geq 0$,*

$$\left|\sum_{t=1}^n r_t Z_t\right| \leq R\sqrt{2\left(1 + \sum_{t=1}^n Z_t^2\right)\log\left(\frac{\sqrt{1 + \sum_{t=1}^\tau Z_t^2}}{\alpha}\right)}$$

**Corollary H.13.** *((Vershynin, 2018)) Let $A$ be an $n \times n$ symmetric random matrix whose entries $A_{ij}$ on and above the diagonal are independent, mean zero, sub-gaussian random variables. Then, for any $t > 0$ we have*

$$\|A\| \leq CK\left(\sqrt{n} + t\right)$$

*with probability at least $1 - 4\exp(-t^2)$. Here $K = \max_{i,j} \|A_{ij}\|_{\psi_2}$ and $\|X\|_{\psi_2}$ is defined as follows:*

$$\|X\|_{\psi_2} := \inf\left\{t > 0 : \mathbb{E}\left[\exp(X^2/t^2)\right] \leq 2\right\}$$

*Specifically, if $X \sim N(0, \sigma^2)$, then:*

$$\|X\|_{\psi_2} \leq C\sigma$$

*for some $C > 0$.*

**Theorem H.14.** *(Theorem 5 in (Gordon et al., 2006)) Let $(x_i)_{i=1}^n$ be a sequence of real numbers and $\zeta_1, \ldots, \zeta_n$ be random variables satisfying the following condition:*

$$P\left[|\zeta_i| \leq t\right] \leq \alpha t \ \forall i \in [n]$$

*where $t \geq 0$ and $\alpha > 0$. Let $p > 0$, then:*

$$\frac{1}{1+p}\alpha^{-p}\left(\sum_{i=1}^n \frac{1}{|x_i|}\right)^{-p} \leq \mathbb{E}\left[\min_{1\leq i \leq n} |x_i \zeta_i|^p\right]$$

**Lemma H.15.** *If $\zeta \sim N(0, \sigma^2)$, then:*

$$P\left[|\zeta| \leq t\right] \leq \frac{\sqrt{2}}{\sigma\sqrt{\pi}}t$$

*Proof.* Using the CDF of Gaussian random variable we have:

$$P\left[|\zeta| \leq t\right] = \Phi\left(\frac{t}{\sigma}\right) - \Phi\left(\frac{-t}{\sigma}\right)$$

$$= \frac{1}{2}\left[1 + \mathrm{erf}\left(\frac{t}{\sigma\sqrt{2}}\right)\right] - \frac{1}{2}\left[1 + \mathrm{erf}\left(\frac{-t}{\sigma\sqrt{2}}\right)\right]$$

Since $\mathrm{erf}(\cdot)$ is an odd function:

$$= \mathrm{erf}\left(\frac{t}{\sigma\sqrt{2}}\right)$$

$$= \frac{2}{\sqrt{\pi}}\int_0^{\frac{t}{\sigma\sqrt{2}}} \exp(-x^2)dx$$

$$\leq \frac{2t}{\sigma\sqrt{2\pi}}$$

$$= \frac{t\sqrt{2}}{\sigma\sqrt{\pi}}$$

$\square$

**Theorem H.16.** *(Azuma-Hoeffding Inequality) For a sequence of Martingale Difference Sequence random variable $\{D_t\}_{t=1}^\infty$ with respect to some other sequence of random variable $\{X_t\}_{t=1}^\infty$, if we have $D_t \in [a_t, b_t]$ almost surely for some constants $a_t, b_t$ and $t = 1, 2, \ldots, T$, then:*

$$P\left[\sum_{t=1}^T D_t \geq \epsilon\right] \leq \exp\left(\frac{-2\epsilon^2}{\sum_{t=1}^T (b_t - a_t)^2}\right)$$

