# OpenReview forum: "Improved rate for Locally Differentially Private Linear Bandits"
_TMLR — Rejected by TMLR_

### Review · Reviewer_yEpJ · 2024-04-15

**Summary Of Contributions:**

The paper proposes a locally differentialy private variant of LinUCB.

**Audience:**

Yes

**Broader Impact Concerns:**

No concerns

**Claims And Evidence:**

No

**Requested Changes:**

First, the paper needs to be made complete (adding an appendix if needed), and the writing needs to be made rigorous and self-contained.
Then, the claims in the paper should be corrected, i.e., the authors should avoid incorrect claims, and position their work in relation to the state of the art (e.g., why can / can't the existing $O(\sqrt{T})$ approach for the non-private setting be applied here, given that LDP in contrast to central DP preserves the rounds to be i.i.d. ?).

**Strengths And Weaknesses:**

### Evaluation

While not a TMLR criterion, it is unclear whether the contribution is novel.  In particular, an $O(\sqrt{T})$ regret bound is known for the non-private case, which may already be noisy (the authors point to [Li et al 2019] and [Lattimore & Szepesvari 2020]), and local differential privacy typically comes down to just making each training instance more noisy using the same privatization mechanism.  Hence, the existing $O(\sqrt{T})$ result seems applicable to the LDP setting too.  In particular, suppose we are using the Gaussian Mechanism to make x_t and y_t LDP, then $y_i=<\theta^*,x_t>+\eta_t$ can be obtained by letting $\eta_t$ include not only the "natural" noise but also the privatization noise.

While TMLR doesn't require novelty, it requires correct claims.  The paper contains a few incorrect claims or technical mistakes, which probably all can be corrected.   In fact, the claim isn't precisely stated as the problem statement section 2 doesn't precisely say how the local differential privacy applies in the considered setting.  A few variations are possible depending on who are the individuals/users considered (e.g., are the "rounds" the individuals/users?).

In general, the evidence supporting the claim (the algorithm, the derivations) are not sufficiently clear to meet the TMLR criterion.  It is probably possible to meet the criterion by (much) more rigorous writing.  Also, maybe an appendix (including Algorithm 3?) seems missing.


### Details

#### Abstract

* "(SOTA) (Zheng ...)" : It is awkward to have open/close brackets closely following, I suggest to either drop the acronym SOTA and introduce it in the body of the paper (preferred) or else separating by a ",", i.e., "(SOTA), (Zheng ...)".
* The word "rate" may be suboptimal here.  why not use "guarantee" or "bound" ?
* "privacy for free" : this is not fully true.  Privacy always comes at a cost, even if in this case the cost may be a constant factor and hence invisible in your big O notation.
* I wouldn't call local DP "stronger" than central DP without further details, in fact, sometimes central DP is called stronger as it requires less noise and hence gives a stronger utility/privacy trade-off.

#### Section 1

* Second line: as "learner" is already introduced (using correctly "a learner"), it seems better to use in the second sentence "the reader" (referring to the former sentence).
* First equation $Regret_T=...$ : the second equation is not necessarily correct if $\phi$ is injective.
* "doing as well as the best possible actions" : this isn't true as the regret may still be non-zero.
* "the optimal regret is O(..)" -> "the asymptotically optimal regret is O(...)"
* At the bottom of page 1, the authors argue that in central DP, data is sent and seen by a trusted central curator.  While this is one scenario where this "central DP" level of noise is needed, it is only one of the options.  There have been several proposals, e.g., using multi-party computation or other encryption techniques, where there is no central party seeing the data, but where still the samen central DP level of noise is needed on the decisions / models inferred from that data which are revealed as output.  So moving to a threat model where you don't trust the central party can be compensated by moving to a better encrypted computation without affecting the statistical privacy of the output.
* "Roughly speaking, LDP ensures that the output of a randomized algorithm using any pair of users" -> ... on any pair of users
* The authors make additional claims, such as "local DP is more user-friendly than (central) DP", without providing any evidence for it.  Please note I'm not claiming that local DP is not interesting, I'm only saying that the authors give incorrect arguments to justify it.
* "Another line of work on private stochastic contextual linear is" : Given that "linear" is an adjective, a noun is needed, e.g., "linear bandits".
* The central question "Is it possible to achieve $O(\sqrt{T})$ regret in a stronger privacy setting than the central model?" is therefore ill-formulated, as "stronger" isn't fully accurately defined.  Maybe you just want to say "in the local differential privacy model" (which is exactly what you research in this paper); if you want to really make the comparison to other threat models, you could say something like "a more malicious threat model than ... while assuming adversaries have unbounded computational power and there is no setup phase to exchange encryption keys in a trusted way" (the "while ..." condition eliminates the encryption-based strategies suggested above but must be carefully formulated).


#### Section 2

* Are you sure the learner receives in every round a new / potentially different function $\phi$ ?  If so, it may be better to denote it by $\phi_t$ ?
* "$\eta_t$ is zero-mean ... variable" -> ... is a zero-mean ...
* Please define $R$ or $R^2$.
* "to hide additional logarithmic terms." : do you mean "... logarithmic factors" ?
* Please define $D$ (apparently an upper bound for $\\|\theta^*\\|$, but according to Algorithm 1 also the diameter of an unspecified domain)
* In the problem statement section, please define accurately the implications of applying local differential privacy.

#### Section 3

* At the point the text refers to Algorithm 1, several variables in that algorithm are not yet defined
    - The probability distribution over $x_i$ is unclear in $\lambda_{min} = \mathbb{E}[x_i x_i^\top]$
    - The failure parameter $\beta$ isn't introduced early in Section 3, but given that $\beta_t$ is said to be the width of some confidence set it is probably a vector of dimension T.
    - $C$ is said to be an universal constant without further explanation, even if from the text we can infer that its component $C_t$ is a confidence set (and hence probably here too $C$ is a vector of dimension $T$).
* Eq (1) assumes the matrix $V_{t-1}$ can be inverted, which is unlikely if $t$ is small (as the rank of $V_{t-1}$ is at most $t$ as $V_t$ is the sum of $t$ rank-one matrices.
* The text is not self-contained, e.g., Eq (2) is justified using a citation where the context/pre-conditions may not be exactly the same as in the current work.
* In general, at least a self-contained explanation of Algorithm 1, defining and explaining all used variables and techniques is desirable.
* Please make sure that at each point in the text it is clear whether you are discussing an existing method, an adaptation of an existing method, or the newly proposed method.  Currently, the structure of the text is rather unclear.  Section 3 first says "Our private method described in Algorithm 1" and then after some discussion says on the next page "In this section, we propose a new approach for designing the confidence sets LDP LinUCB. Our approach is based on the online-to-confidence-set conversion ...".  So is this "new approach" now different from "Algorithm 1" ?  After this claim, we get Equations such as (3) and (4) introducing variables which didn't occur in Algorithm 1 and citations to various earlier papers without making clear what is the implication on the current work.
* "Maler (described in Algorithm 3)" : I don't find this Algorithm, which is referenced twice and also used in Algorithm 1.

---

> ### Author Response · Authors · 2024-04-20
>
> We thank the reviewer for the detailed review and many insightful comments. We would incorporate the reviewer’s suggestions in the final version of this paper. We will address your concerns below.
>
> - Concerns in the evaluation:
>
> 1. 'Can we use simple LinUCB for LDP': We would like to stress that guaranteeing $\tilde O(\sqrt{T})$ regret in any DP setting that is stronger than the central DP is definitely not a trivial task. In fact, we are not aware of any result that was able to achieve this regret bound. [1][2][3] all failed to achieve this. It’s true that by increasing $\eta_t$ in $y_t = \langle \theta^\star, x_t \rangle + \eta_t$, we can in fact make the reward $y_t$ private. However, $x_t$ is not private if we just do this. Now, if we want to naively add noise to $x_t$ to make it private, then as discussed in section 3 of our paper, the regret bound would be $\tilde O(T^{3/4})$ like in [3] instead of $\tilde O(\sqrt{T})$.
>
> 2. 'Is ‘round’ a user': Yes, in each “round”, our algorithm interacts with user $t$ so we can think of a ‘round’ as one interaction with a user $t$.
>
> 3. 'Problem setting': Sorry for the confusion! We quickly mentioned we want to make the action and the reward private in the motivating example in section 1 and also in the proof of theorem 3.2 but we will try to make our problem setting clearer.
>
> 4. 'Missing appendix': The appendix is submitted through the supplementary material (and the description of algorithm 3 is also in there). Sorry for the confusion!
>
> - Concerns in the abstract:
>
> 1. 'The use of rate': We borrow the word ‘rate’ from the optimization literature where the use of convergence rate/bound is interchangeable. However, we are happy to change it to the reviewer’s suggestion if the reviewer think it is more appropriate for our results.
>
> 2. 'privacy for free': You’re absolutely right that privacy always comes with some additive cost! What we mean by this is the regret bound is dominated by the utility term instead of the privacy cost term, thus we have privacy ‘for free’ in the sense that privacy doesn’t make the regret bound asymptotically worse.
>
> 3. 'LDP is stronger than central DP': Please refer to our response in section 1.
>
> - Concerns in section 1:
>
> 1. 'Learner': The reason we use “learner” is because it’s a standard way to refer to an online optimization algorithm (refer to section 1 in [4]). We will try to not repeat this notion too much.
>
> 2. '$Regret_T$ is not correct': We actually don’t need $\phi$ to be injective. After receiving $c_t$, we will compute a set $\mathcal{D}$ containing all possible values of $\phi(c_t,\cdot)$, then we can pick the best value from this set (if there’s a tie, we can arbitrarily pick any value out of all equally good values). This is the standard definition of regret from [1], [2], [3], and [5].
>
> 3. "doing as well as the best possible actions":  You’re right that the regret is non-zero (in fact, it’s almost guaranteed to be non-zero). What we mean here is if the regret is $O(\sqrt{T})$ (or just sublinear), then the average regret is $O(1/\sqrt{T})$ and if we have a lot of users, this average regret would be roughly zero so our learner would do almost as well as if we know the best possible action. It’s an unfortunate convention in the field to call algorithms with this property “zero-regret”, but you’re right it’s imprecise and we’ll add this qualifier.
>
> 4. 'Multiple settings of central DP': This is a good point that in some cases we can replace statistical guarantees with computational hardness - thanks for bringing it up. We’ll add a discussion of how one can also other cryptographic notions such as multiparty computation or homomorphic encryption can be used to deal with untrusted central servers. However, we’d like to emphasize that using LDP usually results in simpler and more efficient algorithms while also providing statistical rather than computational guarantees.
>
> 5. 'local DP is more user-friendly than (central) DP':  Due to the lack of space, we decided to cite [6] instead of discussing it in the paper. Overall, LDP is appealing since it doesn’t require a trusted server and could be efficiently implemented in practice.
>
> 6. 'LDP is stronger than Central DP': When we say LDP is stronger, it means that any algorithm that is LDP is also Central DP but the converse is not true. This is a fairly standard result [3,7] and can be found in section 12.1 of [8].
>
> - Concerns in section 2:
> Thanks for all the suggestions!
>
> Due to the lack of space, we will continue our response in the next comment.

---

> ### Author Response · Authors · 2024-04-20
>
> - Concerns in section 3:
>
> 1. 'The probability distribution over $x_i$  is unclear': We don’t define the distribution of $x_i$ because we don’t really know it. Our algorithm would work with any distribution of $x_i$ as long as the action set is finite. We will clarify that there is some unknown (and time-varying) distribution for each $x_i$.
>
> 2. '$\beta$ vs $\beta_t$/ $C$ vs $C_t$': Sorry for the confusion,$\beta$ and $\beta_t$ are both scalar constants. $\beta$ is defined in Algorithm 1 but we will make sure to mention it sooner. $C_t$ is a set and $C$ is a universal scalar constant. What we mean by that is there always exists a constant $C>0$ such that our statement is true. This is a common statement in the optimization literature [9] but we will try to make it clearer and also change the notations so there's no confusion.
>
> 3. '$V_t$ is invertible': You’re right that $ \sum_{i=1}^t \tilde x_t \tilde x_t^T$ could be non-invertible (though the independent Gaussian noise added in $x_t$ might help). However, notice that we initialize $\tilde V_{0} = I$ in Algorithm 1, thus $\tilde V_t$ is actually $\sum_{i=1}^t \tilde x_t \tilde x_t^T + I $. Since the sum of the outer products is PSD, $\tilde V_t$ is always invertible. This is the trick to ensure invertibility from the original LinUCB paper [10].
>
> 4. 'Text is not self-contained': Eq.2 uses the result from Proposition 4 in [5] and the conditions are the same. We will add this result in the appendix.
>
> 5. 'Eq (3) and (4)': We define $M_T$ and $Regret_{OL}$ a few lines above the equations since they are not the variables of the algorithm but the variables of the theoretical analysis. The loss $l_t$ is defined in line 16 of Algorithm 1 and also in the paragraph above Eq (3) and (4).
>
> - Requested change:
>
> Please refer to the first section of our response. If we only naively use LinUCB then $x_t$ is not private, thus making the algorithm not LDP. If we were to add noise to $x_t$ to make the algorithm LDP, then the noise would damage the regret bound to no longer be $O(\sqrt{T})$. Also, the rounds are not $i.i.d$ since the actions/rewards of earlier rounds influence later rounds. The only $i.i.d$ variable of our setting is the context $c_t$ that we receive in every round.
>
> We thank the reviewer once again for the time and the feedback! We would greatly appreciate if the reviewer would reassess the work in light of the response, and are happy to continue the discussion if there are further questions or concerns.
>
> [1] Chowdhury, Sayak Ray, and Xingyu Zhou. "Shuffle private linear contextual bandits." arXiv preprint arXiv:2202.05567 (2022).
>
> [2] Garcelon, Evrard, et al. "Privacy amplification via shuffling for linear contextual bandits." International Conference on Algorithmic Learning Theory. PMLR, 2022.
>
> [3] Zheng, Kai, et al. "Locally differentially private (contextual) bandits learning." Advances in Neural Information Processing Systems 33 (2020): 12300-12310.
>
> [4] Shalev-Shwartz, Shai. "Online learning and online convex optimization." Foundations and Trends® in Machine Learning 4.2 (2012): 107-194.
>
> [5] Shariff, Roshan, and Or Sheffet. "Differentially private contextual linear bandits." Advances in Neural Information Processing Systems 31 (2018).
>
> [6] Cormode, Graham, et al. "Privacy at scale: Local differential privacy in practice." Proceedings of the 2018 International Conference on Management of Data. 2018.
>
> [7] Kang, Yilin, et al. "Input perturbation: A new paradigm between central and local differential privacy." arXiv preprint arXiv:2002.08570 (2020).
>
> [8] Dwork, Cynthia, and Aaron Roth. "The algorithmic foundations of differential privacy." Foundations and Trends® in Theoretical Computer Science 9.3–4 (2014): 211-407.
>
> [9] Pacchiano, Aldo, Christoph Dann, and Claudio Gentile. "Data-Driven Regret Balancing for Online Model Selection in Bandits." arXiv preprint arXiv:2306.02869 (2023).
>
> [10] Abbasi-Yadkori, Yasin, Dávid Pál, and Csaba Szepesvári. "Improved algorithms for linear stochastic bandits." Advances in neural information processing systems 24 (2011).

---

> > ### Comment · Reviewer_yEpJ · 2024-05-16
> >
> > > The probability distribution over $x_i$ is unclear': We don’t define the distribution of $x_i$ because we don’t really know it. Our algorithm would work with any distribution of $x_i$ as long as the action set is finite. We will clarify that there is some unknown (and time-varying) distribution for each $x_i$.
> >
> > If the distribution on $x_i$ is unknown, how then can you provide as input to algorithm 1 the smallest eigenvalue of $\mathbb{E}[x_i x_i^\top]$ ?  This input is required in Algorithm 1 according to the current writing.
> >
> > * Maybe you don't know the full distribution of $x_i$ but you know the exact value of the smallest eigenvalue $\lamda_{min}$ using a method you didn't explain.  In that case, please explain how you compute $\lambda_{min}$.
> > * Maybe the algorithm doesn't really need as input the smallest eigenvalue of $\mathbb{E}[x_i x_i^\top]$, but only needs a lower bound or upper bound for this $\lambda_{min}$.  In that case, please correct the algorithm description.
> >
> > > What we mean by that is there always exists a constant $C>0$ such that our statement is true.
> >
> > This may make sense.  It is another example of my comment about "vague writing", please write what you mean, not an approximation of what you mean.
> >
> > > Please refer to the first section of our response. If we only naively use LinUCB then
> >
> > Let's ignore the fact that I'm not aware the which first section of which response you refer, no first section seems to offer more clarity.  Before going into this argument, please ensure the text makes crystal clear what the application of LDP to this bandit setting means.  The first few lines of Section 1 explain the considered bandit setting.  Definition 1.1 defines LDP abstractly.  Immediately after definition 1.1, the text starts to argue about achieving a regret bound under LDP.  Ideally, between definition 1.1 and this discussion, the text should first explain what are the instances and what the application of LDP to this setting exactly means.
> >
> >
> >
> > > We thank the reviewer once again for the time and the feedback! We would greatly appreciate if the reviewer would reassess the work in light of the response, and are happy to continue the discussion if there are further questions or concerns.
> >
> > In general, as pointed out in my general comment before seeing your answer, while there is progress on the specific comments I listed, please consider I only offered a small sample of unclear sentences in the text.  A reviewer is not a professional proofreader who should find each and every writing problem.  It is the task of the authors to proofread their paper, and only the task of the reviewer to assess (for example by taking a sample) whether the paper fulfills the criteria.

---

> > > ### Author Response · Authors · 2024-05-16
> > >
> > > "If the distribution of $x_t$ is unknown, how can we know the eigenvalue of $\E[x_tx_t^T]$": First, we want to say that you're right that if we don't have any information about the distribution of $x_t$, then it's not likely to know the exact value for $\lambda_{min}$.  It might be possible to empirically evaluate it but we have to sacrifice some privacy guarantee for it. This is the weakness of Algorithm 1 and we have discussed that in the first paragraph of Section 4. That is why we come up with a new variant of Algorithm 1 (Algorithm 2) that can adapt to different values of $\lambda_{min}$ on the fly. Using Algorithm 2, if $\lambda_{min} \approx O(1)$, then Algorithm 2 automatically guarantees $\tilde O(\sqrt{dT})$ regret without requiring any information on the distribution of $x_t$. Furthermore, as we discussed in the final sentence of Section 4, if we have no information of $x_t$ and still want to use Algorithm 1, we can just set $\lambda_{min} = 0$ and Algorithm 1 guarantees $\tilde O(\sqrt{d}T^{3/4})$ regret which is still $O(d^{1/4})$ better than the previous state-of-the-art bound.
> > >
> > > "Absolute constant C": We agree that it is better to have a closed-form formula for $C$ but since the expression for $C$ is quite complicated, we decide to do this to simplify the result. We are happy to provide a more explicit formula for all the constants. We note that the use of an absolute constant in the upper bound is fairly standard in theory work. See Lemma B.1 of [1] or Proposition 2.6.1/2.7.1 of [2]
> > >
> > > "First section of our response": Sorry for the confusion! We were referring to our response to your question "Can we use simple LinUCB for LDP". And the answer is no. It’s true that by increasing $\eta_t$ in $y_t = \langle \theta^\star, x_t \rangle + \eta_t$, we can in fact make the reward $y_t$ private. However, $x_t$ is not private if we just do this. Now, if we want to naively add noise to $x_t$ to make it private, then as discussed in section 3 of our paper, the regret bound would be $\tilde O(T^{3/4})$ like in [3] instead of $\tilde O(\sqrt{T})$.
> > >
> > > "Application of LDP": As we stated in the third paragraph of the Introduction and the second paragraph of Section 2, the goal of LDP is to protect every user's information (including the reward, action, and context). Since the focus of our paper is more on the theory, for real-world application, we only discuss two examples and refer interested readers to a more comprehensive review [3]. However, we are happy to add more examples in the appendix.
> > >
> > >
> > >
> > >
> > > [1]  Pacchiano, Aldo, Christoph Dann, and Claudio Gentile. "Data-Driven Regret Balancing for Online Model Selection in Bandits." arXiv preprint arXiv:2306.02869 (2023).
> > >
> > > [2] Vershynin, Roman. High-dimensional probability: An introduction with applications in data science. Vol. 47. Cambridge university press, 2018.
> > >
> > > [3] Cormode, Graham, et al. "Privacy at scale: Local differential privacy in practice." Proceedings of the 2018 International Conference on Management of Data. 2018.

---

> ### Comment · Action_Editor_KzCi · 2024-05-16
>
> Dear authors,
>
> You have corrected the Readers part of the comment above, but please note that there are still two comments (one for Reviewer yEpJ and the other for Reviewer xfSC) that have not been updated accordingly.
>
> AE

---

### Review · Reviewer_xfSC · 2024-04-24

**Summary Of Contributions:**

In this paper, the authors propose a stochastic linear contextual bandit algorithm that ensures local differential privacy (LDP). Their algorithm is $(\epsilon,\delta)$−Locally Differentially Private and guarantees $\sqrt{d} T^{3/4}$ regret with high probability. According to their reference search, this is a factor of $d^{1/4}$ improvement over the previous state-of-the-art (SOTA)(Zheng et al., 2020).

Furthermore, their regret guarantee improves to $O(\sqrt{dT})$ when the action space is well-conditioned. This rate matches the optimal non-private asymptotic rate, thus demonstrating that their work can achieve privacy for free even in the stringent LDP model.

Their algorithm structure is also very simple. They are based on LinUCB and for privacy, they added intentional noise on actions and rewards. After that, they measure the loss using a 'negative regularizer' which is an interesting approach. They know that this is not convex, but they argue that this is convex in expectation. Structure itself follows the path of LinUCB, except that for the estimator it depends on the Maler algorithm.

**Audience:**

Yes

**Claims And Evidence:**

Yes

**Requested Changes:**

- It would be great if authors show practical examples where $\epsilon\geq 1$ and $\epsilon \leq \frac{1}{\sqrt{d}}$. In this paper, their algorithm works better only when $\epsilon>1/\sqrt{d}$. If authors suggest several practical examples of $\epsilon>\frac{1}{\sqrt{d}}$ or introduce other works when $\epsilon > \frac{1}{\sqrt{d}}$, it would be more convincing on their result.

- It would be great if the authors wrote more details about the 'classic Gaussian Mechanism' in Theorem 3.2.

- Question: What is the special point of the Maler optimization that makes authors use Maler optimization? Please introduce multiple other online regression methods that are not working well in this scenario.

- Question: What is the benefit of the gradient clipping? According to Corollary B.1, it seems like the gradient above the threshold is already a 'terribly bad event.' What is the benefit of using gradient clipping?



Minor typo
- second last paragraph of page 4, $l_t(\theta)$ and $l_t (\theta_t)$ are not consistent.

**Strengths And Weaknesses:**

Strength

- They devised a new algorithm with a stronger bound, even $\sqrt{T}$ regret bound for differential private algorithm in a specific case.
- Clear demonstrations of their contribution, explanation, and challenges on page 4.
- Simple algorithm, except for the use of Maler.


Weakness

- Their algorithm heavily relies on the online regression algorithm, Maler optimization. Though they briefly explained the advantage of using Maler optimization on page 5, I think it would be great why 'they should use Maler optimization', and not other optimization. I guess it is because Maler optimization works well even if the loss is convex 'in expectation?'
- (Minor)

---

> ### Author Response · Authors · 2024-04-29
>
> We thank the reviewer for their detailed reading and comments. We appreciate all the suggestions and will incorporate them in the final revision. We address the comments and questions below.
>
> 1. Why Maler: Thanks for the suggestion! We will add a discussion on why we chose Maler in the appendix. There are 2 main reasons for this choice:
>
> + Maler can adapt to various types of loss functions, including convex, strongly convex, and exp-concave, without the need for manual adjustment of its hyperparameters. It ensures an optimal regret bound automatically for each loss setting. On the other hand, other popular online algorithms like Online Gradient Descent (OGD) can guarantee optimal regret bounds for various loss settings but they usually require knowledge of loss so we can set their hyperparameters.
>
> + Unlike previous adaptive algorithms such as Metagrad (which Maler is based on), Maler guarantees dimension-free regret bounds for strongly convex loss, which is crucial for us to improve the dimensional dependence in the final regret bound from $O(d^{3/4})$ to $O(\sqrt{d})$.
>
> 2. Realistic value of $\epsilon$: Some articles that discuss the practical values of $\epsilon$ are [1][2]. Typically, $\epsilon$ is between 1 and 10. In the industry, Apple uses $\epsilon$ in the range of 4 to 8 [4] and Microsoft uses $\epsilon \approx 1$ [3].
>
> 3. Gradient clipping: Thanks for catching that, you’re right that clipping does not provide any theoretical benefit in our analysis since we do our analysis conditioned on the good event $\mathcal{E}$. At first, we thought we needed to clip the gradient since Maler requires a uniform bound on the gradient norm. However, as you've pointed out, we already condition on the good event $\mathcal{E}$ so the clipping is not necessary. We will make modifications accordingly.
>
> [1] Justin Hsu, Marco Gaboardi, Andreas Haeberlen, Sanjeev Khanna, Arjun Narayan, Benjamin C. Pierce, Aaron Roth:Differential Privacy: An Economic Method for Choosing Epsilon. CSF 2014: 398-410.
>
> [2]Natalia Ponomareva, Sergei Vassilvitskii, Zheng Xu, Brendan McMahan, Alexey Kurakin, and Chiyaun Zhang. 2023. How to DP-fy ML: A Practical Tutorial to Machine Learning with Differential Privacy. In Proceedings of the 29th ACM SIGKDD Conference on Knowledge Discovery and Data Mining (KDD '23). Association for Computing Machinery, New York, NY, USA, 5823–5824. https://doi.org/10.1145/3580305.3599561.
>
> [3]https://cloudblogs.microsoft.com/opensource/2020/05/19/new-differential-privacy-platform-microsoft-harvard-opendp/.
>
> [4]https://www.apple.com/privacy/docs/Differential_Privacy_Overview.pdf.

---

### Review · Reviewer_eMcw · 2024-04-30

**Summary Of Contributions:**

The paper gives a local differentially private (LDP) algorithm for stochastic linear contextual bandits with state-of-the-art $\sqrt{d} T^{3/4}$ regret. They improve this bound further when the action space is well conditioned.

**Audience:**

Yes

**Broader Impact Concerns:**

None.

**Claims And Evidence:**

No

**Requested Changes:**

Please see above.

**Strengths And Weaknesses:**

**Strengths:**

-  The main result is solid, improving over the SOTA

- The techniques and ideas are interesting

**Weaknesses:**

- The condition $\mathbb{E}[x_t x_t^T] \geq \lambda_{min} I$ seems a bit restrictive. In Remark 3.1, isn't $k$ often very large in practice, so that the condition breaks?

- The presentation and writing could use quite a bit of work. I will list some specific examples, but I believe the authors should try to significantly improve the organization, writing, and overall presentation beyond these changes. Some of the suggestions below are not writing-related, but most are:

-- Before diving into the mathematical formulation of the LCB problem, it would be nice to motivate it and introduce the problem at a high level

--Clarify that each user has one sample in your set up; otherwise, the proposed privacy notion seems to be more of a "user-level LDP" than traditional LDP

-- use parenthetical citations: e.g. instead of writing, "real-life application Cormode et al. (2018)." you can write "real-life application (Cormode et al., 2018)."

-- Place Alg 1 pseudocode later in the paper after explaining it at a high level; also within the pseudocode, G can be simplified; also you're missing a return/output step at the end; also explain the if statement at the beginning; also, explain the entire algorithm more clearly

-- Maler should also be described in the main body

-- "For appropriate \beta_t, the optimal parameter is inside the ellipsoid with high probability." Why? Add a citation or a lemma.

-- Explain how the non-private Lin UCB algorithm designs the ellipsoid. Also explain how the Abbasi-Yadkori construction of a confidene set works (maybe in the appendix if space is constrained).

-- When writing $\mathbb{E}[x_t x_t^T] \geq \lambda_{min} I$ on p.4, refer to Remark 3.1 which explains this condition and gives an example.

-- Some of the mathematical claims in the final paragraph on p.4 need proof (or citations or references to proofs that are later provided in this paper)

-- Remark 3.1: why is $\sqrt{dT}$ regret optimal? No lower bound is referenced or provided here.

-- Theorem 3.2 proof: how is $\theta_t$ computed? Privacy is unclear without digging through the appendix and examining Maler, which I did not do. You should keep the privacy proof self-contained in the main body.

-- Comparisons with previous results should come earlier in the paper, perhaps after theorem statements but before the proofs.

-- "regret of $d T^{3/5}$, and matches the rate of the central model" is unclear. What is "the rate" of the central model? Do you mean the optimal rate? SOTA rate?

-- Related to the above, discussion of lower bounds and remaining gaps in the rates appears to be absent from the submission unless I missed it. This is unfortunate.

---

> ### Author Response · Authors · 2024-05-07
>
> We thank the reviewer for their detailed reading and comments. We appreciate all the suggestions on the presentation and will incorporate them in the final revision. We address the comments and questions below.
>
>
> 1. Condition on $k$: You’re right that $k$ could be large in practice but $T$ can be very large too (usually much larger than $k$). Thus as long as $k$ doesn’t grow as $T$ grows, we are still in the good regime. Worst case scenario when $k \ge \Omega(T)$, our algorithm still guarantees $\tilde O(\sqrt{d}T^{3/4})$ which is $O(d^{1/4})$ better than the previous results.
>
> 2. Motivating LCB: Thanks for the suggestion! In the introduction, we briefly discussed the example of a personalized medical app that could be modeled as a Linear Bandits problem. We’d be happy to add more examples if the reviewer think it helps the clarity of our paper.
>
> 3. On Algorithm 1: Sorry for the confusion! Due to lack of space, we didnt really discuss the actual working of the algorithm carefully since our algorithm is almost the same as the standard LinUCB algorithm [1][2]. Instead, we chose to focus on the technical novelties of our algorithm and discussed how they affected the final theoretical guarantees. We will add more discussion on the original LinUCB in the appendix.
>
> 4. “For appropriate \beta_t, the optimal parameter is inside the ellipsoid with high probability”: This is more an algorithm design strategy, introduced by [2] rather than a statement to prove on its own (and we cite [2] several times, including at the beginning of Section 3). If we were to interpret this sentence as a Lemma it would be immediate from setting $\beta_t=\infty$. For this reason, we do not provide a lemma at this point in the paper. Instead, the goal of our algorithm is to carefully choose our points $\theta_t$ and a value of $\beta_t$ such that $\beta_t$ is as small as possible, and yet still the optimal parameter $\theta_\star$ is inside the ellipsoid. This is formally accomplished in Lemma 3.3. We will rephrase the start of the section to make it clearer that this is describing the overall *goal* of our algorithm design rather than a specific result.
>
> 6. Proofs in page 4: Sorry for the confusion! We cited the paper where the results came from in the main text of page 4 (for example, Eq.3, Eq.4 come from [2], cited in the fourth paragraph, or Eq.2 come from [3] cited in the second paragraph). We will try to make these clearer.
>
> 7. Why $\sqrt{dT}$ optimal: It is optimal because the $\tilde O(\sqrt{T})$ is the optimal regret for linear bandit as mentioned in the second paragraph of section 1. The results come from [4][5].
>
> 8. Why is $\theta_t$ private: As shown in Algorithm 1, $\theta_t$ is the output of the online learning algorithm Maler, whose goal is to minimize the loss $l_t(\theta)$ (defined in Algorithm 1).  And since $\theta_t$ is computed using a sequence of private parameters $\tilde x_t$ and $\tilde y_t$,  $\theta_t$ is private by post-processing [6]. This is a very nice property of Differential Privacy (DP). We don’t even have to understand how Maler works, as long as Maler is only allowed to access private information, its output is also private.
>
> 9. Rate: Sorry for being unclear! Whenever we say “rate”, we mean the regret bound. We borrow the word ‘rate’ from the optimization literature where the use of convergence rate is common. However, we are happy to change it to the reviewer’s suggestion if the reviewer thinks it is more appropriate for our results.
>
> 10. Lower bound: Currently there’s no lower bound on LDP linear bandits. The only lower bound that we have is the bound from the non-private literature which we mentioned in section 1 and discussed in section 3.2
>
> We thank the reviewer once again for the time and the feedback! We would greatly appreciate it if the reviewer would reassess the work in light of the response, and are happy to continue the discussion if there are further questions or concerns.
> (Citations in the next comment)

---

> ### Author Response · Authors · 2024-05-07
>
> [1] Yasin Abbasi-Yadkori, David Pal, and Csaba Szepesvari. Online-to-confidence-set conversions and application to sparse stochastic bandits. In Neil D. Lawrence and Mark Girolami (eds.), Proceedings of the Fifteenth International Conference on Artificial Intelligence and Statistics, volume 22 of Proceedings of Machine Learning Research, pp. 1–9, La Palma, Canary Islands, 21–23 Apr 2012. PMLR. URL https://proceedings.mlr.press/v22/abbasi-yadkori12.html.
>
> [2] Improved algorithms for linear stochastic bandits. In J. Shawe-Taylor, R. Zemel, P. Bartlett, F. Pereira, and K.Q. Weinberger (eds.), Advances in Neural Information Processing Systems, volume 24. Curran Associates, Inc., 2011. URL https://proceedings. neurips.cc/paper/2011/file/e1d5be1c7f2f456670de3d53c7b54f4a-Paper.pdf.
>
> [3] Roshan Shariff and Or Sheffet. Differentially private contextual linear bandits. Advances in Neural Information Processing Systems, 31, 2018.
>
> [4] Tor Lattimore and Csaba Szepesvári. Bandit Algorithms. Cambridge University Press, 2020.
>
> [5] Nearly minimax-optimal regret for linearly parameterized bandits.
>
> [6] Cynthia Dwork and Aaron Roth. The algorithmic foundations of differential privacy. Found. Trends Theor. Comput. Sci., 9(3–4):211–407, aug 2014. ISSN 1551-305X. doi: 10.1561/0400000042. URL https://doi.org/10.1561/0400000042.

---

### Author Response · Authors · 2024-05-07
**Comment to All Reviewers**

We thank the reviewers for their detailed comments. We have updated our paper based on the reviewer's suggestions. We highlighted most of our revisions with blue text. Specifically, we made the following modifications:

1. Fixed the typo.

2. Discussions on the original LinUCB and LinUCB with online-to-confidence conversion were added in the appendix.

3. More explanation of how Algorithm 1 works was added in section 3.

4. More usage example of Linear Bandit was added in the Introduction.

5. We made it clearer on the goal of our algorithm and the problem settings. For example, the goal of the algorithm is to keep the sequence $\{(x_1,y_1), \dots, (x_T,y_T)$ LDP.

6. Defined what we meant by LDP was stronger than Central DP.

7. Fixed some of the languages based on the reviewer's suggestions (rate -> regret bound).

Please let us know if the reviewers think the modifications help with the clarity of our paper! We are happy to continue the discussion and make more modifications if necessary.

---

> ### Comment · Reviewer_yEpJ · 2024-05-16
>
> The authors didnt provide a reply specific to my review.  I hence only checked the colored text to see (a) whether it addresses my concerns and (b) whether it introduces new problems.
>
>
> It seems also the revisions often contain vague or unclear language, e.g.,
>
> * "user friendly" is not defined but i believe some users will find it less user friendly to have more noisy output.  Please note that central DP does not necessarily imply the user must trust a central curator (e.g., the users could use appropriate cryptographic techniques to avoid the need to trust)
> * The paper now assumes "access to T unique users", while LDP usually avoids the need to make this assumption.  The new assumption is not justified.
>
> It seems that while the authors fixed some specific language issues i listed in my sample, they didnt perform a global proofread and the colored text introduces new language mistakes, e.g.,
>
> * an algorithm that achieves sublinear regret bound -> an algorithm that achieves a sublinear regret bound
>
> The colored text does not address my concern that the new algorithm may be unnecessary, as specifically for LDP we can see noise additions as independent, and existing algorithms achieving a sublinear regret could be used (and would probably be better)

---

> ### Comment · Action_Editor_KzCi · 2024-05-16
>
> Dear all,
>
> I found that the authors forgot to include the reviewers in the Readers list when submitting some of their Response. As a result, Reviewer xfSC and Reviewer yEpJ are unable to see the Response provided by the authors.
>
> ---
>
> Dear authors,
>
> Please edit your Response or resubmit it to ensure that the reviewers are included in the Readers list.
>
> ---
>
> Dear reviewers,
>
> Once the issue has been resolved by the authors, please take some time to read the Response and discuss it with the authors if necessary. You could delay submitting your final recommendations to accommodate this review.
>
> Best
>
> AE

---

> ### Author Response · Authors · 2024-05-16
> **some issue in the comments**
>
> We did in fact provide a response specific to your review several weeks ago - it seems like you were not able to see it!
> We have asked the action editor to check about this.
>
> However, to address your point about the simple algorithm that applies independent noise to each round and uses a standard $\sqrt{T}$ regret algorithm, we'd like to emphasize that this does not work, and is in fact already discussed in the paper in section 3. Guaranteeing $\tilde O(\sqrt{T})$ regret with LDP is definitely not a trivial task: [1][2][3] all failed to achieve this. If we follow the suggestion to simply add noise to quantities that need to be private such as $x_t$, the regret bound actually increases to $\tilde O(T^{3/4})$, which is the result obtained by [3], instead of our $\tilde O(\sqrt{T})$.
>
> Regarding access to $T$ unique users: It is very straightforward to allow a user to show up more than once: the $k$th time a user shows up we simply add noise that's scaled by $\sqrt{k}$ to their $x_t$ and $y_t$. This will maintain LDP by advanced composition. To see the effect on the regret bound in the naive case that each user shows up $k$ times, notice that this scales up the loss $l_t$ by at most a factor $k$. Thus, the regret of Maler will be multiplied by $k$ and so the final regret will scale by $\sqrt{k}$. If each user shows up a different number of times, then we will have a smooth interpolation to the case that all users show up once.

---

> > ### Comment · Reviewer_yEpJ · 2024-05-16
> >
> > As the action editor pointed out, in order for me to be able to read your comment, you should include me in the reader list of that comment.  Until you give me read access to your comment, I can't answer to it.
> >
> > You say somewhere in section 3, i.e., between page 4 and page 10, you say something related to the use of a classic algorithm with a sublinear regret bound.  In my original reading of the paper, no such comment looked very convincing to me, and in the currently colored text there doesn't seem such discussion.  Can you be more specific where this comment can be found?
> >
> > Independently from this quite important question, please notice that both $O(T^{1/2})$ and $O(T^{3/4})$ are sublinear, so maybe the name "sub-linear" is not optimal for your purpose.

---

> ### Author Response · Authors · 2024-05-16
>
> In section 3.2, we discussed how [1] can only achieve $\tilde O(d^{3/4}T^{3/4})$ by adding independent Gaussian noise to both $x_t$ and $y_t$. The main issue with adding noise to $x_t$ is the blow-up in the matrix $V_t$. Let's say we add $\zeta_t = O(1)$ Gaussian noise to $x_t$ to get $\tilde x_t$. Then $ \tilde V_t = \sum_{i=1}^t (x_ix_i^T + \zeta_ix_i^T + x_i\zeta_i^T + \zeta_i\zeta_i^T) + \lambda I$. Let $G_t = \sum_{i=1}^t \zeta_i\zeta_i^T$. As we can see from section A.1 in the appendix, the way LinUCB works is it guarantees that the optimal parameter $\theta^\star$ is in the ellipsoid $\|\theta - \hat \theta_t\|_{\tilde V_t} \le \beta_t$ with high probability (Eq. 7 in the section A.1). However, now with the added Gaussian noise, $\beta_t$ is not $O(\sqrt{d})$ like in section A.1 anymore but rather $\sqrt{d}+ \|\theta^\star\| _{G_t}\approx O(\sqrt{t})$ due to the concentration of Gaussian (for a more detailed derivation, please refer to Proof of Theorem 2 in [2] or Proposition 4 in [4]). This results in a $\tilde O(d^{3/4}T^{3/4})$ regret bound. That's why we need to come up with another way to derive the confidence bound (which is our Algorithm 1) rather than relying on the classic LinUCB. Other optimal algorithms for stochastic linear bandits such as Thompson Sampling also run into the same issue since they also rely on the $V_t$ to derive the confidence bound [4].
>
> Sublinear regret: You're right that there are different levels of sublinear. We tend to consider any algorithm that guarantees sublinear to be a "working" algorithm due to the convention of online optimization literature that a sublinear regret will do as well as the best competitor on average if the number of training iterations $T$ is large. We are not exactly sure what your concern is since we do not claim that all sublinear algorithms are equal. This is why we use Maler since it's the optimal sublinear algorithm in multiple loss settings. But our algorithms do work for any sublinear online optimization algorithms of user's choice.
>
> [1] Zheng, Kai, et al. "Locally differentially private (contextual) bandits learning." Advances in Neural Information Processing Systems 33 (2020): 12300-12310.
>
> [2] Yasin Abbasi-yadkori, Dávid Pál, and Csaba Szepesvári. Improved algorithms for linear stochastic bandits.
> In J. Shawe-Taylor, R. Zemel, P. Bartlett, F. Pereira, and K.Q. Weinberger (eds.), Advances in Neural
> Information Processing Systems, volume 24. Curran Associates, Inc., 2011.
>
> [3] Shariff, Roshan, and Or Sheffet. "Differentially private contextual linear bandits." Advances in Neural Information Processing Systems 31 (2018).
>
> [4] Abeille, M. &amp; Lazaric, A.. (2017). Linear Thompson Sampling Revisited. <i>Proceedings of the 20th International Conference on Artificial Intelligence and Statistics</i>, in <i>Proceedings of Machine Learning Research</i> 54:176-184 Available from https://proceedings.mlr.press/v54/abeille17a.html.

---

### Author Response · Authors · 2024-05-16
**General comment to reviewers**

As the discussion period is coming to a close, we kindly remind the reviewers to take a look at our responses and our revised paper. We have incorporated the reviewers' suggestions on the writing and have included more background information for our problem settings. We hope that the new changes address the reviewer's concern and we are happy to continue the discussion and make more modifications if necessary. Thank you!

---

### Decision · Action_Editor_KzCi · 2024-06-07

**Recommendation:** Reject

**Comment:**

This paper proposes a stochastic linear contextual bandit algorithm that ensures local differential privacy (LDP). The algorithm guarantees
$(\epsilon,\delta)$-LDP and achieves a regret bound of $\tilde{O}(\sqrt{d} T^{3/4})$ with high probability, which is an improvement over previous algorithms. Additionally, the regret bound improves to $\tilde{O}(\sqrt{Td})$ when the action space is well-conditioned.

After discussions, two reviewers still expressed serious concerns. The writing remains unclear and lacks rigor despite revisions. Additionally, the modeling of local differential privacy (LDP) noise and the condition involving $\lambda_\min$ require further discussions.

Therefore, I recommend that the authors revise their paper based on the reviewers' comments and resubmit it later.

**Audience:**

Yes

**Claims And Evidence:**

After discussions, two reviewers still expressed serious concerns. Firstly, the writing is found to be insufficiently clear and rigorous. Despite the authors addressing many initial comments, the revised version still contains unclear sections.

Secondly, there are significant issues regarding the modeling of local differential privacy (LDP) noise. One reviewer points out that treating LDP noise as simple environmental noise reduces the problem to a classic bandit problem, where existing algorithms offer similar complexity. The authors' explanations did not sufficiently address this concern, leaving it unresolved.

Additionally, the condition involving $\lambda_{\min}$ requires further justification. It appears that the paper may conceal the dimensional dependency on the action-set geometry constant.

**Resubmission Of Major Revision:**

The authors may consider submitting a major revision at a later time.